# Terminal Dimension Reduction for Time Series with Applications

**Alexander Munteanu** [1]    **Matteo Russo** [2]    **David Saulpic** [3]    **Chris Schwiegelshohn** [4]

## Abstract

Terminal embeddings have emerged as a powerful tool for dimension reduction. Given a set of points $P \subset \mathbb{R}^d$, a terminal embedding is a mapping $f : \mathbb{R}^d \to \mathbb{R}^t$ that preserves the pairwise distance between any pair of points $p \in P$ and $q \in \mathbb{R}^d$ up to small distortion under this mapping. Terminal embeddings have been particularly fruitful for constructing $k$-means and $k$-median coresets, where the objective is to find a typically weighted subset $\Omega$ of $P$ such that for any candidate solution, the cost of the clustering objective on $\Omega$ approximates the cost of the clustering objective on $P$ up to small distortion. Unfortunately, these techniques have not been extended to more complicated structures such as clustering time-series data under common straight-line interpolation between measurements. The main issue is that terminal embeddings, arguably the central technique in this line of research, cannot be linear and are thus not immediately suitable to preserve linear structures. In this work, we develop a generalization of terminal embeddings to affine line-segments that overcomes this issue. We showcase their applicability by using our lines-preserving terminal embeddings to obtain the first dimension-free coresets for clustering time-series under the Fréchet distance. The underlying dimension reduction uses Johnson-Lindenstrauss (JL) embeddings, and our experiments indicate that terminal embeddings perform similarly to JL and favorably against PCA for synthetic and real-world time-series, while only terminal embeddings extend pairwise distance preservation to the full ambient space.

[1]TU Dortmund, Germany [2]EPFL, Switzerland [3]CNRS & Université Paris Cité, IRIF, Paris, France [4]Aarhus University, Denmark. Correspondence to: Chris Schwiegelshohn <schwiegelshohn@cs.au.dk>.

*Proceedings of the $43^{rd}$ International Conference on Machine Learning*, Seoul, South Korea. PMLR 306, 2026. Copyright 2026 by the author(s).

## 1. Introduction

Time-series and panel data are ubiquitous in the analysis of physical sensor networks, geo-, and weather-information systems, price forecasting, and the stock market (e.g. Zhang et al., 2007; Chapados & Bengio, 2008; Lucas et al., 2015; Zimmer et al., 2018; Huang et al., 2020; 2021; Wang et al., 2025). Often one is not only interested in single-dimensional time-series, that represent one stock or one physical measurement, such as temperature, over time. Rather, we would like to model a large portfolio of stocks, or entire markets. Similarly, we are interested to combine all kinds of weather information into one single high-dimensional measurement at every time instant, to obtain high-dimensional time-series that represent and provide a holistic perspective on weather development over time. For standard machine learning approaches to work for such data, we face two main challenges, (cf. Paparrizos et al., 2024): First, clustering algorithms were initially developed only for vectors. Standard vector embeddings fail to adequately model even one-dimensional time-series in the presence of asynchronous or missing measurements. Vector encodings of higher-dimensional time-series are even more problematic. Second, resources such as runtime, memory and sample complexity depend heavily on the volume and dimension of the data.

As a solution to the first challenge, the fields of theoretical computer science and computational geometry model time-series data as *polygonal curves*. These connect single measurements or *vertices* by straight-line connections that implicitly interpolate linearly along the continuous timeline. Building upon this representation, they introduced clustering algorithms for time-series under the Fréchet distance with rigorous guarantees (Driemel et al., 2016), though initially limited to one-dimensional time-series. An intriguing direction thus remained open, namely an extension and dimension reduction for high-dimensional time-series to address the second of the above challenges.

First random projections for preserving the Fréchet distance appeared in (Driemel & Krivosija, 2018), followed by a Johnson-Lindenstrauss (JL) equivalent for preserving pairwise distances between curves representing time-series (Meintrup et al., 2019). Albeit with additive errors, this enabled efficient algorithms for metric $k$-median under the Fréchet distance, where centers were restricted to be input

curves. Recently, the analysis was refined (Psarros & Rohde, 2023; 2025) to obtain multiplicative errors for preserving pairwise distances by applying JL-embeddings (Johnson & Lindenstrauss, 1984) to a *skeleton* of the curves. However, the method remained restricted to discrete clustering objectives, and estimating the clustering cost. To the best of our knowledge, no *terminal embeddings* exist for the problem, and these are the missing key to remove dimension dependence in *continuous* clustering of time-series and in data reduction via *coresets*.

Terminal embeddings were originally introduced by (Elkin et al., 2017) and have emerged as an important tool for dimension reduction. Given a set of $n$ points $P \in \mathbb{R}^d$, a terminal embedding with distortion $(1 \pm \varepsilon)$ is a mapping $f \colon \mathbb{R}^d \to \mathbb{R}^t$ ensuring for any $p \in P$ and $q \in \mathbb{R}^d$, we have

$$(1 - \varepsilon)\|p - q\| \le \|f(p) - f(q)\| \le (1 + \varepsilon)\|p - q\|,$$

where $\|x\| = \left(\sum_{i=1}^d x_i^2\right)^{1/2}$ denotes the Euclidean norm. This guarantee is remarkably similar to the JL-lemma, where both $p$ and $q$ are *terminals*, i.e., elements of $P$. But in fact it is significantly stronger: in terminal embeddings, $q$ may be an arbitrary *query* point from the ambient space. For points in $\mathbb{R}^d$, a series of results (Elkin et al., 2017; Mahabadi et al., 2018) ultimately led to optimal terminal embeddings with reduced target dimension $t = \Theta(\varepsilon^{-2} \log n)$ (Narayanan & Nelson, 2019; Cherapanamjeri & Nelson, 2024), matching lower bounds that even hold against the weaker JL-guarantee (Larsen & Nelson, 2017; Alon & Klartag, 2017).

The stronger guarantee given by terminal embeddings gave rise to several applications. Notably, terminal embeddings are at the forefront of data summaries known as coresets (Feldman, 2020; Munteanu & Schwiegelshohn, 2018) for the important $k$-means and $k$-median problems. Given a loss function $\text{cost}(P, C)$, where $C$ is a candidate solution, an $\varepsilon$-coreset is a (typically weighted) subset $\Omega \subseteq P$ such that

$$\text{cost}(P, C) = (1 \pm \varepsilon)\,\text{cost}(\Omega, C), \tag{1}$$

for all candidate solutions $C$. For the $(k, z)$-clustering problem, specifically, $C$ is a set of at most $k$ points and we have $\text{cost}(P, C) = \sum_{p \in P} \min_{c \in C} \|p - c\|^z$, where the special case $z = 1$ corresponds to $k$-median and $z = 2$ corresponds to $k$-means. Terminal embeddings have played a central role in constructing *dimension independent* coresets (Becchetti et al., 2019; Braverman et al., 2022; 2021; Huang & Vishnoi, 2020), including the optimal coreset bounds $\tilde{O}\left(k\varepsilon^{-2} \min(\varepsilon^{-z}, k^{z/(z+2)})\right)$ due to (Cohen-Addad et al., 2022b; Huang et al., 2024). Recently, terminal embeddings led to dimension-free learning rates for $k$-means/median for points, and for curves under discrete DTW and Fréchet distances (Bucarelli et al., 2023; Cohen-Addad et al., 2025b; Krivosija et al., 2025). As remarked in (Huang & Vishnoi, 2020), the standard JL-guarantee seems insufficient to

recover any of the above bounds, making terminal embeddings an indispensable technique for several algorithmic applications that crucially rely on strong dimension reduction guarantees.

Unfortunately, the added power of terminal embeddings comes at a cost: the mapping $f$ cannot be linear. To see this, consider that if $f \colon \mathbb{R}^d \to \mathbb{R}^t$ were a linear mapping and $t < d$, then there must exist a non-zero vector $v$ in the nullspace of $f$. The difference vector $p - q$ between any point $p \in P$ to $q := p + v \in \mathbb{R}^d$, whose norm is the non-zero distance that we would like to preserve, is thus mapped to $0$, precluding any multiplicative approximation.

If we only require the weaker JL-guarantee, the mapping can however be linear; indeed most efficient JL-variants are typically sampled from a distribution over random Gaussian or Rademacher matrices. This aspect of JL-transforms has made it a popular technique for many applications in randomized numerical linear algebra such as low-rank approximation and regression (Woodruff, 2014).

While research on both sketching algorithms for numerical linear algebra and coreset algorithms via terminal embeddings for clustering points have seen substantial progress and success for many tasks, there exists a collection of problems for which our knowledge of dimension reduction methods is currently less well developed. One specific example, which we want to focus in the remainder, is clustering time-series under the Fréchet distance. Given the importance that terminal embeddings have for point clustering objectives, the lack of progress can primarily be attributed to the lack of terminal embeddings for lines and line-segments. Unfortunately line-segments are linear structures and – as mentioned earlier – terminal embeddings are inherently non-linear. We thus ask the intriguing open research question:

*Can the framework of terminal embeddings be extended to preserving distances between lines and polygonal curves lying in $d$-dimensional Euclidean ambient space?*

### 1.1. Goals and Problem Setting

The main goal of our work is to introduce new techniques that extend the guarantees of terminal embeddings for points to the setting of high-dimensional time-series. To this end, we aim to develop a terminal embedding framework for affine lines and polygonal curves in $\mathbb{R}^d$.

A polygonal curve $\pi$ of *complexity* $m$ can be formalized as an ordered sequence of vertices $(p_0, \ldots, p_m) \in \mathbb{R}^d$, where consecutive pairs of vertices are connected by straight-line segments. We view a polygonal curve $\pi \colon [0, 1] \to \mathbb{R}^d$ by fixing $m + 1$ values $0 = t_0 < \ldots < t_m = 1$ such that $\pi(t_i) = p_i$ and defining $\pi(t) = (1 - \lambda)p_i + \lambda p_{i+1}$ where $\lambda = \frac{t - t_i}{t_{i+1} - t_i}$ for $t_i \le t \le t_{i+1}$. The (continuous) Fréchet

distance between two curves $\pi_1$ and $\pi_2$ is then defined as

$$d_F(\pi_1, \pi_2) = \inf_{\alpha, \beta \in \mathcal{T}} \sup_{t \in [0,1]} \|\pi_1(\alpha(t)) - \pi_2(\beta(t))\|,$$

where $\mathcal{T} \subseteq \{f : [0,1] \to [0,1]\}$ is a set of continuous, non-decreasing, and surjective mappings, that are called *reparameterizations*.

Note that the straight-line segments of a polygonal curve are continuous connected compact subsets (closed intervals) in a collection of affine lines. Akin to JL-mappings, known constructions of standard point terminal embeddings do not apply to terminal sets of unbounded cardinality. We thus introduce the following generalization.

**Definition 1.1** (Terminal Embeddings for the Fréchet Distance). Let $P$ be a set of $n$ polygonal curves in $\mathbb{R}^d$ of complexity at most $m$ and let $\mathcal{Q}_\ell$ be the set of polygonal curves in $\mathbb{R}^d$ of complexity at most $\ell$. A mapping $f : \mathbb{R}^d \to \mathbb{R}^t$ is an $(\varepsilon, m, \ell)$-preserving terminal embedding for the Fréchet distance, if for all curves $\sigma \in P$ and $\tau \in \mathcal{Q}_\ell$ it holds that

$$d_F(f(\sigma), f(\tau)) = (1 \pm \varepsilon)\, d_F(\sigma, \tau).$$

### 1.2. Our Results

We aim at terminal embeddings for preserving the Fréchet distance between high-dimensional time-series represented as polygonal curves. Instead of points, terminals as well as queries are (collections of) line-segments, requiring a different construction of the terminal embedding.

We state our main result regarding terminal embeddings for curves with respect to the Fréchet distance.

**Theorem 1.2.** *There exist $(\varepsilon, m, \ell)$-preserving terminal embeddings for the Fréchet distance with target dimension $t \in O(\ell \varepsilon^{-4} \log(nm))$.*

Consider $(k, \ell, z)$-clustering under the Fréchet distance as an application. In light of Equation (1), the loss function is

$$\text{cost}(P, C) = \sum_{\sigma \in P} \min_{\tau \in C} d_F(\sigma, \tau)^z,$$

where the $|C| = k$ center curves $\tau \in C \subseteq \mathcal{Q}_\ell$, have restricted complexity at most $\ell$. Algorithms and coreset constructions for this problem received considerable attention recently (Braverman et al., 2022; Conradi et al., 2024; Cohen-Addad et al., 2025a). The best existing coreset are built via an algorithm whose success rely solely on the *existence* of small enumerations over all candidate clusterings (Cohen-Addad et al., 2025a). The terminal embeddings given in Theorem 1.2 allow, in a black-box manner, to remove the dependence on the dimension of those small enumeration. This black-box application yields the first dimension-free coresets of size $\tilde{O}(k \cdot \varepsilon^{-5-z} \cdot \ell^2 \cdot \log^2 m)$.

Unfortunately, such a naïve application degrades the dependence on the precision parameter $\varepsilon$ severely from $\varepsilon^{-1-z}$ to $\varepsilon^{-5-z}$. However, we improve the size bounds of the coreset construction framework due to Cohen-Addad et al. (2021), and hereby refine the coreset analysis to directly incorporate our terminal embedding, avoiding the additional loss. Again, this framework relies only on the existence of small enumerations over candidate clusterings, and does not need to actually compute them, and in particular it does not need to build the terminal embedding. This ultimately leads us to the following result.

**Theorem 1.3.** *Given a set $P$ of $n$ curves in $\mathbb{R}^d$ of complexity $m$, there exists an $\varepsilon$-coreset of size $\tilde{O}(k \varepsilon^{-2-z} \ell^2 \log^2 m)$ for $(k, \ell, z)$-clustering under the Fréchet distance, with center curves in $\mathbb{R}^d$ of complexity at most $\ell$.*

Notably, this bound is optimal for constant $\ell$ and $m$, as it recovers the optimal coreset bounds $\Omega(k \varepsilon^{-2-z})$ for $(k, z)$-clustering with centers being points (Huang et al., 2024; Zhu et al., 2024) whenever $k \gg \varepsilon^{-1}$. Interestingly, coresets for clustering under the Fréchet distance was one of the only settings for which the framework of Cohen-Addad et al. (2021) failed to provide state-of-the-art bounds: our construction therefore provides the missing link and strengthens this framework. An overview and comparison of our result with previous results is given in Table 1 and details on related work can be found in Appendix A.

### 1.3. Our Techniques

**Review of Terminal Embedding Constructions.** Let us first review how normal terminal embeddings are constructed. That is, given a finite set of points $P \subset \mathbb{R}^d$, we wish to preserve distances between any $p \in P$ and any $q \in \mathbb{R}^d$. The image $\mathbb{R}^t$ of the mapping $f$ maps $\mathbb{R}^d$ to $\mathbb{R}^{t-1}$ via a JL-type of embedding (typically a matrix of i.i.d. Gaussians or Rademachers) and adds one ancillary coordinate.

For preserving distances between pairs of input points in $P$, we know that defining $f$ to be the image of the JL-embedding along the first $t - 1$ coordinates suffices, and we can simply set the ancillary coordinate to 0. The embedding of an arbitrary point $q$ can then be computed via semidefinite programming (Mahabadi et al., 2018; Narayanan & Nelson, 2019), typically resulting in the ancillary coordinate being non-zero. An alternative is to embed $q$ as follows: find its nearest neighbor $p$ in the input set and define $f(q) = f(p)$ on the first $t - 1$ coordinates, and set the ancillary coordinate to $\|p - q\|$. However, this simple trick only preserves distances up to a factor of 2 (Elkin et al., 2017). In addition, such embeddings are far from preserving affine lines, as they exhibit strong discontinuities.

Before continuing with the key steps of our analysis, we emphasize one specific novelty of our terminal embeddings:

*Table 1.* Resulting coreset sizes of applying our main result Theorem 1.2 to the $(k, \ell, z)$-clustering problem on polygonal curves of complexity at most $m$ and center curves of complexity at most $\ell$ under $z$-th power of Fréchet distance, and comparison to previous work.

| AUTHORS & VENUE | REFERENCE | CORESET SIZE |
|---|---|---|
| Braverman, Cohen-Addad, Jiang, Krauthgamer, Schwiegelshohn, Toftrup, Wu (FOCS'22) | (Braverman et al., 2022) | $\tilde{O}(k^3 \cdot \varepsilon^{-3} \cdot d \cdot \ell \cdot \log m)$ |
| Cohen-Addad, Saulpic, Schwiegelshohn (FOCS'23) | (Cohen-Addad et al., 2023) | $\tilde{O}(k^2 \cdot \varepsilon^{-3} \cdot d \cdot \ell \cdot \log m)$ |
| Conradi, Kolbe, Psarros, Rohde (SoCG'24) | (Conradi et al., 2024) | $\tilde{O}(k^2 \cdot \varepsilon^{-2} \cdot \log n \cdot d \cdot \ell \cdot \log m)$ |
| Cohen-Addad, Draganov, Russo, Saulpic, Schwiegelshohn (SODA'25) | (Cohen-Addad et al., 2025a) | $\tilde{O}(k \cdot \varepsilon^{-1-z} \cdot d \cdot \ell \cdot \log m)$ |
| Our results (here) | Direct application of Theorem 1.2 | $\tilde{O}(k \cdot \varepsilon^{-5-z} \cdot \ell^2 \cdot \log^2 m)$ |
| | Final result in Theorem 1.3 | $\tilde{O}(k \cdot \varepsilon^{-2-z} \cdot \ell^2 \cdot \log^2 m)$ |

they are *piecewise-linear*. Their construction avoid sophisticated techniques such as solving semidefinite programs or building nearest neighbor data structures as in previous work (Mahabadi et al., 2018; Narayanan & Nelson, 2019; Cherapanamjeri & Nelson, 2024). Our novel constructions thus enhance the simplicity of available algorithms as they merely rely on oblivious random projections and data dependent orthogonal projections to low-dimensional subspaces.

**Terminal Embeddings for the Fréchet Distance.** We actually aim for a stronger guarantee than just preserving the Fréchet distance. Observe that a polygonal curve of complexity $\ell$ is a sequence of $\ell$ line-segments, which is a subset of a collection of $\ell$ affine lines. Our goal is to preserve the distance between an arbitrary point on any input line-segment – belonging to one of the curves in $P$ – to any possible point in any affine line that contains a line-segment of a query curve. Clearly, aiming for this stronger guarantee implies that the Fréchet distance is also preserved, since the Fréchet distance relies only on a subset of the point pairs whose distances are preserved.

When attempting to extend the previous approaches for terminal embeddings to lines or line-segments, we can rely on the property that projecting with Gaussian matrices also preserves distances to subspaces. More specifically, given an arbitrary but fixed 1-dimensional affine query subspace $V$ (i.e. an affine line) it is immediate from Lemma 2.1 (with $j = O(1)$) that if the number of columns of a Gaussian matrix $S$ is in the order of $O(\varepsilon^{-2} \log(nm/\delta))$, then with probability at least $1 - \delta$, it holds for any point on a line of the input $p \in P$ and any point on the query line $q \in V$ that

$$\|S(p - q)\| = (1 \pm \varepsilon)\|p - q\|. \qquad (2)$$

The immediate idea of extending this guarantee by enumerating all possible 1-dimensional affine subspaces of $\mathbb{R}^d$, and setting $\delta$ sufficiently small to apply a union bound over all of them clearly cannot work. Similarly to the case of points, as long as the rank of $S$ is less than $d$, there exists an affine line $V$ for which Equation (2) does not hold.

Let us characterize the boundary cases when Equation (2) holds and when it might not hold. Denote by $\mathcal{V}_P$ the set of affine lines spanned by point pairs in $P$, where we represent any line $L$ as an orthonormal basis matrix $V = [v_0, v_1] \in \mathbb{R}^{d \times 2}$, so that $L$ is included in the span of $v_0, v_1$. Note that many lines could be represented by the same matrix, but preserving distances to any point in the columnspan of $V$ is stronger than merely to $L$, and linear subspaces are analytically more tractable than affine lines. While we cannot extend the guarantee of Equation (2) to all lines in $\mathbb{R}^d$, we can do a union-bound over all lines in $\mathcal{V}_P$: hence, we know that all distances between points in $P$ and any point on a line $V \in \mathcal{V}_P$ can be preserved with high probability.

Now, intuitively, it seems that this should also be true for any query $V'$ that is *sufficiently close* to some line $V \in \mathcal{V}_P$. Specifically, if there exists $V \in \mathcal{V}_P$ such that for all $p \in P$

$$\|V'^T (I - VV^T)p\| \\ \leq \varepsilon \cdot \|(I - VV^T)p\| \cdot \|(I - V'V'^T)p\|, \qquad (3)$$

then we can show that the Gaussian matrix also preserves distances between points $p \in P$ on input lines and points $q \in V'$ on the query line within the desired precision. More precisely, $\|(I - V'V'^T)p\|$ is the minimum distance of $p$ to any point of the subspace spanned by the line $V'$ and Equation (3) ensures that there exists no vector $v'$ on the line $V'$ for which $v'^T p$ is significantly different from $v^T p$; in other words, $V$ and $V'$ are approximately aligned.

When Equation (3) does not hold, suppose in the other extreme, that the line $V'$ is orthogonal to any line $V \in \mathcal{V}_P$, and thus orthogonal to the entire span of line-segments in $P$. This situation can simply be handled using two ancillary coordinates: using the Pythagorean theorem, it holds that $\|p - q\|^2 = \|p\|^2 + \|q\|^2$ for any $p \in P$ and $q$ on the line $V'$. We can thus simply embed the entire line $V'$ along the ancillary coordinates. Since we may assume that $\|Sp\|^2 = (1 \pm \varepsilon)\|p\|^2$ for all $p \in P$ and the embedding of the line $V'$ remains orthogonal to $P$, this immediately implies that $\|p - q\|^2$ is preserved up to a factor $(1 \pm \varepsilon)$.

Unfortunately, there is a significant gap between the two extreme cases sketched above, where a candidate line $V'$ is either nearly identical to some line $V \in \mathcal{V}_P$ or nearly orthogonal to all lines in $\mathcal{V}_P$. To overcome this gap, the main idea is to cover $V'$ with a small union of lines in $\mathcal{V}_P$ – in the sense that $V'$ is close to the span of this union – such that if $V'$ is not included in the cover up to small distortion, then it must be nearly orthogonal to all $V \in \mathcal{V}_P$.

Our covering has some relationship with $\varepsilon$-nets for high-dimensional Euclidean balls, but it has a much stronger guarantee: in case of an $\varepsilon$-net, we would only obtain $\|V'^T(I - VV^T)p\| \leq \varepsilon \cdot \|(I - VV^T)p\|$ which would only yield $\|p - q\| = (1 \pm \varepsilon)\|f(p) - f(q)\| + \varepsilon \cdot \|p\|$, i.e., an additive error approximation, while our covering yields the desired multiplicative error approximation guarantee

$$\|f(p) - f(q)\| = (1 \pm \varepsilon)\|p - q\|.$$

The size of a cover to enforce Equation (3) for a single line is only $O(\varepsilon^{-2})$, but to extend our argument to the entire query, we need to construct a cover for each of $\ell$ lines in the query curve $Q \in \mathcal{Q}_\ell$. We then use the Gaussian matrix to preserve the distance between any $p \in P$ and any point $q'$ in the span of the cover. The remaining parts of $V' \in Q$ that are (nearly) orthogonal to the cover are then embedded along $O(\ell)$ ancillary coordinates.

**Applications.** An application of our terminal embedding is a new coreset construction for $(k, \ell, z)$-clustering of polygonal curves under the Fréchet distance. As mentioned earlier, at this stage in coreset research, we have a variety of frameworks that only leave the existence of small enumerations over candidate clusterings as a problem specific open question. Their mere existence is enough to guarantee the success of coreset algorithms. For describing our enumeration, let us first focus on a simple case: if the vertices of a candidate center curve are all close to a vertex in the union of input curves, then we can approximate the placement of the center curve's vertices via standard discretization arguments for covering Euclidean balls.

The more challenging task is to extend this construction to the case where we have a very long line-segment as part of a curve. Then a center curve may place a vertex somewhere along this line-segment, and this vertex may determine the Fréchet distance. This issue posed a problem with previous attempts at obtaining good dimension reduction methods for the Fréchet distance, see for example (Meintrup et al., 2019). The easiest way to address this, is to discretize the long line-segments themselves and derive a coreset construction depending on bit complexity.

An alternative, more commonly used in coreset construction, is based on using combinatorial measures such as the VC-dimension to enumerate over the set of all curves. This enumeration reduces the number of clusterings to roughly

$\exp(k \log k \cdot d \cdot \ell \cdot \log m)$, see (Driemel et al., 2021; Brüning & Driemel, 2023; Cheng & Huang, 2024). Even with the best available analysis of a VC-dimension driven approach (Cohen-Addad et al., 2025a), the authors could not do better than to replace $d$ with our new target dimension, which fails to achieve a better than $\tilde{O}(k\varepsilon^{-5-z}\ell^2 \log^2 m)$ coreset size.

We address the challenge of enumeration by reducing the problem up to constants to the case where long line-segments of the center curve are parallel to some line-segment of the input. This enables an explicit enumeration in which we can parameterize our dimension reduction gradually. As a result, we obtain a sequence of increasingly finer enumerations. We can now perform a chaining analysis using this sequence, leading to tighter bounds on the metric entropy, akin to constructions that yield optimal coresets for $k$-median and $k$-means (Bansal et al., 2024; Cohen-Addad et al., 2022b; Huang et al., 2024; Zhu et al., 2024). Integrating our terminal embeddings for the Fréchet distance into this construction finally leads to a coreset size bound of $\tilde{O}(k\varepsilon^{-2-z}\ell^2 \log^2 m)$, that is optimal assuming $\ell$ and $m$ are small, as the dependence on $k$ and $\varepsilon$ matches recent $\Omega(k\varepsilon^{-2-z})$ lower bounds (Huang et al., 2024; Zhu et al., 2024) that hold whenever $k \gg \varepsilon^{-1}$.

## 2. Preliminaries

For any non-negative numbers $a$ and $b$, we use $a = (1 \pm \varepsilon) b$ to denote $(1 - \varepsilon) b \leq a \leq (1 + \varepsilon) b$. Given a set of points $P$, the linear subspace $Q$ spanned by $P$ is the set of all points that can be written as linear combinations of points in $P$. We say that a matrix $U$ is an orthonormal basis of $Q$ if each column of $U$ has unit Euclidean norm, the columns of $U$ are pairwise orthogonal, and the columns of $U$ span $Q$. For any vector $x \in \mathbb{R}^d$, we denote its Euclidean norm by $\|x\| = (\sum_{i=1}^d x_i^2)^{1/2}$ and the Euclidean distance between $x, y \in \mathbb{R}^d$ by $\|x - y\|$. For any matrix $X \in \mathbb{R}^{n \times d}$ we denote its spectral norm by $\|X\|_2 = \max_{y \in \mathbb{R}^d, y \neq 0} \|Xy\|/\|y\|$. We will use the notion of a subspace preserving sketch given in the following Lemma.

**Lemma 2.1** (Subspace preserving sketch (Woodruff, 2014)). *Let $\varepsilon, \delta > 0$ and let $c$ be some absolute constant. There exists a distribution $\mathcal{D}$ over random $r \times d$ matrices such that if $S \sim \mathcal{D}$ and $r \geq c \cdot \varepsilon^{-2} \cdot (j + \log(1/\delta))$, then for any fixed matrix $U \in \mathbb{R}^{d \times j}$ with orthonormal columns it holds with probability at least $1 - \delta$ that $S$ is a subspace preserving sketch for $U$. I.e., it holds that $\|U^T U - U^T S^T S U\|_2 \leq \varepsilon$.*

We remark that the property of a subspace preserving sketch dates back to Sarlós (2006), and $\mathcal{D}$ is often chosen to be a distribution over matrices with i.i.d. Gaussian or Rademacher entries, rescaled by a factor $1/\sqrt{r}$, (cf. Clarkson & Woodruff, 2009; Woodruff, 2014). We note that sparser projections also exist that can be computed faster at

the cost of a negligibly worse target dimension. We refer the interested reader to appropriate references (Kane & Nelson, 2014; Cohen, 2016; Chenakkod et al., 2024). However, not all these variants yield advantages for the parameterizations considered in our paper; see Section 3 for more discussion.

## 3. Terminal Embeddings for Polygonal Curves

We are given a set of $n$ polygonal curves of complexity at most $m$ lying in $d$-dimensional Euclidean space. We will denote by $P$ the set of input curves interpreted as the collection of all points lying on any of these input curves. We use $\mathcal{Q}_\ell$ to denote the set of all polygonal curves of complexity at most $\ell$. For the sake of presentation, we split into a mapping $f$ acting on the input curves and $g$ acting on the query curves. Both map $\mathbb{R}^d \to \mathbb{R}^t$, so they can be combined to comply with Definition 1.1. We wish to construct a terminal embedding of the curves in $P$ and query curves $\mathcal{Q}_\ell$ that preserves the Fréchet distance with target dimension $t$. That is, we seek embeddings $f, g \colon \mathbb{R}^d \to \mathbb{R}^t$ such that for any curve $\sigma \in P$ and any query $\tau \in \mathcal{Q}_\ell$,

- $f$ maps $\sigma$ to a curve of the same complexity $m$ and $g$ maps $\tau$ to a curve of the same complexity $\ell$, and
- $d_F(f(\sigma), g(\tau)) = (1 \pm \varepsilon) \, d_F(\sigma, \tau)$.

We restate our main theorem and prove it below.

**Theorem 1.2.** *There exist $(\varepsilon, m, \ell)$-preserving terminal embeddings for the Fréchet distance with target dimension $t \in O(\ell \varepsilon^{-4} \log(nm))$.*

In fact, we will show a slightly more general result. We develop a dimension reduction that preserves distances from any point contained in line-segments of the input curves, to any point contained in the affine 1-dimensional subspaces, each induced by a line in a set of queries (possibly, though not necessarily line-segments of a curve). To this end, for any line-segment in the query starting at point $q_1$ and ending at point $q_2$, we let $q$ be the affine 1-dimensional subspace that contains $q_1$ and $q_2$. An entire set of $\ell$ affine 1-dimensional subspaces in the query will be denoted $Q_\ell$.

Specifically, we wish to find mappings $f, g$ that act linearly on $P, Q_\ell$, and show that for any point $a$ contained in any line-segment of any curve in $P$ and any point $b$ contained in any affine 1-dimensional subspace $q \in Q_\ell$, we have

$$\|f(a) - g(b)\|^2 = (1 \pm \varepsilon) \cdot \|a - b\|^2.$$

Notice that this immediately implies that

$$\|f(a) - g(b)\|^z = (1 \pm \varepsilon) \cdot \|a - b\|^z,$$

when we rescale $\varepsilon$ by a factor $O(z)$; we thus focus on $z = 2$.

Because $f$ is linear on $P$ and $g$ on $Q_\ell$, they preserve the complexity of curves: the image of a query curve $Q_\ell$ is a polygonal curve with complexity (at most) $\ell$, and the image of an input curve in $P$ has complexity (at most) $m$. Since all possible distances between points on two curves are hereby preserved up to $(1 \pm \varepsilon)$, it preserves in particular any distance between points matched by any reparameterization of the curves. This finally implies that the supremum that determines the continuous Fréchet distance is preserved up to a $(1 \pm \varepsilon)$ factor as well.

We start with the following technical lemma.

**Lemma 3.1.** *Let $V \subseteq \mathbb{R}^d$ be an arbitrary set. For any set $S \subseteq \mathbb{R}^d$, there exists a set $S_V \subseteq V$, with $|S_V| = O(|S| \cdot \varepsilon^{-2})$ and a corresponding matrix $U$ with orthonormal columns that form a basis for the span of $S_V$, such that $\forall v \in V, s \in S \colon |v^T(I - UU^T)s| \leq \varepsilon \cdot \|(I - UU^T)v\| \cdot \|s\|$.*

*Proof.* Without loss of generality, let the vectors $s \in S$ have unit Euclidean norm. The claim then follows for arbitrary vectors by scaling. We initialize $S_0 = \emptyset$ and proceed in rounds. For a set $S_i$, we define a matrix $U_i$ with orthonormal columns that form a basis for the span of $S_i$. If in round $i$, there exists a vector $v_i \in V$ such that

$$|v_i^T(I - U_{i-1}U_{i-1}^T)s| > \varepsilon \cdot \|(I - U_{i-1}U_{i-1}^T)v_i\|,$$

we say that there was a violation for $s$ in round $i$, and add the point $u_i / \|u_i\|$ to $U_{i-1}$, where $u_i = (I - U_{i-1}U_{i-1}^T)v_i$. Suppose we have $r$ rounds of violations for $s$. Note that $u_i$ is orthogonal to the span of all previously added vectors. Thus, due to the Pythagorean theorem, we have

$$
\begin{aligned}
1 = \|s\|^2 &\geq \|U_r s\|^2 \\
&\geq \sum_{\substack{\text{violation in} \\ \text{round } i \in \{1 \dots r\}}} \left( \frac{|v_i^T(I - U_{i-1}U_{i-1}^T)s|}{\|(I - U_{i-1}U_{i-1}^T)v_i\|} \right)^2 > r \cdot \varepsilon^2.
\end{aligned}
$$

Thus, we have that $r \leq \varepsilon^{-2}$ for any single $s \in S$. Overall, this implies that after at most $|S| \cdot \varepsilon^{-2}$ many rounds of violations, the claim of our lemma must hold. $\square$

Our goal will be to find a low-dimensional subspace and the associated projection matrix $\Pi$ that projects points onto this subspace with the properties given in the following lemma.

**Lemma 3.2.** *Let $P$ be a set of $n$ curves of complexity at most $m$. For any fixed query $Q_\ell$ of at most $\ell$ affine 1-dimensional subspaces, there exists a projection matrix $\Pi$ onto the span of at most $O(\ell \varepsilon^{-2})$ line-segments of curves in $P$ such that for all $a \in P$ and any $b \in q$ for $q \in Q_\ell$, it holds that $\|a - b\|^2 = (1 \pm \varepsilon) \left( \|\Pi(a - b)\|^2 + \|(I - \Pi)a\|^2 + \|(I - \Pi)b\|^2 \right)$*

This key lemma is proven in Appendix B. Some technical details may differ slightly from the high-level idea discussed in Section 1.3. The following result leans itself immediately.

**Corollary 3.3.** *Let $P$ be a set of $n$ curves of complexity at most $m$. Then there exists a collection $\mathcal{C}$ of at most $|\mathcal{C}| = \exp\big(O(\ell\varepsilon^{-2}\log(nm))\big)$ projection matrices with the following properties.*

1. *Each projection matrix $\Pi \in \mathcal{C}$ maps to a subspace that lies in the span of at most $O(\ell\varepsilon^{-2})$ line-segments of $P$.*
2. *For any set $Q_\ell$ of at most $\ell$ affine $1$-dimensional subspaces, there exists $\Pi \in \mathcal{C}$ satisfying the properties asserted by Lemma 3.2.*

*Proof.* There are $\binom{n \cdot m}{\ell \cdot \varepsilon^{-2}}$ many choices of $\Pi$, as we add both offset and direction of the affine line containing a point added in any repetition resp. iteration of Lemma 3.1. Therefore, there are $(n \cdot m) \cdot \binom{n \cdot m}{\ell \cdot \varepsilon^{-2}} \le \exp\big(O(\ell \cdot \varepsilon^{-2} \cdot \log(nm))\big)$ possible subspaces overall. $\qquad\square$

We are finally ready to prove our main result:

*Proof of Theorem 1.2.* The embedding $f$ acts only on the input $P$. It consists of a subspace preserving random projection matrix $S$ (see Lemma 2.1) with a target dimension $r$ that we will specify later. We denote these $r$ coordinates by $C_1$. For the mapping $g$ that acts on the query curves $Q_\ell$, we have $2\ell$ ancillary coordinates denoted by $C_2$. The total embedding target dimension is thus $t = r + 2\ell$.

Now, we describe how to accomplish the embedding $f \colon P \to \mathbb{R}^t$. Any input point $a \in \mathbb{R}^d$, that is, any point contained in one of the input line-segments of curves in $P$, is mapped by $f$ to $Sa$ for the $r$ coordinates in $C_1$ and to $0$ for all ancillary coordinates in $C_2$.

For the embedding $g \colon \mathbb{R}^d \to \mathbb{R}^t$ the construction is more intricate. For any fixed curve $Q_\ell \in \mathcal{Q}_\ell$, Item 1 of Corollary 3.3 yields the existence of a subspace $\Pi \in \mathcal{C}$ spanned by at most $O(\ell \cdot \varepsilon^{-2})$ points. Then $g$ maps the points $b \in Q_\ell$ lying on line-segments of $Q_\ell$ via $S\Pi b$ for the coordinates in $C_1$. In addition, we map $(I - \Pi)b$ to the coordinates in $C_2$.

To see that both mappings are (piece-wise) linear, note that $f$ is merely a random linear projection of the input curves in $P$. For the query curve $Q_\ell$, and any $b \in Q_\ell$, the mapping $g$ consists of two projections $S\Pi b$ and $(I - \Pi)b$ that are both linear mappings to the coordinates $C_1$ and $C_2$, respectively.

**Error Guarantee.** Assuming that $S$ is a subspace preserving sketch (see Lemma 2.1) simultaneously for all $\Pi \in \mathcal{C}$ and for every line-segment in $P$ containing a point $a$, then

$$\|S(a - \Pi b)\|^2 = (1 \pm \varepsilon) \cdot \|a - \Pi b\|^2$$
$$= (1 \pm \varepsilon) \cdot \big(\|(I - \Pi)a\|^2 + \|\Pi(a - b)\|^2\big). \quad (4)$$

By Item 2 of Corollary 3.3 each $\Pi \in \mathcal{C}$ satisfies the equation asserted in Lemma 3.2. Moreover, $(I - \Pi)b$ is contained in a $2\ell$-dimensional subspace, so our mapping of $(I - \Pi)b$ to

the ancillary coordinates $C_2$ preserves $\|(I - \Pi)b\|^2$ exactly. We thus conclude

$$\|f(a) - g(b)\|^2 = \|S(a - \Pi b)\|^2 + \|(I - \Pi)b\|^2$$
$$\overset{(4)}{=} (1 \pm \varepsilon)\big(\|(I - \Pi)a\|^2 + \|\Pi(a - b)\|^2\big) + \|(I - \Pi)b\|^2$$
$$\overset{(3.2)}{=} (1 \pm \varepsilon)^2 \|a - b\|^2 = (1 \pm 3\varepsilon) \|a - b\|^2.$$

The desired $(1 \pm \varepsilon)$ approximation now follows by rescaling $\varepsilon$ by a factor 3. The conclusion for preserving the Fréchet distance follows because the distances between any points on two curves that can possibly be mapped to each other via reparameterizations, are a subset of the distances preserved up to a factor $(1 \pm \varepsilon)$ by our embedding; the supremum that determines the Fréchet distance is thus also preserved.

**Target Dimension.** Our arguments require a subspace preserving sketch (Lemma 2.1) for all segments of the input curves in $P$. There are $O(nm)$ segments, and each of them lies in a 2-dimensional linear subspace. Using Lemma 2.1 and a union bound, we thus need only $O(\varepsilon^{-2}\log(nm/\delta))$ dimensions to embed all of them. Simultaneously, the subspace preserving sketch is required to hold for all possible subspaces of dimension bounded by $O(\ell\varepsilon^{-2})$ defined by $\Pi \in \mathcal{C}$ (see Corollary 3.3), which clearly dominates the embedding dimension. Setting the target dimension to

$$r \ge c \cdot \varepsilon^{-2} \cdot \big(\ell \cdot \varepsilon^{-2} + \log\big(\exp\big(\ell \cdot \varepsilon^{-2} \cdot \log(nm)\big)/\delta\big)\big)$$

for some absolute constant $c$ and using Lemma 2.1 then guarantees that Equation (4) holds for all choices of $\Pi \in \mathcal{C}$ and all choices of $a \in P$ and $b \in Q_\ell \in \mathcal{Q}_\ell$ with probability $1 - \delta$. Combining this with the $2\ell$ coordinates of $C_2$ ultimately yields the desired target dimension $t \in O\big(\ell \cdot \varepsilon^{-4}\log(nm) + \varepsilon^{-2}\log(\delta^{-1})\big)$. $\qquad\square$

**Running Time.** We first emphasize the fact that most applications of terminal embeddings, in particular their application to coresets we focus on below, only require the *existence* of such an embedding and do not need an explicit construction. Nevertheless, it is possible to obtain the embeddings with reasonable efficiency. We state the running time for the specific range of parameters we studied here.

**Proposition 3.4.** *Let $P$ be a set of $n$ polygonal curves of complexity $m$. Consider the mappings $f, g$ realizing the terminal embedding of Theorem 1.2. The mapping $f(a)$ for any point $a \in P$, can be computed in time $O(d\ell\varepsilon^{-4}\log(nm))$. For a given query polygonal curve $Q_\ell$ of complexity $\ell$, the mapping $g$ can be precomputed in time $O(nmd\ell^2\varepsilon^{-6}\log(nm))$. Then $g(b)$ for any $b \in Q_\ell$ can be computed in time $O(d\ell\varepsilon^{-2})$.*

*Proof.* The mapping $f(a)$ can be realized as an oblivious JL-transform $Sa$ in time $O(td) = O(d\ell\varepsilon^{-4}\log(nm))$. To compute the embedding $g$ for a specific query $Q_\ell$ consisting

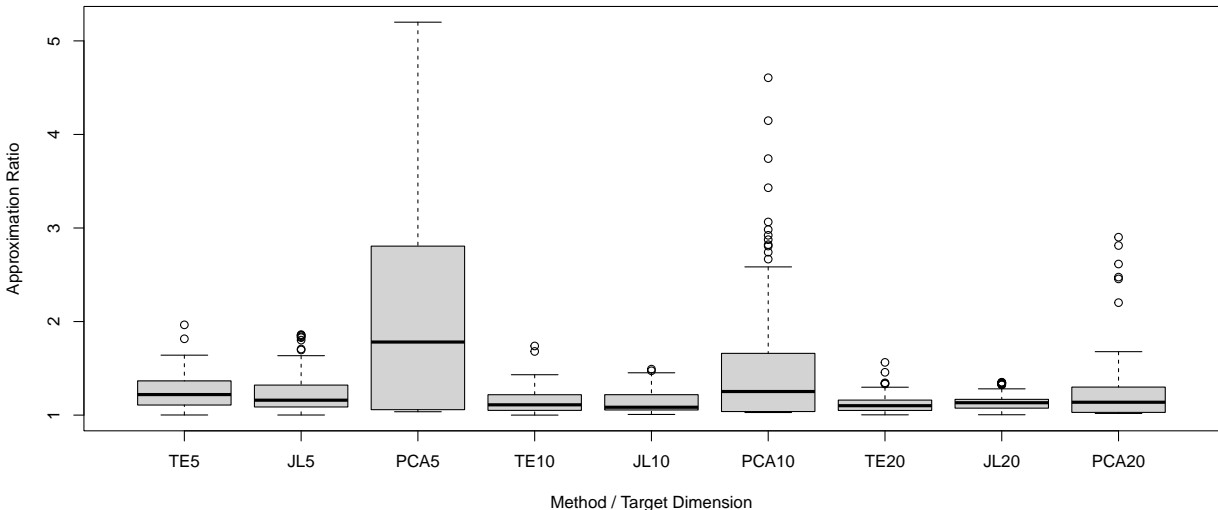

*Figure 1.* Comparison of the approximation ratios (lower is better) of 120 Fréchet distances between time-series, reduced by TE vs. JL and PCA to target dimensions $t \in \{5, 10, 20\}$.

of up to $\ell$ line-segments (or a collection of $\ell$ affine lines), the embedding algorithm needs to construct $\Pi = UU^T$ as asserted by Lemma 3.2 which works by executing the steps of Lemma 3.1 for $nm$ choices of $v$, and $\ell$ choices of $s$. We thus have at most $\ell$ repetitions with $r \leq \varepsilon^{-2}$ iterations each, in which we update $U_i$. In each iteration, we first compute the point $p_i = \mathrm{argmax}_{v_i} \frac{v_i^T (I - U_{i-1} U_{i-1}^T) s}{\|(I - U_{i-1} U_{i-1}^T) v_i\|}$, which takes time $O(nmdi)$, since the rank of $U_{i-1}$ is $i - 1$. Subsequently, we update the basis $U_i$ to the subspace spanned by $U_{i-1}$ and $u_i = \frac{(I - U_{i-1} U_{i-1}^T) p_i}{\|(I - U_{i-1} U_{i-1}^T) p_i\|}$. This takes time $O(di)$, as we merely have to compute $(I - U_{i-1} U_{i-1}^T) p_i$ and normalize. The running time $O(nmd\ell^2 \varepsilon^{-4})$ for the construction of $U$ then follows by summing these terms up over all $\ell\varepsilon^{-2}$ iterations. Now we can precompute $SU$ in time $O(td\ell\varepsilon^{-2}) = O(d\ell^2 \varepsilon^{-6} \log(nm))$ which leads to $O(O(nmd\ell^2 \varepsilon^{-6} \log(nm)))$ overall. Using standard multiplication, computing $S\Pi b = (SU)(U^T b)$ takes time $O(d\ell\varepsilon^{-2} + t\ell\varepsilon^{-2})$, and $(I - \Pi)b = b - U(U^T b)$ takes time $O(d\ell\varepsilon^{-2} + d)$. Since $t < d$, this results in a total running time of $O(d\ell\varepsilon^{-2})$. $\qquad\square$

The proof focuses on dense-JL and standard matrix multiplication for simplicity; using improved JL-constructions or fast matrix multiplication improves the performance straightforwardly. The dependence on $\varepsilon$, for instance, can be improved to obtain a running time of $O(\varepsilon \cdot ndt)$ using sparse JL-transforms (Kane & Nelson, 2014; Høgsgaard et al., 2024). Sparse subspace embeddings can also be applied, if the failure probability is small enough (Cohen, 2016; Chenakkod et al., 2024). We remark that $O(1)$-sparse embeddings (Clarkson & Woodruff, 2013; Meng & Mahoney, 2013; Nelson & Nguyên, 2013) do not apply as they do not

allow a union bound over exponentially many subspaces.

**Application to Coresets for Clustering under the Fréchet Distance.** As discussed before and can be seen in Table 1, a direct application of our Theorem 1.2 to remove the dimension dependence in the best previous coreset construction of Cohen-Addad et al. (2025a) yields $\tilde{O}(k \cdot \varepsilon^{-5-z} \cdot \ell^2 \cdot \log^2 m)$. The proof of Theorem 1.3 improves this to $\tilde{O}(k \cdot \varepsilon^{-2-z} \cdot \ell^2 \cdot \log^2 m)$, optimal in both, $k, \varepsilon$, by integrating our terminal embedding more directly in an improved chaining coreset construction framework, originally due to Cohen-Addad et al. (2021). The extensive details – in particular, how to build the *approximate centroid set* that allows to discretize the set of possible centers even further, required to apply the chaining framework – are entirely in Appendices C and D.

## 4. Experimental Illustration

**Computing Device.** All experiments were conducted on a commodity laptop with Intel(R) Core(TM) i7-8550U CPU, 4 cores at 1.80 GHz, 16 GB DDR4-3200 RAM. Our Python code is available at https://anonymous.4open.science/r/timeseries-7B44.

**Dataset.** We use real-world data that consists of greenhouse gas (GHG) concentrations measured at different sites across California (Lucas et al., 2015). We construct polygonal curves, representing $n = 16$ different GHG concentrations, measured in $m = 327$ regular time periods, each taken simultaneously at $d = 2921$ distinct locations. Additionally, to simulate query curves that are not part of the input, we generated $n = 10$ curves, and a query curve, each of length $m = 5$, in $d = 50$ dimensions. All entries of the generated vertices were Gaussian.

**Methods.** We illustrate the performance of terminal embeddings (TE) as in Theorem 1.2 using the orthogonal projection construction of Lemma 3.1 followed by a Johnson-Lindenstrauss (JL) embedding using matrices with i.i.d. Gaussian $N(0, 1/\sqrt{t})$ entries reducing from the source dimension $d = 50$ to a target dimension $t \in \{5, 10, 20\}$. This is compared to the performance of standard JL for reducing all vertices of curves from the original dimension $d = 2921$ to a target dimension $t \in \{5, 10, 20\}$. As another baseline, we used principal component analysis (PCA), reducing to the same target dimensions. For TE, we provide boxplots of the approximation factors of the Fréchet distances from the query curve to the input curves. For the two baseline methods JL and PCA, we provide boxplots of the approximation factors of the Fréchet distances across all pairs of input time-series.

**Research Question.** We remark that our terminal embeddings are based on subspace embeddings via JL-transforms (Johnson & Lindenstrauss, 1984). JL-transforms are the de-facto standard randomized oblivious dimension reduction technique, while PCA (Pearson, 1901) is the de-facto standard deterministic data-dependent dimension reduction technique not only for high-dimensional time-series data (cf. Paparrizos et al., 2024).

We thus address the following question:

*How do the extended dimension reduction abilities of terminal embeddings (TE) perform in comparison to standard Johnson-Lindenstrauss (JL) transforms and in comparison to PCA for preserving the Fréchet distance between time-series?*

**Results.** The results are displayed in Figure 1. For each target dimension, we compare the resulting approximation ratios for TE against plain JL and PCA. It can be seen that all three become better as the target dimension increases. TE and JL perform consistently better than PCA across all dimensions, in particular TE and JL perform very well already at much lower target dimensions than PCA.

TE perform very similarly, though slightly worse than plain JL at the same target dimension. In this context it must be noted that in contrast to plain JL and PCA, only TE can preserve the Fréchet distance to arbitrary curves in ambient space. This comes at the price of the additional approximation component induced by the orthogonal projection which results in the slightly larger approximation ratios observed in Figure 1 in comparison to plain JL. As shown in our proof of Theorem 1.2, the errors of the orthogonal projection and the subsequent JL mapping accumulate only by $(1 + \varepsilon)^2 \leq (1 + 3\varepsilon)$. The observed additional error (over plain JL) is thus bounded by a small constant and can be compensated by increasing the target dimension according to a constant rescaling of the approximation parameter $\varepsilon$.

## Acknowledgements

C.S. is supported by a Google Research Award. A.M. is supported by the German Research Foundation (DFG) – grant MU 4662/2-1 (535889065) and by the TU Dortmund – Center for Data Science and Simulation (DoDaS).

## Impact Statement

This paper presents work whose goal is to advance the field of Machine Learning. There are many potential societal consequences of our work, none which we feel must be specifically highlighted here.

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

## A. Further Related Work

**Terminal Embeddings.**   Terminal embeddings were introduced by Elkin, Filtser, Neiman (TCS'17) (Elkin et al., 2017). For preserving all Euclidean distances of points in given finite point set $P$ of size $n$, the original reference gave only a constant factor distortion guarantee using an embedding into $O(\log n)$ dimensions. This was soon extended by Mahabadi, Makarychev, Makarychev, Razenshteyn (STOC'18) (Mahabadi et al., 2018) to achieve a strict generalization of the Johnson-Lindenstrauss embedding with $(1 \pm \varepsilon)$ distortion and a target dimension of $O(\varepsilon^{-4} \log n)$. Finally, Narayanan & Nelson (STOC'19) (Narayanan & Nelson, 2019) proved that $O(\varepsilon^{-2} \log n)$ dimensions suffice, which is remarkable, since this matches the lower bound for the weaker Johnson-Lindenstrauss guarantee of Larsen & Nelson (FOCS'17) (Larsen & Nelson, 2017). However, the running time of the embedding was still dominated by solving a semidefinite program for each query point. This issue was resolved by Cherapanamjeri & Nelson (FOCS'21, TCS'24) (Cherapanamjeri & Nelson, 2024) providing sublinear time embeddings based on approximate nearest neighbor techniques.

**Coresets for $(k, \ell, z)$-Clustering of Polygonal Curves under the Fréchet Distance.**   $(k, \ell, z)$-clustering of polygonal curves under the Fréchet distance was introduced by Driemel et al. in (SODA'16) (Driemel et al., 2016). Although it gave no explicit coreset construction, it already contained similar discretization and approximation approaches used in the clustering theory of metric spaces with bounded doubling dimension. The next step towards coresets was taken by Driemel et al. (SoCG'19) (Driemel et al., 2021) by bounding the VC dimension and thus enabling the sensitivity framework to the problem. This gave rise to first but large coreset constructions and centroid sets by Buchin & Rohde (Buchin & Rohde, 2022), which were significantly improved recently by Braverman et al. (FOCS'22) (Braverman et al., 2022) $\tilde{O}(k^3 \cdot \varepsilon^{-3} \cdot d \cdot \ell \cdot \log m)$, Conradi et al. (SoCG'24) (Conradi et al., 2024) $\tilde{O}(k^2 \cdot \varepsilon^{-2} \cdot \log n \cdot d \cdot \ell \cdot \log m)$, and Cohen-Addad et al. (SODA'25) (Cohen-Addad et al., 2025a) $\tilde{O}(k \cdot \varepsilon^{-1-z} \cdot d \cdot \ell \cdot \log m)$ using more and more refined techniques. See also Table 1. For algorithms to compute a solution to $(k, \ell)$-clustering, Buchin, Driemel & Rohde (SODA'21) (Buchin et al., 2023) presented a bi-criteria approximation scheme, and Cheng & Huang (SODA'23) (Cheng & Huang, 2023) improved it to find a proper approximation scheme. They use curve simplification techniques that seem orthogonal to ours.

**Dimension Reduction for Polygonal Curves.**   The first contribution of random projections for preserving Fréchet distance was given by Driemel & Krivošija (Driemel & Krivosija, 2018), followed by a Johnson-Lindenstrauss equivalent for preserving the pairwise distances of input curves, by Meintrup et al. (NeurIPS'19) (Meintrup et al., 2019) within $O(\varepsilon^{-2} \log(nm))$ dimensions. Albeit introducing additive errors, this enabled efficient algorithms for $k$-median, where centers are restricted to the input curves. Recently, Psarros et al. (SoCG'24,DCG'25) (Psarros & Rohde, 2023; 2025) refined to multiplicative errors for the pairwise distances within the same target dimension applying Johnson-Lindenstrauss embeddings to a sophisticated 'skeleton' net construction. We note that the problem can be solved much simpler by embedding low dimensional subspaces containing line segments. However, the method is restricted to discrete clustering objectives, and estimating the clustering cost, as described in (Psarros & Rohde, 2023; 2025). To our knowledge, no terminal embeddings exist for the problem, and these are the missing key methodology to remove dimension dependence in continuous clustering applications and coreset constructions.

## B. Omitted Proofs of the Main Part

**Lemma 3.2.** *Let $P$ be a set of $n$ curves of complexity at most $m$. For any fixed query $Q_\ell$ of at most $\ell$ affine 1-dimensional subspaces, there exists a projection matrix $\Pi$ onto the span of at most $O(\ell \varepsilon^{-2})$ line-segments of curves in $P$ such that for all $a \in P$ and any $b \in q$ for $q \in Q_\ell$, it holds that $\|a - b\|^2 = (1 \pm \varepsilon) \left( \|\Pi(a - b)\|^2 + \|(I - \Pi)a\|^2 + \|(I - \Pi)b\|^2 \right)$*

*Proof of Lemma 3.2.*   Consider an arbitrary $q \in Q_\ell$ and fix an arbitrary $a_0 \in \operatorname{argmin}_{a \in P: \, b \in q, q \in Q_\ell} \|a - b\|$, that is, $a_0$ is any point in $P$ that is closest to $q$. Denote by $b_0$ the projection of $a_0$ onto $q$, that is $b_0 \in \operatorname{argmin}_{b \in q} \|a_0 - b\|$, ties are again resolved arbitrarily. Consider arbitrary points $a \in P$ and $b \in Q_\ell$. For any orthogonal projection matrix $\Pi$, we can write

$$\|a - b\|^2 = \|\Pi(a - b)\|^2 + \|(I - \Pi)a\|^2 + \|(I - \Pi)b\|^2 - 2a^T(I - \Pi)b. \tag{5}$$

For any point $b \in q$, we may write $b = b_0 + \alpha \cdot v = a_0 + (b_0 - a_0) + \alpha \cdot v$, where $v$ is a unit vector and $\alpha$ is a scaling factor. Plugging this into Equation (5), we have

$$\begin{aligned} \|a - b\|^2 = \, &\|\Pi(a - (b_0 + \alpha \cdot v))\|^2 + \|(I - \Pi)(a - a_0)\|^2 \\ &+ \|(I - \Pi)(b_0 - a_0 + \alpha \cdot v)\|^2 - 2(a - a_0)^T(I - \Pi)(b_0 - a_0 + \alpha \cdot v). \end{aligned}$$

If we always include the line-segment containing $a_0$ to the set of points that define the span of $\Pi$, then $(I - \Pi)a_0 = 0$ and the above term simplifies again to

$$\|a - b\|^2 = \|\Pi(a - b)\|^2 + \|(I - \Pi)a\|^2 + \|(I - \Pi)b\|^2 - 2(a - a_0)^T(I - \Pi)(b_0 - a_0 + \alpha \cdot v).$$

Our goal thus reduces to select $\Pi$ s.t.

$$|2(a - a_0)^T(I - \Pi)(b_0 - a_0 + \alpha \cdot v)| \leq \varepsilon \cdot \|a - b\|^2,$$

which implies the equation in Lemma 3.2 as desired. Specifically, by the triangle inequality and a suitable constant rescaling of $\varepsilon$, it suffices to show

$$|2(a - a_0)^T(I - \Pi)(b_0 - a_0)| \leq \varepsilon \cdot \|(I - \Pi)(a - a_0)\| \cdot \|a - b\| \tag{6}$$

and

$$|2(a - a_0)^T(I - \Pi)\alpha \cdot v| \leq \varepsilon \cdot \|(I - \Pi)(a - a_0)\| \cdot \|a - b\|. \tag{7}$$

We first argue Equation (6). Define $s = b_0 - a_0$. Notice that $\|s\| = \|a_0 - b_0\| \leq \|a - b\|$. Note that for $Q_\ell$ there exist at most $\ell$ such vectors. Therefore, invoking Lemma 3.1, there exists a $U$ with $|U| = O(\ell \cdot \varepsilon^{-2})$ such that

$$|(a - a_0)^T(I - UU^T)s| \leq \varepsilon \cdot \|(I - UU^T)(a - a_0)\| \cdot \|s\| \leq \varepsilon \cdot \|(I - UU^T)(a - a_0)\| \cdot \|a - b\| \tag{8}$$

For Equation (7), we require several additional arguments. First recall that $b = b_0 + \alpha \cdot v$. Notice that

$$\|a - (a_0 + \alpha \cdot v)\| \leq \|a - (b_0 + \alpha \cdot v)\| + \|a_0 - b_0\| = \|a - b\| + \|a_0 - b_0\| \leq 2\|a - b\|. \tag{9}$$

Next, due to the Pythagorean theorem, we have the following lower bound

$$\|a - (a_0 + \alpha \cdot v)\|^2 = \|(I - vv^T)(a - a_0)\|^2 + \|v(v^T(a - a_0) - \alpha)\|^2$$
$$\geq \max(\|(I - vv^T)(a - a_0)\|, \|v(v^T(a - a_0) - \alpha)\|)^2,$$

which together with Equation (9) implies

$$\max\left(\|(I - vv^T)(a - a_0)\|, \|v(\alpha - v^T(a - a_0)\|\right) \leq 2\|a - b\|. \tag{10}$$

Furthermore, we have

$$(a - a_0)^T(I - \Pi)\alpha \cdot v = (a - a_0)^T(I - \Pi)(vv^T(a - a_0) + v(\alpha - v^T(a - a_0))), \tag{11}$$

allowing us to control the left hand side via the two terms on the right hand side by means of Equation (10). For the latter term, we have using Lemma 3.1 and Equation (10)

$$|(a - a_0)^T(I - \Pi)v(\alpha - v^T(a - a_0)|$$
$$\leq \quad \varepsilon \cdot \|(I - \Pi)(a - a_0)\| \cdot \|v(\alpha - v^T(a - a_0)\|$$
$$\leq \quad 2\varepsilon \cdot \|(I - \Pi)(a - a_0)\| \cdot \|a - b\|. \tag{12}$$

For the former term, suppose $\Pi$ is initialized with $u := \frac{u'}{\|u'\|}$ where $u' = \text{argmax}_{(a - a_0) \in P}|(a - a_0)^T v|$.

We then get, using Lemma 3.1, that

$$|(a - a_0)^T(I - \Pi)(I - uu^T)vv^T(a - a_0)|$$
$$\leq \quad \varepsilon \cdot \|(I - \Pi)(a - a_0)\| \cdot \|(I - uu^T)v\| \cdot |v^T(a - a_0)|$$
$$\leq \quad \varepsilon \cdot \|(I - \Pi)(a - a_0)\| \cdot \left\|\left(I - \frac{(a - a_0)(a - a_0)^T}{\|a - a_0\|^2}\right)v\right\| \cdot |v^T(a - a_0)|$$
$$= \quad \varepsilon \cdot \|(I - \Pi)(a - a_0)\| \cdot \|(I - vv^T)(a - a_0)\| \cdot \frac{|v^T(a - a_0)|}{\|a - a_0\|}$$
$$\leq \quad \varepsilon \cdot \|(I - \Pi)(a - a_0)\| \cdot \|(I - vv^T)(a - a_0)\|$$
$$\leq \quad 2\varepsilon \cdot \|(I - \Pi)(a - a_0)\| \cdot \|a - b\|, \tag{13}$$

where the final inequality follows from Equation (10). Combining Equations (11), (12) and (13), we then obtain

$$|2(a - a_0)^T(I - \Pi)v(\alpha - v^T(a - a_0))| \leq 8 \cdot \varepsilon \cdot \|(I - \Pi)(a - a_0)\| \cdot \|a - b\|$$

which by a suitable constant rescaling of $\varepsilon$ implies Equation (7). $\qquad\square$

# C. Coresets for Clustering Polygonal Curves under the Fréchet Distance

In this section, we give our improved coreset construction for the Fréchet distance. Our algorithm has the same running time as all existing coreset constructions for the Fréchet distance. That is, assuming that we are given a constant factor approximate clustering or a bicriteria approximation[1] along with an assignment of input curves to centers of the solution, the algorithm runs in linear time. Conradi et al. (Conradi et al., 2024) showed how to compute a suitable bicriteria approximation in $O(nm^2 \mathrm{poly}(k))$ running time, which, to the best of our knowledge, is the state of the art for our purposes.

Since we aim to prove slightly better bounds than a black-box application of our terminal embedding could provide, we have to introduce and then refine a lot of the standard coreset machinery.

## C.1. Improved Coreset Size Bounds for the (Cohen-Addad et al., 2021) Framework

The coreset framework presented in (Cohen-Addad et al., 2021) provided an algorithm for computing coresets and an enumeration condition which determines the coreset size.

In order to enumerate over all solutions, the coreset framework presented by (Cohen-Addad et al., 2021) uses a type of net they call an approximate centroid set defined as follows.

**Definition C.1** ($(\alpha, k, z)$-approximate centroid set). Let $\mathcal{M} = (\mathcal{X}, d)$ be a metric space, $P$ a set of points, $k, z$ two positive integers, and let $\alpha \in (0, \frac{1}{2})$ be a precision parameter. Consider an (optimal or approximate) solution $\mathcal{A}$, and let $\mathcal{C} \in \mathcal{M}^k$ be a set of (potentially infinite) $k$-clusterings for the $(k, z)$-clustering objective. We say that $\mathcal{N}$ is an $(\alpha, k, z, \mathcal{A})$-approximate centroid set of $P$ if, for every solution $\mathcal{S} \in \mathcal{C}$, there exists another solution $\tilde{\mathcal{S}} \in \mathcal{N}$ such that

$$|\mathrm{cost}(p, \tilde{\mathcal{S}}) - \mathrm{cost}(p, \mathcal{S})| \leq \frac{\alpha}{z \log(z/\alpha)} \cdot (\mathrm{cost}(p, \mathcal{S}) + \mathrm{cost}(p, \mathcal{A})),$$

for all $p \in P$ with $\mathrm{cost}(p, \mathcal{S}) \leq \left(\frac{4z}{\alpha}\right)^z \cdot \mathrm{cost}(p, \mathcal{A})$.

Given a bound $|\mathcal{N}|$ on the size of an $(\alpha, k, z, \mathcal{A})$ approximate centroid set, the analysis of (Cohen-Addad et al., 2021) presents a coreset construction as follows.

**Theorem C.2** (Theorem 1 of (Cohen-Addad et al., 2021)). *Let $\mathcal{M} = (\mathcal{X}, d)$ be a metric space. Assume that for any set of points $P$ there exists an $(\varepsilon, k, z, \mathcal{A})$-approximate centroid set $\mathcal{N}$. Then, there exists an $(\varepsilon, k, z)$-coreset of size*

$$|\Omega| \leq \tilde{O}\left(2^{O(z \log z)} \cdot k \cdot (\varepsilon^{-2} + \varepsilon^{-z}) \cdot \log(|\mathcal{N}|)\right).$$

Unfortunately, this easy-to-use reduction is not always tight and it also turn out to not be tight here. A refinement of the analysis, based on the chaining technique from Gaussian process theory (Talagrand, 1996), yields the following theorem:

**Theorem C.3.** *Let $\mathcal{M} = (\mathcal{X}, d)$ be a metric space, and let $U \geq 0$ be a value inherent to the metric space and let $c > 0$ be an absolute constant. Assume that for any set of points $P$ and for every $h > 0$, the $(2^{-h}, k, z, \mathcal{A})$-approximate centroid set (succinctly denoted as) $\mathcal{N}_h$ satisfies*

$$|\mathcal{N}_h| \leq \exp\left(Ukz^2 \log |P| \cdot 2^{ch} \log\left(2^h \varepsilon^{-1}\right)\right).$$

*Then, there exists an $(\varepsilon, k, z)$-coreset of size*

$$|\Omega| \leq \tilde{O}\left(2^{O(z \log z)} \cdot Uk \cdot (\varepsilon^{-c} + \varepsilon^{-2}) \cdot \min(\varepsilon^{-z}, k)\right).$$

Some of the arguments used to prove Theorem C.3 are implicit in previous works, most notably (Cohen-Addad et al., 2022a), but were not given in full generality as expressed above. We provide a self-contained proof in Appendix D. We note that the space of polygonal curves equipped with the Fréchet distance is known to be a metric space. We stress that the above theorem goes beyond bounding coreset sizes for this notion of a *Fréchet metric*; it is, in fact, general and may be convenient for other researchers attempting to prove coreset bounds for other metrics of interest.

---

[1]In this case, an $(\alpha, \beta)$-bicriteria approximation means that the algorithm may use more than $\beta \cdot k$ clusters and that the clustering cost is within a factor of $\alpha$ of the optimal $k$-clustering

The algorithm to build the coresets from Theorem C.2 and Theorem C.3 is the same, from (Cohen-Addad et al., 2021). A brief description of this algorithm is as follows. First, compute a bi-criteria approximate solution (with cost whitin a constant factor of the optimal cost, but using $O(k)$ centers instead of exactly $k$), using (Conradi et al., 2024). Then, partition each cluster of that solution into annuli of exponentially growing radius. Group the radius of the different clusters into "group", where annulus in the same group have the same ratio of (radius of the annulus) / (average distance to the center in the cluster). In each of them, take a uniform sample: the union of those samples forms the desired coreset.

### C.2. Centroid Sets for the Fréchet Distance

The difficulty in applying the above logic to the metric space of curves under the Fréchet distance is that modeling the points of the polygonal curve is no longer sufficient for the entire curve. For example, consider the setting where every curve is a long parallel line of only two vertices. Centers can now be somewhere in the middle of these long lines, making it challenging to enumerate over all potential center sites. Straightforward ways of limiting the number of candidate centers by imposing a discretization along the lines lose parameters depending on the bit complexity of the input. Notably, the VC dimension is our only tool of addressing this challenge, as it is scale-invariant and thus depends only on combinatorial parameters. Unfortunately, the scale-invariance loses a dependence on the ambient dimension $d$, which can be mitigated somewhat with dimension reduction techniques, but will never be optimal in $\varepsilon$.

Despite these challenges, we will prove that we can efficiently enumerate over all potential solutions in a scale sensitive way:

**Lemma C.4.** *Let $P$ be a set of $n$ curves in $T_d^m$ and let $\mathcal{A} \in T_d^{mk}$ be a candidate solution. Then there exists an $(\alpha, k, z, \mathcal{A})$-approximate centroid set $\mathcal{N}$ for $(k, z, \ell)$-clustering on $P$ under the Fréchet distance such that*

$$|\mathcal{N}| \leq \exp\{O\left(\ell(k \log(nm) + kd \log(\ell/\alpha))\right)\}.$$

In the notation of Theorem C.3, the above bound can be written as $|\mathcal{N}| \leq \exp\{Uk \log n \cdot \log \ell\alpha^{-1}\}$, with $U \in O(\ell d \log m)$. Together, Theorem C.2 and Lemma C.4 imply the following:

**Corollary C.5.** *Let $P \in T_d^m$ be a set of input curves to be clustered under the Fréchet distance. Then the $(k, 1, \ell)$-clustering objective admits $(\varepsilon, k, 1)$-coresets of size*

$$|\Omega| \leq \tilde{O}\left(k\varepsilon^{-2} \cdot \ell d \log m\right).$$

**Removing the Dependence on $d$.** This corollary can be easily combined with the linear terminal embeddings, to get coreset of size independent of the dimension $d$. A first, direct corollary follows from Theorem 1.2. Indeed, if $f$ is a terminal embedding, the pre-image of a coreset for $f(P)$ is a coreset for $P$ (Huang & Vishnoi, 2020). Thus, this directly replaces the dependence on $d$ in Corollary C.5 by $\ell\varepsilon^{-4} \log(nm)$.

We can reduce even further the dependence on $\ell$: instead of creating one embedding that works simultaneously for all center curves, we can use Corollary 3.3 to create $\exp(O(\ell\varepsilon^{-2} \log(nm)))$ many different embeddings, such that for each center curve there is one correct embedding. Building nets for each embedding then yields a net for the original space with the benefit of saving an $\varepsilon^{-2}$ factor.

**Theorem 1.3.** *Given a set $P$ of $n$ curves in $\mathbb{R}^d$ of complexity $m$, there exists an $\varepsilon$-coreset of size $\tilde{O}(k\varepsilon^{-2-z}\ell^2 \log^2 m)$ for $(k, \ell, z)$-clustering under the Fréchet distance, with center curves in $\mathbb{R}^d$ of complexity at most $\ell$.*

*Proof.* Corollary 3.3 provides a collection $S$ of $\exp(O(\ell\varepsilon^{-2} \log(nm)))$ projection matrices such that, for any $\sigma \in T_d^\ell$, there is a matrix $\Pi \in S$ such that the equation of Lemma 3.1 hold.

For fixed $\Pi \in S$ and $\sigma \in T_d^\ell$, we can build on the equation of Lemma 3.1 to compute linear mappings $f$ (independent of $\sigma$) and $g$ such that, for any point $a$ in an input line and any $b \in \sigma$, $\|f(a) - g(b)\| = (1 \pm \varepsilon)\|a - b\|$. For this, $f$ and $g$ map $\mathbb{R}^d$ to $\mathbb{R}^{\ell\varepsilon^{-2}+2}$: it is merely enough to add 2 ancillary coordinates to encode $\|(I - \Pi)a\|$ and $\|(I - \Pi)b\|$.

Hence, it is enough to build a net in a space of dimension $\ell\varepsilon^{-2} + 2$. For this, we use Lemma C.4 as a black-box, to get a net of size

$$|\mathcal{N}_\Pi| \leq \exp\{O\left(\ell(k \log(nm) + k\ell\varepsilon^{-2} \log(\ell/\alpha))\right)\}.$$

This net is valid only for the curves in $T_d^\ell$ that respect the equation of Lemma 3.1 with this particular $\Pi$. To get a net for all curves in $T_d^\ell$, it is enough to take the union of nets for all the $\exp(O(\ell\varepsilon^{-2} \log(nm)))$ many $\Pi$, hence providing a net with

size

$$|\mathcal{N}| \leq \exp\big\{O\left(\ell\varepsilon^{-2}\log(nm)\right)\big\} \cdot \exp\big\{O\left(\ell(k\log(nm) + k\ell\varepsilon^{-2}\log(\ell/\alpha))\right)\big\}.$$

Combined with the chaining framework provided in Theorem C.3, this provides the desired coreset size. □

The remainder of this section is dedicated to providing a proof of Lemma C.4. Throughout this section, we use $d_F(a, b)$ to indicate the Fréchet distance.

### C.3. A Warm Up: Parallel Lines

As a warm-up for the Fréchet distance, we will show how to build an approximate centroid set in the particular case where *all input curves are parallel*.

#### C.3.1. SET UP

Assume that there is a single center $\gamma$, and that all input curves satisfy $d_F(p^i, \gamma) \leq \frac{8}{\alpha} d_F(p_i, \mathcal{A})$. Let $\Delta = \min d_F(p^i, \mathcal{A}) = d_F(p^{\min}, \mathcal{A})$, and assume that all input curves are at distance at most $\Delta\alpha^{-2}$ from $\gamma$. We will see later that those assumptions are without loss of generality.

If all vertices of $\gamma$ were close to input vertices, then a simpler discretization of the Euclidean balls centered input vertices is sufficient. Hence, the issue is when some vertex from $\gamma$ is far from every input vertex. This implies that all input curves have "long" edges, and the center's vertices are in the middle of those long edges. Furthermore, those edges are at distance at most $\Delta\alpha^{-2}$ from the corresponding edge of $p^{\min}$: the angle between those segments is therefore somewhat tiny, and we will assume for now the segments are even parallel. We will show later how it is possible to reduce to that case.

All those assumptions simplify the input: it simply consists of parallel segments of same length. Also, suppose that $p^{\min} = p^1 = \langle p_1^1, p_2^1 \rangle$. Let $e := \frac{p_2^1 - p_1^1}{\|p_2^1 - p_1^1\|}$ be the common direction of the segments $p^i$.

#### C.3.2. CONSTRUCTION OF $\widetilde{\gamma}$ FOR PARALLEL LINES

For a given center curve $\gamma$, our goal will be to construct a curve $\widetilde{\gamma}$ with the same distance to any segment as $\gamma$, but where all vertices are "close" to the vertex $p_1^1$. That way, it will be possible to take a net of the space close to $p_1^1$, so as to bound the number of possible curves $\widetilde{\gamma}$.

For that, we start by breaking the cylinder $\text{Cyl}(p^1, \Delta\alpha^{-1})$ (all points at distance at most $\Delta\alpha^{-1}$ from $p^1$) into pieces of length $3\Delta\alpha^{-1}$. Call each piece a "chunk", an let $A_1, A_2, \ldots$ be the chunks taken in order – from $p_1^1$ to $p_2^1$. Since $\gamma$ has $\ell$ vertices, at most $\ell$ chunks contain one vertex from $\gamma$. Note that since $d_F(p^1, \gamma) \leq \frac{8\Delta}{\alpha}$, all vertices from $\gamma$ are in $\text{Cyl}(p^1, \Delta\alpha^{-1})$.

Our construction of $\widetilde{\gamma}$ removes area $A_i$ if neither $A_{i-1}$ nor $A_i$ contain a vertex from $\gamma$. By "removing", we mean that the vertices from subsequent chunks are shifted by $\frac{3\Delta}{\alpha} \cdot e$. More precisely, the construction is as follows: for any $i$, let $R_i$ be the number of chunks $A_j$ such that $j < i$ and $A_{j-1}, A_j$ do not contain any vertex from $\gamma$. $\widetilde{\gamma}$ is now defined as follows: its $i$-th vertex is $\gamma_i - R_{i'} \cdot \frac{3\Delta}{\alpha} \cdot e$, where $i'$ is the chunk that contains $\gamma_i$. Furthermore, we complete the curve by adding a copy of the last vertex of $\gamma$ at the end of $\widetilde{\gamma}$. The key property of that construction is the following:

**Fact C.1.** *All vertices of $\widetilde{\gamma}$ are at distance at most $9\ell \cdot \Delta\alpha^{-1}$ of $p_1^1$.*

*Proof.* All areas have diameter at most $3\Delta\alpha^{-1}$, and all areas containing vertices of $\widetilde{\gamma}$ are separated by at most 2 empty chunks: therefore, they are all among the first $3\ell$ areas. □

Note that $\widetilde{\gamma}$ may be far from $\gamma$: we can nonetheless show it has same distance to every input curve, which is enough for our purposes. This is done in the following lemma.

**Lemma C.6.** *For any input segment $p^i$ and any continuous bijection $h : [0, 1] \to [0, 1]$ with $\max_{t \in [0,1]} \|p^i(h(t)) - \gamma(t)\| \leq \Delta\alpha^{-1}$, there exists a continuous bijection $\tilde{h} : [0, 1] \to [0, 1]$ such that*

$$\max_{t \in [0,1]} \|p^i(\tilde{h}(t)) - \widetilde{\gamma}(t)\| \leq \max_{t \in [0,1]} \|p^i(h(t)) - \gamma(t)\|.$$

*Similarly, for all $h : [0,1] \rightarrow [0,1]$ with $\max_{t \in [0,1]} \|p^i(h(t)) - \widetilde{\gamma}(t)\| \leq \Delta \alpha^{-1}$, there exists a continuous bijection $\bar{h} : [0,1] \rightarrow [0,1]$ such that*

$$\max_{t \in [0,1]} \|p^i(\bar{h}(t)) - \gamma(t)\| \leq \max_{t \in [0,1]} \|p^i(h(t)) - \widetilde{\gamma}(t)\|.$$

*Proof.* For simplicity, we drop the superscript $i$: $p := p^i$. We also assume that the segment $p$ is parameterized linearly: $p(t) = p(0) + (p(1) - p(0)) \cdot t$.

Let $t_i$ such that $t_i = \gamma^{-1}(\gamma_i)$. We assume, up to reparameterization, that $\widetilde{\gamma}$ verifies $\widetilde{\gamma}(t_i) = \widetilde{\gamma}_i$ and that $\widetilde{\gamma}$ is parameterized linearly between vertices: $\forall t \in [t_i, t_{i+1})$, $\widetilde{\gamma}(t) = \widetilde{\gamma}_i + \frac{\widetilde{\gamma}_{i+1} - \widetilde{\gamma}_i}{t_{i+1} - t_i} \cdot (t - t_i)$ – and we do the same assumptions for $\gamma$. Combined with Thales' theorem, this ensures the following property:

$$\forall t, \|\gamma(t) - p(\Pi_{\gamma(t)})\| = \|\widetilde{\gamma}(t) - p(\Pi_{\widetilde{\gamma}(t)})\|,$$

where $\Pi$ is defined such that $p(\Pi_x)$ is the orthogonal projection of $x$ onto $p$.

In order to define $\tilde{h}$, let us defines zones: a zone is a maximal group of consecutive non-removed chunks, i.e., $A_i$ and $A_j$ are in the same zone if, for any $i' \in [i, j]$, $A_{i'}$ or $A_{i'-1}$ contain a vertex of $\gamma$. All vertices from a zone have the same *shift*, defined as $R_i = R_j$.

Let us now define $\tilde{h}$. For all $t$ such that $t \in [t_i, t_{i+1}]$ and $t_i, t_{i+1}$ are in the same zone with shift $R$, let $\tilde{h}(t) = h(t) - R \cdot \frac{3\Delta}{\alpha}$. Hence, on those intervals, both the curves and their traversal are simply shifted by the same constant, and we get an equality $\|p(\tilde{h}(t)) - \widetilde{\gamma}(t)\| = \|p(h(t)) - \gamma(t)\|$.

For times $t \in (t_i, t_{i+1})$ such that $t_i$ and $t_{i+1}$ are not in the same zone, we need to proceed differently. We consider intervals $(t_i, t_{i+1})$ in increasing order of $i$. In particular, this implies that $\tilde{h}(t_i)$ is defined we constructing $\tilde{h}$ for $t \in (t_i, t_{i+1})$. We would ideally like to set $\tilde{h}(t) = \Pi_{\widetilde{\gamma}(t)}$, so that $\|p(\tilde{h}(t)) - \widetilde{\gamma}(t)\| = \|p(\Pi_{\gamma(t)}) - \gamma(t)\| \leq \|p(h(t)) - \gamma(t)\|$, as the maximum is more than the distance from any point to its projection on $p$. This however may break continuity, as it may be that $\Pi_{\gamma(t_i)} \neq h(t_i)$ or $\Pi_{\gamma(t_{i+1})} \neq h(t_{i+1})$.

We therefore construct $\tilde{h}$ the following way. For all $t$ where $\Pi_{\gamma(t)} \leq h(t_i)$, we let $\tilde{h}(t) = \tilde{h}(t_i)$. This only decreases the distance, compared to $h$: indeed, it holds that $\Pi_{\gamma(t)} \leq h(t_i) \leq t$, so it must be that $\|\gamma(t) - p(h(t_i))\| \leq \|\gamma(t) - p(h(t))\|$. This handles the case where $\Pi_{\gamma(t_i)} < h(t_i)$. For the case $\Pi_{\gamma(t_i)} > h(t_i)$, we proceed similarly: for all $t$ with $h(t) \leq \Pi_{\gamma(t_i)}$, define $\tilde{h}(t) = h(t_i)$. This also ensures that $\|\gamma(t) - p(h(t_i))\| \leq \|\gamma(t) - p(h(t))\|$.

The case $\Pi_{\gamma(t_{i+1})} \neq h(t_{i+1})$ is handled exactly the same way. For all $t$ where $\Pi_{\gamma(t)} \geq h(t_{i+1})$, we let $\tilde{h}(t) = \tilde{h}(t_{i+1})$, and for all $t$ with $h(t) > \Pi_{\gamma(t_{i+1})}$, define $\tilde{h}(t) = h(t_{i+1})$.

In those cases, we get $\|p(\tilde{h}(t)) - \widetilde{\gamma}(t)\| \leq \|p(h(t)) - \gamma(t)\|$. For all other $t$, we let $\tilde{h}(t) = \Pi_{\widetilde{\gamma}(t)}$. By property of orthogonal projection, this ensures that $\|p(\tilde{h}(t)) - \widetilde{\gamma}(t)\| = \|\gamma(t) - p(\Pi_{\gamma(t)})\| \leq \max_{t \in [0,1]} \|p(h(t)) - \gamma(t)\|$, as the maximum is more than the distance from any point to its projection on $p$.

The function $\tilde{h}$ defined that way is non-decreasing and continuous: when $t_i$ and $t_{i+1}$ are in the same zone, this is a direct consequence of the monotonicity and continuity of $h$. When they are not in the same zone, the first property holds because $\Pi_{\gamma(\cdot)}$ is non-increasing on $[t_i, t_{i+1}]$. Indeed, when $\gamma_i$ and $\gamma_{i+1}$ are not in two consecutive areas, it must be that $\Pi_{\gamma_{i+1}} > \Pi_{\gamma_i}$, as otherwise either $\|p(h(t_i)) - \gamma_i\| > \Delta \alpha^{-1}$ or $\|p(h(t_{i+1})) - \gamma_{i+1}\| > \Delta \alpha^{-1}$. Last, continuity holds at $t_i$ and $t_{i+1}$ by our construction, and in between by continuity of $\Pi_{\gamma(t)}$. Hence, we get

$$\max_{t \in [0,1]} \|p^i(\tilde{h}(t)) - \widetilde{\gamma}(t)\| \leq \max_{t \in [0,1]} \|p^i(h(t)) - \gamma(t)\|.$$

The definition of $\bar{h}$ is completely symmetric. □

## C.4. General Case

We now turn to the general case, and show the main lemma C.4. We first describe how the set $\mathcal{N}^C$ is constructed, to then prove it is an approximate centroid set.

C.4.1. CONSTRUCTION OF $\mathcal{N}^C$

To build $\mathcal{N}^C$, we first discretize the potential locations of vertices from the center curves. We first build a set $\mathbb{C}$ consisting of those. For all input curves $p^{min}$ (consisting of a "guess" of the closest input curve to the center curve), we add several sets to $\mathbb{C}$, as follows. Let $\Delta = d_F(p^{min}, \mathcal{A})$.

- First, to handle the case where all vertices from the center curves are close to input vertices, we first add to $\mathbb{C}$ precise nets around each input vertex: for all input curves $p^i$ and every vertex $p^i_j$ on it, let $N_{i,j}$ be an $\alpha\Delta$-net of $B(p^i_j, \ell \cdot 100\Delta\alpha^{-2})$.

- Next, for the case where vertices from the center curves are far away from input vertices, we proceed as follows. Fix a segment $s^{min}$ of $p^{min}$: we first construct a set $NC_{s^{min}}$ of points of $s^{min}$. For any segment $s$ from another input curve. Let $I$ be the projection of $s$ onto $s^{min}$ and let $I_1, ..., I_{100/\alpha^3}$ be $1/\alpha$ points regularly spaced on $I$. Add those points to $NC_{s^{min}}$. Further, let $s'$ be the projection of $s^{min}$ onto $s$, and $I'$ the projection of $s'$ onto $s^{min}$: add to $NC_{s^{min}}$ $100/\alpha^3$ points regularly spaced on $I'$.

  Now, for each point $v \in CN_{s^{min}}$, let $N_{s^{min},v}$ be an $\alpha\Delta$-net of $B(v, 100\ell \cdot \Delta\alpha^{-2})$ ($v$ is the "center" for net $N_{s^{min},v}$, hence the name $NC_{s^{min}}$ for "net centers").

$\mathbb{C}$ is now made the union of all $N_{i,j}$'s and all $N_{s^{min},v}$'s.

The approximate centroid set $\mathcal{N}^C$ consists of all curves consisting of $\ell$ vertices from $\mathbb{C}$. In other words, we restrict the candidate centers to have $\ell$ vertices from $\mathbb{C}$.

C.4.2. PROOF OF LEMMA C.4

We can now show the proof of our main lemma C.4, by showing $\mathcal{N}^C$ has small size and is an approximate centroid set. We first show the size guarantee.

**Lemma C.7.** $\mathcal{N}^C$ *has size* $\left(n^2 m\right)^{k\ell} \cdot \left(\frac{\ell}{\alpha}\right)^{O(dk\ell)}$.

*Proof.* There are $n$ curves $p^{min}$, hence $n^2 m$ many different $N_{i,j}$. Each of them has size $\left(\frac{\ell}{\alpha}\right)^{O(d)}$. Similarly, there are $nm$ many segments $s^{min}$, and for each of them the set $NC_{s^{min}}$ contains at most $2nm/\alpha$ many points. Each net $N_{s^{min},v}$ has size $\left(\frac{\ell}{\alpha}\right)^{O(d)}$. Hence, in total, there are $n^2 m \cdot \left(\frac{\ell}{\alpha}\right)^{O(d)}$ many points in $\mathbb{C}$.

Since the approximate centroid set $\mathcal{N}^C$ is constructed by choosing $k$ curves made of $\ell$ vertices from $\mathbb{C}$ for each candidate solution, it has size at most $\left(n^2 m\right)^{k\ell} \cdot \left(\frac{\ell}{\alpha}\right)^{O(dk\ell)}$. $\qquad\square$

To show that $\mathcal{N}^C$ is an approximate centroid set, we proceed as follows. First, we fix a center curve $\gamma$: we want to construct $\widetilde{\gamma}$ whose vertices are all in $\mathbb{C}$ such that, for any input curve $p$ with $d_F(p, \gamma) \leq \frac{8}{\alpha} d_F(p, \mathcal{A})$, then

$$|d_F(p, \gamma) - d_F(p, \widetilde{\gamma})| \leq \frac{\alpha}{\log(1/\alpha)}(d_F(p, \gamma) + d_F(p, \mathcal{A})).$$

Doing so for all the $k$ curves in a candidate solution $\mathcal{S}$ shows the existence of the solution $\tilde{\mathcal{S}}$ from the approximate centroid set that approximates $\mathcal{S}$.

We assume for simplicity that all input curves satisfy $d_F(p, \gamma) \leq \frac{8}{\alpha} d_F(p, \mathcal{A})$: there is nothing to prove for the others. Next, we let $p^{min}$ be the curve closest to $\mathcal{A}$, and let $\Delta = d_F(p^{min}, \mathcal{A})$.

First, we note that all curves $p$ with $d_F(p, \gamma) \geq \frac{25\Delta}{\alpha^2}$ are dealt with easily.

**Lemma C.8.** *If* $d_F(p^{min}, \gamma) = (1 \pm \alpha) d_F(p^{min}, \widetilde{\gamma}) + \alpha\Delta$, *then for all curves $p$ with $d_F(p, \gamma) \geq \frac{25\Delta}{\alpha^2}$, it holds that*

$$d_F(p, \widetilde{\gamma}) \in (1 \pm \alpha) d_F(p, \gamma).$$

*Proof.* The assumption implies that $d_F(\gamma, \widetilde{\gamma}) \leq d_F(\gamma, p^{min}) + d_F(\widetilde{\gamma}, p^{min}) \leq 3 d_F(\gamma, p^{min}) + \alpha\Delta \leq \frac{25}{\alpha}\Delta$.

Hence, if $\Delta \leq \frac{\alpha^2}{25} d_F(p, \gamma)$, then it directly holds that $d_F(p, \widetilde{\gamma}) \in (1 \pm \alpha) d_F(p, \gamma)$. $\qquad\square$

Thus, from now on, we may assume all input curves satisfy $d_F(p, \gamma) \leq \frac{25\Delta}{\alpha^2}$.

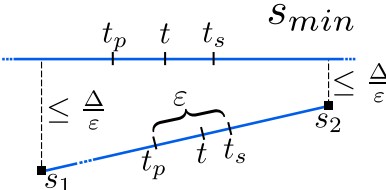

*Figure 2.* Illustration of the natural conditions. $s^{\min}$ is the top segment, $s$ the lower one.

### C.4.3. CONSTRUCTION OF $\widetilde{\gamma}$

We construct $\widetilde{\gamma}$ as follows. For each vertex $\gamma_v$ of $\gamma$, we distinguish two cases.

First, if $\gamma_v$ is in some $B(p_j^i, \ell \cdot 100\Delta\alpha^{-2})$ we simply define $\widetilde{\gamma}_v$ to be the closest point to $\gamma_v$ in $N_{i,j} \subset \mathbb{C}$. Let $\widetilde{\gamma}^1$ be the curve obtained after all those replacements.

**Fact C.2.** $d_F(\gamma, \widetilde{\gamma}^1) \leq \alpha\Delta$.

*Proof.* By construction of $\widetilde{\gamma}^1$, we can map each vertex of $\gamma$ with one of $\widetilde{\gamma}^1$ such that pair of vertices are at distance at most $\alpha\Delta$. It is well-known (see e.g. Section 2 in (Aronov et al., 2006)) that this is an upper-bound on Fréchet distance. $\square$

Now, we deal from the vertices of $\widetilde{\gamma}^1$ that are not in any ball $B(p_j^i, \ell \cdot 100\Delta\alpha^{-2})$, by using the construction of Section C.3.

We start by making some simplifying assumptions and fixing some notations. For any input curve $p^i$ let $h_i$ a bijection such that $d_F(p^i, \widetilde{\gamma}) = \max_{t \in [0,1]} \|\widetilde{\gamma}(t) - p^i(h^i(t))\|$. Up to parameterizing each input curve $p^i$ differently, we can assume $h^i$ to be the identity. This implies the following:

**Fact C.3.** *For any $t$, $\|p^{\min}(t) - p(t)\| \leq \frac{33\Delta}{\alpha^2}$.*

*Proof.* By the parameterization chosen above, it holds that $\|p^{\min}(t) - \gamma(t)\| \leq d_F(p^{\min}, \gamma) \leq \frac{8\Delta}{\alpha}$ and $\|p(t) - \gamma(t)\| \leq d_F(p, \gamma) \leq \frac{25\Delta}{\alpha^2}$. We conclude with triangle inequality. $\square$

Let $\widetilde{\gamma}_v^1$ be a vertex not in any ball $B(p_j^i, \ell \cdot 100\Delta\alpha^{-2})$, and let $t$ be such that $\widetilde{\gamma}^1(t) = \widetilde{\gamma}_v^1 = \gamma_v$. Let $s^{\min}$ be the segment of $p^{\min}$ that is traveled along at time $t$, and let $s_1^{\min}$ and $s_2^{\min}$ be its two endpoints. For $i = 1, 2$, let $t_i^{\min}$ be such that $p^{\min}(t_i^{\min}) = s_i^{\min}$. We parameterize $s^{\min}$ such that $\forall t' \in [t_1^{\min}, t_2^{\min}], s^{\min}(t') = p^{\min}(t')$.

Let $t_p$ (as predecessor) and $t_s$ (as successor) such that $s^{\min}(t_p)$ and $s^{\min}(t_s)$ are the points from $NC_s^{\min}$ that are respectively the last point of $NC_s^{\min}$ before $t$, and the first one after $t$. The idea of our analysis is that, if $\widetilde{\gamma}_v^1$ is far from $s^{\min}(t_p)$ and $s^{\min}(t_s)$ (which corresponds to our assumption on $\widetilde{\gamma}_v^1$), then each input curve can be made parallel to $s^{\min}$, in between time $t_p$ and $t_s$. This allows us to apply Lemma C.6.

For that, we introduce three conditions, that are satisfied by any "natural" input (and we will see afterwards how to deal with the particular cases where they are not fulfilled).

For any input curve $p$ and segment $s$ on $p$ that is traveled along at time $t$, we have:

(i). The projection of $s$ onto $s^{\min}$ intersects $s^{\min}(t_p, t_s)$.

(ii). $s[t_p : t_s]$ is an $\alpha^3/33$-fraction of $s$.

(iii). Let $s_1$ and $s_2$ be the extremities of $s$: it holds that $\text{dist}(s_1, s^{\min}) \leq 33\Delta\alpha^{-2}$ and $\text{dist}(s_2, s^{\min}) \leq 33\Delta\alpha^{-2}$.

Those conditions are illustrated in Figure 2. Our main conceptual idea here is that, when for all such $s$ the three conditions are fulfilled, then it is possible to reduce to the case where all segments are parallel. We show later how to handle the case where they are not fulfilled.

**Lemma C.9.** *Assume that, for all input curves $p^i$ and segment $s^i$ on $p$ that is traveled along at time $t$, the three conditions (i), (ii) and (iii) are fulfilled. Then, there exists curves $\tilde{p}^i$ such that*

$$\forall i, d_F(p^i, \tilde{p}^i) \leq \alpha\Delta,$$

and $\tilde{p}^i$ is parallel to $s^{\min}$ between time $t_p$ and $t_s$.

*Proof.* We focus on a curve $p^i$ that fulfills conditions (i), (ii) and (iii). We drop the index $i$ for simplicity. To build $\tilde{p}$, we simply replace the curve between $p(t_p)$ and $p(t_s)$ by a segment of same length parallel to $s^{\min}$ that starts at $p(t_p)$.

We claim that $\max_{t \in [0,1]} \|p(t) - \tilde{p}(t)\| \le \alpha\Delta$. For this, it is enough to focus on $t \in [t_p, t_s]$: further, by construction, the maximal distance is attained at $t_s$. We show that the $\|p(t_s) - \tilde{p}(t_s)\|$ is at most an $\alpha^3/33$-fraction of the projected distance to $s^{\min}$, which is at most $33\Delta\alpha^{-2}$ by condition (iii): this would conclude.

This turns out to be a direct consequence of Thales' theorem. Indeed, by condition (iii), $s[t_p : t_s]$ is a $\alpha^3/33$ fraction of $s$, and $p(t_s) - \tilde{p}(t_s)$ is orthogonal to $s^{\min}$. Therefore, Thales' theorem shows that $\|p(t_s) - \tilde{p}(t_s)\|$ is an $\alpha^3/33$ fraction of the maximum distance between the points of $s$ projected onto the orthogonal of $s^{\min}$. By condition (iii), this maximum distance is at most $33\Delta\alpha^{-2}$, which concludes the lemma. $\square$

As a direct corollary of this lemma, one can apply Lemma C.6 with curves $\tilde{p}^i$ to build the curve $\widetilde{\gamma}$ between time $t_p$ and $t_s$: by Lemma C.9, this curve $\widetilde{\gamma}$ will at the same distance as $\gamma$ to all $p^i$. We give more details in Section C.4.5.

### C.4.4. DEALING WITH BORDER CASES

We now turn to the cases where the natural conditions are not satisfied. In that case, we can show that the point $\widetilde{\gamma}_v^1$ is actually in some ball $B(p_j^i, \ell \cdot 100\Delta\alpha^{-2})$, which contradicts its definition.

For the following lemmas, we consider for all input curves $p$ the segment $s$ that is traveled along at time $t$ (i.e., $p(t) \in s$).

**Lemma C.10.** *If condition (i) is not satisfied for some segment $s$, then either $\gamma_v \in B\left(p^{min}(t_p), \frac{60\Delta}{\alpha^2}\right)$, or $\gamma_v \in B\left(p^{min}(t_s), \frac{60\Delta}{\alpha^2}\right)$.*

*Proof.* The projection of $s$ onto $s^{\min}$ falls either before or after $s^{\min}(t_p, t_s)$. Up to considering a symmetric input, we can assume it falls before.

For $i = 1, 2$ let $t_i$ such that $s(t_i) = s_i$ (where $s_1$ and $s_2$ are the two extremities of $s$). We define $\tau_1 = \max(t_1, t_p)$ and $\tau_2 = \min(t_2, t_s)$. Note that the time $t$ such that $\gamma(t) = \gamma_v$ is comprised in the interval $[\tau_1, \tau_2]$, as it is both in $[t_1, t_2]$ and $[t_p, t_s]$.

Our first step is to show that, for $i = 1, 2$:

$$\|s^{\min}(t_p) - s(\tau_i)\| \le 34\Delta\alpha^{-2}. \tag{14}$$

If $\tau_1 = t_p$, Equation (14) holds by Observation C.3. In the other case, we use the fact that the projection of $s_1 = s(\tau_1)$ onto $s^{\min}$ is before $s^{\min}(t_p)$ on the segment $s^{\min}$ (which holds by the lemma's assumption), which is itself before $s^{\min}(\tau_1)$ (by definition of $\tau_1$): therefore, $\|s(\tau_1) - s^{\min}(t_p)\| \le \|s(\tau_1) - s^{\min}(\tau_1)\| \le 34\Delta\alpha^{-2}$, where the last inequality follows from Observation C.3. Hence, Equation (14) holds for $i = 1$.

Consider now the case $i = 2$. The projection of $s(\tau_2)$ lies before $s^{\min}(t_1)$, and $\tau_2 \ge t_1$ (since $\tau_2 \ge t \ge t_1$): hence, as previously, $\|s^{\min}(t_1) - s(\tau_2)\| \le \|s^{\min}(\tau_2) - s(\tau_2)\| \le 33\Delta\alpha^{-2}$. This concludes the proof of Equation (14).

Now, we conclude the lemma: since $s$ is a segment, and $t \in [\tau_1, \tau_2]$, it holds that

$$\|s(t) - s^{\min}(t_p)\| \le \max_i \|s^{\min}(t_p) - s(\tau_i)\| \le 33\Delta\alpha^{-2}.$$

Therefore,

$$\|\gamma(t) - s^{\min}(t_p)\| \le \|\gamma(t) - s(t)\| + \|s(t) - s^{\min}(t_p)\| \le 25\Delta\alpha^{-2} + 33\Delta\alpha^{-2}.$$

This concludes the proof. $\square$

**Lemma C.11.** *If there is a segment $s$ such that condition (i) is fulfilled but (ii) is not, then either $\gamma_v \in B\left(p^{min}(t_p), \frac{140\Delta}{\alpha^2}\right)$, or $\gamma_v \in B\left(p^{min}(t_s), \frac{140\Delta}{\alpha^2}\right)$.*

*Proof.* Since condition (i) holds, the projection of $s$ onto $s^{\min}$ intersects $s^{\min}[t_p : t_s]$. By maximality of $t_p$ and minimality of $t_s$, this implies that $s^{\min}[t_p : t_s]$ is included in the projection, and is at most an $\alpha^3/100$-fraction of it (by construction of $t_p$ and $t_s$).

Our first claim is that

$$\|\text{Proj}_{s^{\min}}(s(t_p)) - s^{\min}(t_p)\| \geq \|s^{\min}(t_p) - s^{\min}(t_s)\| \tag{15}$$

$$\text{or } \|\text{Proj}_{s^{\min}}(s(t_s)) - s^{\min}(t_s)\| \geq \|s^{\min}(t_p) - s^{\min}(t_s)\|. \tag{16}$$

If both equation did not hold, then by triangle inequality we would have $\|\text{Proj}_{s^{\min}}(s(t_s)) - \text{Proj}_{s^{\min}}(s(t_p))\| \leq 3\|s^{\min}(t_p) - s^{\min}(t_s)\|$, and therefore it would be a $\alpha^3/33$ fraction of the projection of $s$ onto $s^{\min}$. This would contradict the assumption. Hence, one of Equation (15) and Equation (16) does not hold: assume without loss of generality that Equation (15) does not hold.

We use this equation to bound the following:

$$
\begin{aligned}
\|s(t_p) - s^{\min}(t)\| &\leq \|s(t_p) - s^{\min}(t_p)\| + \|s^{\min}(t_p) - s^{\min}(t)\| \\
&\leq \frac{33\Delta}{\alpha^2} + \|s^{\min}(t_p) - s^{\min}(t_s)\| && \text{(Observation C.3)} \\
&\leq \frac{33\Delta}{\alpha^2} + \|\text{Proj}_{s^{\min}}(s(t_p)) - s^{\min}(t_p)\| && \text{(Equation (15))} \\
&\leq \frac{33\Delta}{\alpha^2} + \|\text{Proj}_{s^{\min}}(s(t_p)) - s(t_p)\| + \|s(t_p) - s^{\min}(t_p)\| \\
&\leq \frac{33\Delta}{\alpha^2} + 2\|s(t_p) - s^{\min}(t_p)\| \\
&\leq \frac{99\Delta}{\alpha^2}.
\end{aligned}
$$

We conclude with the triangle inequality:

$$
\begin{aligned}
\|s^{\min}(t_p) - \gamma(t)\| &\leq \|s^{\min}(t_p) - s(t_p)\| + \|s(t_p) - s^{\min}(t)\| + \|s^{\min}(t) - \gamma(t)\| \\
&\leq \frac{33\Delta}{\alpha^2} + \frac{99\Delta}{\alpha^2} + \frac{8\Delta}{\alpha} \\
&\leq \frac{140\Delta}{\alpha^2}. \qquad\qquad \square
\end{aligned}
$$

**Lemma C.12.** *If there is an input $p$ with segment $s$ such that condition (iii) is not fulfilled, then either $\gamma_v \in B\left(s_1^{\min}, \frac{66\Delta}{\alpha^2}\right)$, or $\gamma_v \in B\left(s_2^{\min}, \frac{66\Delta}{\alpha^2}\right)$, or (1) condition (iii) holds for $s'$, the projection of $s^{\min}$ onto $s$, and (2) $s(t) \in s'$.*

*Proof.* We first start with an observation. For $i = 1, 2$, let $t_i$ such that $s(t_i) = s_i$ (where $s_1$ and $s_2$ are the two extremities of $s$). If $t_i \in [t_1^{\min}, t_2^{\min}]$, then $\text{dist}(s_i, s^{\min}) \leq \|s(t_i) - s^{\min}(t_i)\| \leq \frac{33\Delta}{\alpha^2}$ (from Observation C.3) and condition (iii) is fulfilled. Hence, either $t_1 \notin [t_1^{\min}, t_2^{\min}]$ or $t_2 \notin [t_1^{\min}, t_2^{\min}]$.

Focus on the first case. Our key (simple) observation is that $t_1^{\min} \in [t_1, t_2]$: indeed, $t$ is in both $[t_1, t_2]$ and $[t_1^{\min}, t_2^{\min}]$, so it holds that $t_1 \leq t_1^{\min} \leq t \leq t_2$. Furthermore, $s'_1 = \text{Proj}_s\left(s_1^{\min}\right)$. Hence, $\text{dist}(s_1^{\min}, s') \leq \|s_1^{\min} - s(t_1^{\min})\| \leq \frac{33\Delta}{\alpha^2}$, so condition (1) is satisfied.

If condition (2) is satisfied as well, we are done. Otherwise, we can show that $\gamma_v$ is close to $s_1^{\min}$. Indeed, in that case, it holds that $p(t) = s(t) \in [s(t_1^{\min}), \text{Proj}_s\left(s_1^{\min}\right)]$. Hence,

$$
\begin{aligned}
\|s_1^{\min} - s(t)\| &\leq \|s_1^{\min} - s(t_1^{\min})\| + \|s_1^{\min} - \text{Proj}_s\left(s_1^{\min}\right)\| \\
&\leq 2\|s_1^{\min} - s(t_1^{\min})\| \\
&\leq \frac{66\Delta}{\alpha^2},
\end{aligned}
$$

where the second inequality holds by property of the projection. This concludes. $\qquad \square$

### C.4.5. PUTTING EVERYTHING TOGETHER: $\mathcal{N}^C$ IS AN APPROXIMATE CENTROID SET

We now combine the previous lemmas to show how to build the curve $\widetilde{\gamma}$ with vertices in $\mathcal{N}^C$, and that $|d_F(p^i, \widetilde{\gamma}) - d_F(p^i, \gamma)| \leq \alpha(d_F(p^i, \gamma) + \Delta)$, which concludes that $\mathbb{C}$ is an approximate centroid set.

As explained, start by constructing $\widetilde{\gamma}^1$, the curve obtained by replacing all vertices of $\gamma$ that are in some $B(p^i_j, \ell \cdot 100\Delta\alpha^{-2})$ by their closest point $N_{i,j} \subset \mathbb{C}$. Fact C.2 shows that this transformation preserves all distances from input curves to $\gamma$.

Let $\gamma_v$ be a vertex that is outside of all balls $B(p^i_j, \ell \cdot 100\Delta\alpha^{-2})$, $t$ be such that $\gamma(t) = \gamma_v$, and $s^{\min}$ the segment of $p^{min}$ that is traveled along at time $t$. We define $t_p$ and $t_s$ such that $s^{\min}(t_p)$ and $s^{\min}(t_s)$ are the points from $NC_s^{\min}$ that are respectively the last point of $NC_s^{\min}$ before $t$, and the first one after $t$, and we restrict all input curves to $[t_p : t_s]$.

Lemma C.9 shows that, when the three conditions (i), (ii) and (iii) are satisfied, then we can consider all those segments are parallel. We furthermore restrict those segments to have same length as follows: let $L$ be the smallest of their length. Restrict each segment to one of length $L$, that starts at a point of $NC$ and contains $t$. The following Fact C.4 shows that this restriction is doable. We then build $\gamma'$ using Lemma C.6: with this construction, distances between $\gamma'$ and all input segments are preserved, and Fact C.1 shows that all vertices of $\gamma'$ are at distance at most $9\ell \cdot \Delta\alpha^{-1}$ of an input vertex: they can therefore be replaced by points of $\mathbb{C}$ that are close-by, preserving the Fréchet distance up to an additive $\alpha\Delta$. Applying this strategy for all vertices of $\gamma$ outside of all balls $B(p^i_j, \ell \cdot 100\Delta\alpha^{-2})$, this builds a curve $\widetilde{\gamma}$ with same distance as $\gamma$ to every input curve, up to an additive $\alpha\Delta$.

Now, Lemma C.10 and Lemma C.11 shows that the first two conditions are satisfied for all input segments. Lemma C.12 shows that either the third is satisfied, it is satisfied for a subsegment $s'$ of $s$ that contains $s(t)$. In that case, we can simply replace $s$ by $s'$ in the application of Lemma C.6 – as our construction of $\mathbb{C}$ ensures that there are net points regularly placed on $s'$ as well.

**Fact C.4.** *Let $p[t_p : t_s]$ be an input segment. Then, there is a point of $NC_p$ at distance less than $L$ from $p(t)$.*

*Proof.* Triangle inequality ensures that, for any $i$, $\|p^i(t_p) - p(t_p)\| \leq 16\Delta\alpha^{-1}$, and $\|p^i(t_s) - p(t_s)\| \leq 16\Delta\alpha^{-1}$. Hence, the length of $p[t_p : t_s]$ is at most $L + 32\Delta\alpha^{-1}$, which is at most $2L$ as otherwise the vertex of $\gamma_v$ would be close to an input point. Thus, points of $NC_p$ are placed at distance at most $\alpha^3 L/50$ along the segments, and there is one length $L$ segment starting at this point containing $p(t)$. $\qquad\square$

## D. An Improved Coreset Construction Framework via Chaining

### D.1. Outline of the Argument

In our setting, let $P$ be a set of points in a metric space $\mathcal{M} = (\mathcal{X}, d)$, admitting an $(\varepsilon, k, z)$-clustering net (as per Definition D.5), whose size is function of some $\varepsilon > 0$ (but does not grow too fast in $\varepsilon$). The goal is to describe a framework that, given a metric space and the size of its $(\varepsilon, k, z)$-clustering net, yields small and good coresets. Formally, we prove Theorem D.8, for which we provide an implication diagram for the reader to go back to as well as an outline of the argument below.

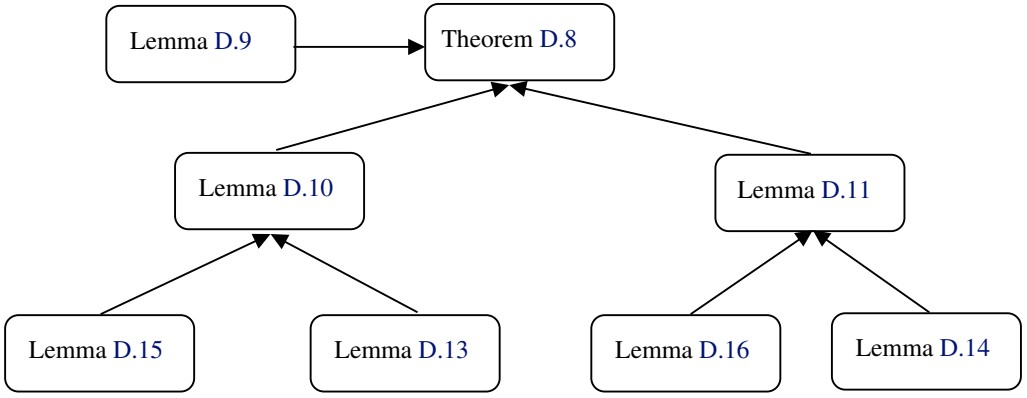

*Figure 3.* Proof overview of the coreset construction framework guarantee: Theorem D.8 is implied by Lemmas D.9, D.10, D.11. In turn, Lemma D.10 is implied by Lemmas D.15, D.13 and Lemma D.11 by Lemmas D.16, D.14.

To prove Theorem D.8, we generalize the conditions under which clustering nets give coresets in general metric spaces. Although parts of this analysis are available elsewhere in the literature (see e.g., (Cohen-Addad et al., 2022a)), we provide the framework in its entirety in this section in hopes that it may help readers entering this field. We now present an overview of the arguments.

In order to show Theorem D.8, we first start from a constant factor approximation $\mathcal{A}$ of the $(k, z)$-clustering objective for point set $P$ on space $\mathcal{M}$. We refer to a cluster in the approximation as $C_i$, which represents the set of points that are closest to center $c_i \in \mathcal{A}$. We then deconstruct the dataset into several non-overlapping *groups* such that they have two properties. First, the intersection of any cluster in $\mathcal{A}$ with a group has roughly equal cost. Second, every point in a group constitutes a roughly equal percentage of its cluster's cost. These conditions ensure[2] that a point from cluster $C_i$ in the group gets sampled with probability roughly $\frac{1}{k|C_i|}$. Furthermore, there are a logarithmic number of such groups (as well as a set of outliers that do not satisfy these properties). Thus, by the fact that coreset property is preserved under composition, if we make a coreset for each group and one for the outliers then we have a coreset for the full dataset.

We thus want to show that sampling from a group gives us a coreset for that group. First, we define the smallest group so that it only induces an $O(\varepsilon)$ fraction of the cost, so each point in this smallest group can be snapped to its closest center without breaking the coreset requirement. Similarly, some clusters in a group will only induce an $O(\varepsilon)$-fraction of the entire cost of $\mathcal{A}$. These can be handled accordingly. The union of those points that can be handled in this way is discussed in Lemma D.9. What remains are those clusters that constitute a significant amount of the cost (the 'standard' groups) and the outliers of the clusters (which we call the 'outer' groups), both of which generally follow a similar argument. We therefore restrict ourselves to the standard groups for the purposes of this exposition.

Given any solution $\mathcal{S}$, each point in the group has a cost with respect to $\mathcal{A}$ and a cost with respect to $\mathcal{S}$. It is the cost with respect to $\mathcal{S}$ that we would like to approximate via the coreset. To do this, we will subdivide every cluster in our group into two sub-types, conditioned on $\mathcal{S}$. The (relatively) easy type consists of those points who have significantly larger cost to $\mathcal{S}$ than to $\mathcal{A}$, by at least a factor of $O(\varepsilon^{-2})$. Intuitively, these points are much closer to each other than they are to their center in $\mathcal{S}$. Thus, as long as we sample enough points from this type, its cost with respect to $\mathcal{S}$ should be well-approximated. Lemma D.11 states that they are sampled sufficiently and that their cost is therefore well-approximated. Thus, those points which incur a huge cost with respect to solution $\mathcal{S}$ are approximated by the sampling.

It remains to show that we appropriately represent those centers that have roughly equal cost in $\mathcal{S}$ and $\mathcal{A}$. This is the most involved part of the analysis. For it, we consider a infinite nested set of clustering nets over the costs of those centers in $\mathcal{S}$ that are in the $O(\varepsilon^{-2})$ range. Specifically, we define a $\frac{1}{2}$-net, a $\frac{1}{4}$-net, and so on. Since the limit of these nets will approach any point, it must be the case that the we can approximate the cost of a center by summing the difference from the first net to the second, from the second to the third, and so on. That is, since the clustering nets give the point in the limit and everything in this telescoping sum cancels, this is equal to the cost of the center with respect to any point. This equality between nested clustering nets and the cost vectors is precisely what we exploit to make the analysis go through: if we can bound the differences in costs between ever-finer epsilon nets, then we can bound the amount that our cost is distorted.

We first model the expected error over the telescoping nets as a Gaussian variable. In this sense, the supremum over the Gaussian variable bounds our desired term. Furthermore, this variable's supremum is less than the sum of suprema over each element in the telescoping sum. Thus, if we can bound the amount of variation at each step in this telescoping sum, we can bound the entire sum and, by extension, the expected error of our center.

A single step in the telescoping sum is expressed by all the possible interactions between the $2^{-i}$-net and the $2^{-(i+1)}$ net. Thus, we use the fact that the maximum of a set of random variables is bounded by their variance times the logarithm of the number of variables:

$$\mathbb{E}\left[\max_{p \in [n]} |g_p|\right] \leq 2\sigma\sqrt{\log n}.$$

where there are $n$ random variables $g_i$ with maximum variance $\sigma$. In our setting, $n$ is the size of the $2^{-i}$ net times the size of the $2^{-(i+1)}$ net. It therefore suffices to bound $\sigma$ – the variance in cost when shifting from an $\varepsilon$-net to a $(\frac{\varepsilon}{2})$-net. Our analysis concludes by doing precisely this.

**Rings.** Let $\Delta_{C_i} = \frac{\text{cost}(C_i, \mathcal{A})}{|C_i|}$ be the average cluster cost and define for each cluster:

---

[2]To see this, consider that we are picking a point from $k$ clusters of equal cost and, conditioned on having picked cluster $C_i$, there are then $|C_i|$ points to pick from.

- Ring $R_{ij} = \{p \in C_i \mid 2^j \Delta_{C_i} \leq \text{cost}(p, \mathcal{A}) \leq 2^{j+1} \Delta_{C_i}\}$, i.e., the set of points that have roughly the same multiple (up to a factor of 2) of the average point's cost in that cluster in the approximate solution.

- Inner ring $R_I(C_i) = \bigcup_{j \leq z \log(\varepsilon/z)} R_{ij}$, i.e., the set of points whose cost in the approximate solution is significantly smaller than the average point's cost in that cluster. We write $R_I = \bigcup_{i \in [k]} R_I(C_i)$.

- Outer ring $R_O(C_i) = \bigcup_{j > 2z \log(z/\varepsilon)} R_{ij}$, i.e., the set of points whose cost in the approximate solution is much larger than the average cost of the cluster they belong to. We write $R_O = \bigcup_{i \in [k]} R_O(C_i)$.

- All those rings that are neither inner nor outer are the main rings. In particular, we let $R_j = \bigcup_{i \in [k]} R_{ij}$, that is the set of points that contribute roughly the same amount of cost to their respective cluster centers.

The fundamental property of any ring $R_{ij}$ is that every point in it contributes roughly the same amount to $\text{cost}(\mathcal{A})$.

**Groups.** Before proceeding with the formal definition of a group $G$, we first give an intuition as to how they are constructed. Each group is comprised of sets of rings. Consider the set of $j$-th rings $R_j$ around the centers in $\mathcal{A}$. Then each ring $R_{ij}$ has some total cost over all of its points. A group $G_{jb}$ is then the subset of these $j$-th rings that have roughly the same total cost (where the $b$ subscript determines how large this cost is). This means that a group $G_{jb}$ is comprised of the $j$-th rings around some subset of the centers in our approximate solution. As a result, groups satisfy the following two properties:

- Each point contributes roughly the same amount to its cluster's cost.

- Each group's cost is evenly divided over the clusters that comprise it.

Formally, main rings are partitioned into main groups as follows. For each $j$, a group is defined as

$$G_{jb} = \left\{ p \mid \exists\, i \text{ s.t. } p \in R_{ij} \text{ and } \frac{2^b}{k}\left(\frac{\varepsilon}{4z}\right)^z \leq \frac{\text{cost}(R_{ij}, \mathcal{A})}{\text{cost}(R_j, \mathcal{A})} \leq \frac{2^{b+1}}{k}\left(\frac{\varepsilon}{4z}\right)^z \right\}.$$

Similarly to rings, we define the "cheap" groups $G_{\min}^M$ to be the union of all groups with $b < 0$. That is, $G_{\min}^M = \bigcup_j \bigcup_{b \leq 0} G_{jb}$. The "main" groups are all those that are not cheap, i.e. $G^M = \bigcup_j \bigcup_{b > 0} G_{jb}$.

With this definition, we say that a ring $R_{ij}$ "belongs" to a group $G_{jb}$ if it contributes roughly the same amount to the total cost as all the other rings in $G_{jb}$. We visualize this in the following diagram:

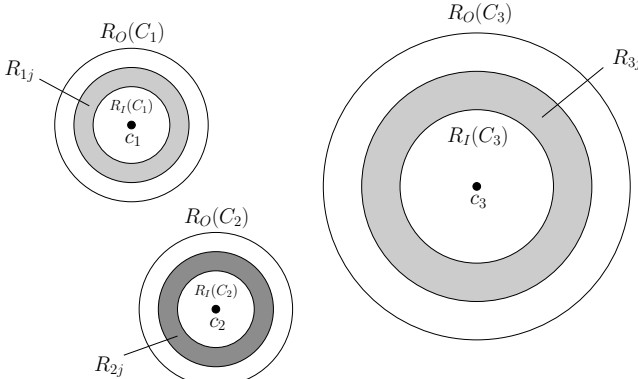

*Figure 4.* We plot the sample rings for $\mathcal{A} = \{c_1, c_2, c_3\}$ in which $\text{cost}(C_1) = \text{cost}(C_2) < \text{cost}(C_3)$. We have labeled the rings $R_{1j}, R_{2j}$ and $R_{3j}$ for $j = 1$ – note that the width of each ring is proportional to the cost of its cluster. Now let us assume that $R_{1j}$ and $R_{3j}$ have equal densities of points but that $R_{2j}$ has higher density (as evidenced by the shading). Then we might have $G_{jb} = R_{1j}$ and $G_{jb'} = R_{2j} \cup R_{3j}$ for $b' > b$, since rings $R_{2j}$ and $R_{3j}$ each have a higher total cost than $R_{1j}$.

Analogous definitions can be given for outer rings:

$$G_b^O = \left\{ p \mid \exists\, i, p \in R_{ij} \text{ and } \frac{2^b}{k}\left(\frac{\varepsilon}{4z}\right)^z \leq \frac{\text{cost}(R_O(C_i), \mathcal{A})}{\text{cost}(R_O, \mathcal{A})} \leq \frac{2^{b+1}}{k}\left(\frac{\varepsilon}{4z}\right)^z \right\}.$$

$G_{\min}^O$ and $G^O$ are defined similarly to $G_{\min}^M$ and $G^M$ above. That is, $G^O$ is the set of all outer groups that are not cheap. For an outer group $G \in G^O$, let $P^G = \{p \mid \exists C, \; p \in C \text{ and } C \cap G \neq \emptyset\}$ be the set of points belonging to the union of clusters that intersect the group. We define the "cheap" group to be $G_{\min} = G_{\min}^M \cup G_{\min}^O \cup R_I$.

**Cluster Types.** When building coresets, we must show that the cost is preserved with respect to *every* solution. It proves useful, however, to consider these solutions in two separate types. Namely, consider a candidate solution $\mathcal{S}$ with respect to cluster $C_i \cap G$, for $C_i \in \mathcal{A}$. If the cost of every point in $C_i \cap G$ is roughly similar in $\mathcal{S}$ as in $\mathcal{A}$, then it implies that there is a candidate center in $\mathcal{S}$ somewhere near center $C_i$. This setting must be handled with care and is what necessitates the upcoming net arguments. We therefore provide the following definition:

**Definition D.1** (Interesting Clusters). Consider an arbitrary solution $\mathcal{S}$ and a main group $G \in G^M$. A cluster $C_i \cap G$ induced by $\mathcal{S}$ is said to be *interesting* if there exists a point $p \in C_i \cap G$ such that $\mathrm{cost}(p, \mathcal{S}) \leq \left(\frac{4z}{\varepsilon}\right)^z \cdot \mathrm{cost}(p, \mathcal{A})$.

On the other hand, if the cost of every point in $C_i \cap G$ is significantly larger with respect to $\mathcal{S}$ than it is to center $c_i$, then it implies that every center $\mathcal{S}$ is far outside of the radius of $C_i \cap G$. In this case, every point is roughly equivalent in terms of $\mathrm{cost}(P, \mathcal{S})$ and it is straightforward to show that we preserve their cost if we sample them sufficiently. We therefore provide the following definitions of these clusters in the main and outer groups:

**Definition D.2** (Huge Clusters). Consider an arbitrary solution $\mathcal{S}$ and a main group $G \in G^M$. A cluster $C_i \cap G$ induced by $\mathcal{S}$ is said to be *huge* if there exists a point $p \in C_i \cap G$ such that $\mathrm{cost}(p, \mathcal{S}) \geq \left(\frac{4z}{\varepsilon}\right)^z \cdot \mathrm{cost}(p, \mathcal{A})$. The set of all *huge* clusters induced by $\mathcal{S}$ in $G$ is indicated by $H_{G, \mathcal{S}}$.

**Definition D.3** (Far Clusters). Consider an arbitrary solution $\mathcal{S}$ and an outer group $G \in G^O$. A cluster $C_i \cap G$ induced by $\mathcal{S}$ is said to be *far* if there exists a point $p \in C_i \cap G$ such that $\mathrm{cost}(p, \mathcal{S}) \geq 4^z \cdot \mathrm{cost}(p, \mathcal{A})$. The set of all *far* clusters induced by $\mathcal{S}$ in $G$ is indicated by $F_{G, \mathcal{S}}$.

**Approximate Centroid Sets, $(\varepsilon, k, z)$-Clustering Nets and $(\varepsilon, k, z)$-Coresets.** For readability's sake, we restate the definition of $(\varepsilon, k, z, \mathcal{A})$-approximate centroid sets, and we introduce another fundamental notion, that of $(\varepsilon, k, z)$-clustering nets, which will prove crucial for the analysis of the coreset construction.

**Definition D.4** ($(\varepsilon, k, z, \mathcal{A})$-approximate centroid set, restatement of Definition C.1). Let $\mathcal{M} = (\mathcal{X}, d)$ be a metric space, $P$ a set of points, $k, z$ two positive integers, and let $\varepsilon \in (0, \frac{1}{2})$ be a precision parameter. Consider an (optimal or approximate) solution $\mathcal{A}$, and let $\mathcal{C} \in \mathcal{M}^k$ be a set of (potentially infinite) $k$-clusterings for the $(k, z)$-clustering objective. We say that $\mathcal{N}$ is an $(\varepsilon, k, z, \mathcal{A})$-approximate centroid set of $P$ if, for every solution $\mathcal{S} \in \mathcal{C}$, there exists another solution $\tilde{\mathcal{S}} \in \mathcal{N}$ such that

$$|\mathrm{cost}(p, \tilde{\mathcal{S}}) - \mathrm{cost}(p, \mathcal{S})| \leq \frac{\varepsilon}{z \log(z/\varepsilon)} \cdot (\mathrm{cost}(p, \mathcal{S}) + \mathrm{cost}(p, \mathcal{A})),$$

for all $p \in P$ with $\mathrm{cost}(p, \mathcal{S}) \leq \left(\frac{4z}{\varepsilon}\right)^z \cdot \mathrm{cost}(p, \mathcal{A})$.

**Definition D.5** ($(\varepsilon, k, z)$-clustering net). Let $\mathcal{M} = (\mathcal{X}, d)$ be a metric space, $P$ a set of points, $k, z$ two positive integers, and let $\varepsilon \in (0, \frac{1}{2})$ be a precision parameter. For a given (optimal or approximate) solution $\mathcal{A}$, let $G$ be a group. Let $\mathcal{C} \in \mathcal{M}^k$ be a set of (potentially infinite) $k$-clusterings for the $(k, z)$-clustering objective. We say that a set of cost vectors $\mathcal{N} \in \mathbb{R}^{|P|}$ is an $(\varepsilon, k, z, \mathcal{A})$-clustering net if, for every solution $\mathcal{S} \in \mathcal{C}$, there exists a vector $v \in \mathcal{N}$ such that the following conditions hold:

$$|v_p - \mathrm{cost}(p, \mathcal{S})| \leq \varepsilon \cdot (\mathrm{cost}(p, \mathcal{S}) + \mathrm{cost}(p, \mathcal{A})), \qquad \forall p \in C \cap G \text{ and } C \notin H_{G, \mathcal{S}} \cup F_{G, \mathcal{S}}$$
$$v_p = 0, \qquad \forall p \in C \cap G \text{ and } C \in H_{G, \mathcal{S}} \cup F_{G, \mathcal{S}}.$$

*Remark* D.6. We remark that one could always obtain an $(\varepsilon, k, z, \mathcal{A})$-clustering net from an $(\varepsilon, k, z, \mathcal{A})$-approximate centroid set, since the latter further requires a solution $\tilde{\mathcal{S}}$, while the former only a set of vectors $\mathcal{N}$.

In the later, we will drop mention of $\mathcal{A}$ in the mention of approximate centroid set and clustering nets, as it is a fixed constant-factor approximation.

**Definition D.7** ($(\varepsilon, k, z)$-coreset). Let $\mathcal{M} = (\mathcal{X}, d)$ be a metric space, $P$ a set of points, $k, z$ two positive integers, and let $\varepsilon > 0$ be a precision parameter. A (weighted) subset $\Omega \subseteq P$ with weights $w : \Omega \to \mathbb{R}$ is said to be an $(\varepsilon, k, z)$-coreset if, for every solution $\mathcal{S}$ to the $(k, z)$-clustering problem, it satisfies

$$\sum_{p \in \Omega} w_p \mathrm{cost}(p, \mathcal{S}) \in (1 \pm \varepsilon) \mathrm{cost}(P, \mathcal{S}).$$

With this setup at hand, we can state the main coreset framework theorem, this time in terms of $(\varepsilon, k, z)$-clustering nets as opposed to $(\varepsilon, k, z)$-approximate centroid sets, due to Remark D.6:

**Theorem D.8** (Restatement of Theorem C.3). *Let $\mathcal{M} = (\mathcal{X}, d)$ be a metric space, and let $U, c \geq 0$ be two constants inherent to the metric space. Assume that $P$ is a set of points such that, for every $h > 0$, the $(2^{-h}, k, z)$-clustering net (succinctly denoted as) $\mathcal{N}_h$ satisfies*

$$|\mathcal{N}_h| \leq \exp\left(Ukz^2 \log |P| \cdot 2^{ch} \log\left(2^h \varepsilon^{-1}\right)\right).$$

*Then, there exists an $(\varepsilon, k, z)$-coreset of size*

$$|\Omega| \leq O\left(2^{O(z \log z)} \cdot Uk \cdot (\varepsilon^{-c} + \varepsilon^{-2}) \cdot \min(\varepsilon^{-z}, k) \cdot \log^6 \varepsilon^{-1} \cdot \log(Uk)\right).$$

## D.2. Preliminary Facts and Considerations on Groups

We list a series of well-known bounds that will be useful in the remainder:

**Fact D.1** (Bernstein's Concentration Inequality). *Let $X_1, \ldots, X_\omega$ be $\omega \in \mathbb{N}$ non-negative independent random variables. Let $S = \sum_{i=1}^{\omega} X_i$. If there exists an almost-sure upper bound $M \geq X_i$, then*

$$\mathbb{P}\left[|S - \mathbb{E}[S]| \geq t\right] \leq \exp\left(-\frac{t^2}{2 \sum_{i=1}^{\omega} \mathbb{V}[X_i] + \frac{2}{3}Mt}\right).$$

**Fact D.2** (Maximum of Independent Gaussian Random Variables, Lemma 2.3 in (Massart, 2007)). *Let $g_p \sim \mathbf{N}(0, \sigma_p)$ be one of $n$ independent normal random variables with $\sigma_p \leq \sigma$ almost surely. It holds that*

$$\mathbb{E}\left[\max_{p \in [n]} |g_p|\right] \leq 2\sigma \sqrt{\log n}.$$

**Fact D.3** (Triangle Inequality for Powers (Makarychev et al., 2019)). *Let $p, q, r$ be three arbitrary points in a metric space $\mathcal{M} = (\mathcal{X}, d)$ and let $z$ be a positive integer. Then, for any $\vartheta > 0$,*

$$d^z(p, q) \leq (1 + \vartheta)^{z-1} \cdot d^z(p, r) + \left(\frac{1+\vartheta}{\vartheta}\right)^{z-1} \cdot d^z(q, r)$$

$$|d^z(p, q) - d^z(p, r)| \leq \varepsilon \cdot d^z(p, r) + \left(\frac{z+\vartheta}{\vartheta}\right)^{z-1} \cdot d^z(q, r).$$

We also provide the following properties of the decomposition into groups:

**Observation D.1.** *Since $2^b \in [0, z\log(4z/\varepsilon)]$ and $2^j \in [2z\log(\varepsilon/z), 2z\log(z/\varepsilon)]$, there are at most $O(z^2 \log^2(z/\varepsilon))$ groups.*

**Claim D.1.** *Let $G \in G^M$ and $C$ be an arbitrary cluster such that $C \cap G \neq \emptyset$. Then,*

$$\mathrm{cost}(G, \mathcal{A}) \leq 2k \cdot \mathrm{cost}(C \cap G, \mathcal{A}) \leq 4k \cdot |C \cap G| \cdot \mathrm{cost}(p, \mathcal{A}),$$

*for all points $p \in C \cap G$.*

*Proof.* Without loss of generality, let $G$ be $G_{jb}$ for some $j$ and $b$ and let $C$ be $C_i$ and recall that $G \subseteq R_j$. Then $R_{ij} = G \cap C_i$ and, by the definition of groups, we have

$$\frac{2^b}{k}\left(\frac{\varepsilon}{4z}\right)^z \leq \frac{\mathrm{cost}(G \cap C_i, \mathcal{A})}{\mathrm{cost}(R_j, \mathcal{A})} \leq \frac{2^{b+1}}{k}\left(\frac{\varepsilon}{4z}\right)^z.$$

Now let $R_{\ell j}$ be the cheapest ring in $R_j$ that is also in group $G_{jb}$. So $\mathrm{cost}(R_{\ell j}, \mathcal{A}) \leq \mathrm{cost}(R_{ij}, \mathcal{A})$. Then

$$\frac{2^b}{k}\left(\frac{\varepsilon}{4z}\right)^z \mathrm{cost}(R_j, \mathcal{A}) \leq \mathrm{cost}(R_{\ell j}, \mathcal{A}) \quad \text{and} \quad \mathrm{cost}(R_{ij}, \mathcal{A}) \leq \frac{2^{b+1}}{k}\left(\frac{\varepsilon}{4z}\right)^z \mathrm{cost}(R_j, \mathcal{A}),$$

---

**Algorithm 1** Group Sampling

---

**Input:** Dataset $P$, approximate solution $\mathcal{A}$ and number of clusters $k$
**Output:** A coreset $\Omega$ of points and weights

1: Let $\omega = 2^{\lambda z \log z} \cdot U k z^2 \cdot \varepsilon^{-c} \log^3 \varepsilon^{-1} \cdot \min(\varepsilon^{-z}, k)$
2: Initialize $\Omega \leftarrow \emptyset$
3: **for** $i = 1, \ldots, k$ **do**
4:     Let $w_{c_i} = |C_i \cap G_{\min}|$
5:     Update $\Omega \leftarrow \Omega \cup \{(c_i, w_{c_i})\}$
6: **end for**
7: **for** $G \in G^M \cup G^O$ **do**
8:     **for** $t = 1, \ldots, \omega$ **do**
9:         Sample point $p \in G$ with probability $\frac{\text{cost}(p, \mathcal{A})}{\text{cost}(G, \mathcal{A})}$
10:         Let $w_p = \frac{\text{cost}(G, \mathcal{A})}{\omega \cdot \text{cost}(p, \mathcal{A})}$
11:         Update $\Omega \leftarrow \Omega \cup \{(p, w_p)\}$
12:     **end for**
13: **end for**
14: Return $\Omega$

---

which, rearranged, give

$$\text{cost}(R_j, \mathcal{A}) \leq \frac{k}{2^b} \left(\frac{4z}{\varepsilon}\right)^z \text{cost}(R_{\ell j}, \mathcal{A}) \quad \text{and} \quad \frac{k}{2^{b+1}} \left(\frac{4z}{\varepsilon}\right)^z \text{cost}(R_{ij}, \mathcal{A}) \leq \text{cost}(R_j, \mathcal{A}).$$

Since $R_{\ell j}$ is cheaper than $R_{ij}$, we can then write

$$\frac{k}{2^{b+1}} \left(\frac{4z}{\varepsilon}\right)^z \text{cost}(R_{\ell j}, \mathcal{A}) \leq \frac{k}{2^{b+1}} \left(\frac{4z}{\varepsilon}\right)^z \text{cost}(R_{ij}, \mathcal{A}) \leq \frac{k}{2^b} \left(\frac{4z}{\varepsilon}\right)^z \text{cost}(R_{\ell j}, \mathcal{A})$$

$$\text{cost}(R_{\ell j}, \mathcal{A}) \leq \text{cost}(R_{ij}, \mathcal{A}) \leq 2\text{cost}(R_{\ell j}, \mathcal{A})$$

This immediately gives us the first inequality:

$$\text{cost}(G, \mathcal{A}) \leq \text{cost}(R_j, \mathcal{A}) = \sum_{a \in [k]} \text{cost}(R_{aj}, \mathcal{A})$$

$$\leq \sum_{a \in [k]} 2\text{cost}(R_{\ell j}, \mathcal{A}) \leq 2k \cdot \text{cost}(R_{ij}, \mathcal{A}) = 2k \cdot \text{cost}(C_i \cap G, \mathcal{A}).$$

For the second inequality, recall the definition of ring $R_{ij}$: for all $p \in R_{ij}$, it holds that $2^j \Delta_{C_i} \leq \text{cost}(p, \mathcal{A}) \leq 2^{j+1} \Delta_{C_i}$. We can again do a similar argument as above: let $q$ be the cheapest point in $R_{ij}$, so that $\text{cost}(q, \mathcal{A}) \leq \text{cost}(p, \mathcal{A})$ for all $p \in C_i \cap G$. As was the case above, this implies that $\text{cost}(p, \mathcal{A}) \leq 2\text{cost}(q, \mathcal{A})$. Now observe that

$$2k \cdot \text{cost}(C_i \cap G, \mathcal{A}) = 2k \cdot \sum_{p \in C_i \cap G} \text{cost}(p, \mathcal{A}) \leq 4k \cdot |C_i \cap G| \cdot \text{cost}(q, \mathcal{A}) \leq 4k \cdot |C_i \cap G| \cdot \text{cost}(p, \mathcal{A}).$$

This concludes the proof. $\qquad\square$

### D.3. Group Sampling Algorithm

In the previous section, we observed that there are at most $O(z^2 \log^2(z\varepsilon^{-1}))$ groups. Thus, the idea is to construct coresets $\Omega_G$ for each such group and then output the union of all these coresets. As groups are well-structured and points in them contribute about the same to their overall cost, a sensitivity sampling procedure is all we need to build small coresets.

Algorithm 1 has two main steps. We first take all of the points in the 'cheap' group and collapse them onto the centers of our approximate solution. We then perform sensitivity sampling on the remaining main and outer groups.

### D.4. The Cheap Group Doesn't Matter

We first observe that we can collapse the points in the cheap group to their closest cluster centers in $\mathcal{A}$.

**Lemma D.9** (Cheap Group Estimate). *Let $P$ be a set of points in a metric space $\mathcal{M} = (\mathcal{X}, d)$, and let $\mathcal{S}$ be an arbitrary solution for $(k, z)$-clustering. Then,*

$$\left| \text{cost}(G_{\min}, \mathcal{S}) - \sum_{i \in [k]} |C_i \cap G_{\min}| \cdot \text{cost}(c_i, \mathcal{S}) \right| \leq \varepsilon \left( \text{cost}(P, \mathcal{S}) + \text{cost}(P, \mathcal{A}) \right).$$

*Proof.* Recall that the cheap group consists of three different subsets: the cheap rings $R_I$, the cheap main groups $G_{\min}^M$ and the cheap outer groups $G_{\min}^O$. Let $\mathcal{S}$ be any solution and let $\text{cost}(G_{\min}, \mathcal{S})$ be the cost of the cheap group with respect to $\mathcal{S}$. Let $p$ be any point in $G_{\min}$ such that $p \in C_i$. Then, by the triangle inequality for powers with $\vartheta = \varepsilon$, we can upper bound $\text{cost}(p, \mathcal{S}) \leq (1 + 1/\varepsilon)^{z-1}\text{cost}(p, c_i) + (1 + \varepsilon)^{z-1}\text{cost}(c_i, \mathcal{S})$. Doing this for all such points in $C_i \cap G_{\min}$ gives us

$$\text{cost}(C_i \cap G_{\min}, \mathcal{S}) \leq \left(1 + \frac{1}{\varepsilon}\right)^{z-1} \text{cost}(C_i \cap G_{\min}, \mathcal{A}) + (1 + \varepsilon)^{z-1}|C_i \cap G_{\min}| \cdot \text{cost}(c_i, \mathcal{S}).$$

Applying this over all centers in $\mathcal{A}$ leads to

$$\text{cost}(G_{\min}, \mathcal{S}) \leq \left(1 + \frac{1}{\varepsilon}\right)^{z-1} \text{cost}(G_{\min}, \mathcal{A}) + (1 + \varepsilon)^{z-1} \sum_{i \in [k]} |C_i \cap G_{\min}| \cdot \text{cost}(c_i, \mathcal{S})$$

$$\leq \left(1 + \frac{1}{\varepsilon}\right)^{z-1} \text{cost}(G_{\min}, \mathcal{A}) + (1 + z^{z+1}\varepsilon) \sum_{i \in [k]} |C_i \cap G_{\min}| \cdot \text{cost}(c_i, \mathcal{S}),$$

where the inequality holds because $(1 + \varepsilon)^{z-1} \leq 1 + z^{z+1}\varepsilon$ for $0 \leq \varepsilon < 1$. We now plug this into the statement from the lemma:

$$\text{cost}(G_{\min}, \mathcal{S}) - \sum_{i \in [k]} |C_i \cap G_{\min}| \cdot \text{cost}(c_i, \mathcal{S}) \leq z^{z+1}\varepsilon \sum_{i \in [k]} |C_i \cap G_{\min}| \cdot \text{cost}(c_i, \mathcal{S})$$

$$+ \left(1 + \frac{1}{\varepsilon}\right)^{z-1} \text{cost}(G_{\min}, \mathcal{A}),$$

which implies

$$\left| \text{cost}(G_{\min}, \mathcal{S}) - \sum_{i \in [k]} |C_i \cap G_{\min}| \cdot \text{cost}(c_i, \mathcal{S}) \right| \leq z^{z+1}\varepsilon \sum_{i \in [k]} |C_i \cap G_{\min}| \cdot \text{cost}(c_i, \mathcal{S}) \tag{17}$$

$$+ \left(1 + \frac{1}{\varepsilon}\right)^{z-1} \text{cost}(G_{\min}, \mathcal{A}). \tag{18}$$

It remains to show that Equation (17) is less than $O(\varepsilon)\text{cost}(P, \mathcal{S})$ and that Equation (18) is less than $O(\varepsilon)\text{cost}(P, \mathcal{A})$. The former is trivial to show, since $\sum_{i \in [k]} |C_i \cap G_{\min}| \cdot \text{cost}(c_i, \mathcal{S}) \leq \text{cost}(P, \mathcal{S})$. For the latter, we break it up into its constituent components and show that each one is very cheap:

Consider the sets comprising $G_{\min}$. For the inner rings $R_I$, we have for any $p \in R_I \cap C_i$

$$\text{cost}(p, \mathcal{A}) \leq 2^{z \log(\varepsilon/z)} \Delta_{C_i} = \left(\frac{\varepsilon}{z}\right)^z \frac{\text{cost}(C_i, \mathcal{A})}{|C_i|}.$$

Therefore,

$$\text{cost}(R_I \cap C_i, \mathcal{A}) \leq \left(\frac{\varepsilon}{z}\right)^z \text{cost}(C_i, \mathcal{A}) \implies \text{cost}(R_I, \mathcal{A}) \leq \left(\frac{\varepsilon}{z}\right)^z \text{cost}(P, \mathcal{A}) = 4^z \left(\frac{\varepsilon}{4z}\right)^z \text{cost}(P, \mathcal{A}).$$

For the inner main groups, we have for any $R_{ij} \in G_{\min}^M \cap R_j$

$$\frac{\text{cost}(R_{ij}, \mathcal{A})}{\text{cost}(R_j, \mathcal{A})} \leq \frac{2}{k} \left(\frac{\varepsilon}{4z}\right)^z \implies \text{cost}(R_{ij}, \mathcal{A}) \leq \frac{2}{k} \left(\frac{\varepsilon}{4z}\right)^z \text{cost}(R_j, \mathcal{A}),$$

and thus,

$$\text{cost}(G_{\min}^M \cap R_j, \mathcal{A}) \leq 2 \left(\frac{\varepsilon}{4z}\right)^z \text{cost}(R_j, \mathcal{A}) \leq 4^z \left(\frac{\varepsilon}{4z}\right)^z \text{cost}(R_j, \mathcal{A}).$$

We can similarly show for the outer groups that

$$\text{cost}(G_{\min}^M \cap R_O, \mathcal{A}) \leq 4^z \left(\frac{\varepsilon}{4z}\right)^z \text{cost}(R_O, \mathcal{A}).$$

Since $G_{\min} = R_I \cup G_{\min}^M \cup G_{\min}^O$, we therefore have

$$\text{cost}(G_{\min}, \mathcal{A}) \leq 4^z \left(\frac{\varepsilon}{4z}\right)^z \left(\text{cost}(P, \mathcal{A}) + \sum_j \text{cost}(R_j, \mathcal{A}) + \text{cost}(R_O, \mathcal{A})\right)$$

$$\leq 2^{2z} \left(\frac{\varepsilon}{4z}\right)^z \cdot 2 \cdot \text{cost}(P, \mathcal{A})$$

$$\leq \frac{2}{z^z} \varepsilon^z \text{cost}(P, \mathcal{A}).$$

Hence,

$$\left(1 + \frac{1}{\varepsilon}\right)^{z-1} \text{cost}(G_{\min}, \mathcal{A}) \leq \left(1 + \frac{1}{\varepsilon}\right)^{z-1} \frac{2}{z^z} \varepsilon^z \text{cost}(P, \mathcal{A}).$$

It is a matter of algebra[3] to show that $\left(1 + \frac{1}{\varepsilon}\right)^{z-1} \frac{2}{z^z} \varepsilon^z \leq 2z\varepsilon$. Thus, we have

$$\left(1 + \frac{1}{\varepsilon}\right)^{z-1} \text{cost}(G_{\min}, \mathcal{A}) \leq 2z\varepsilon \cdot \text{cost}(P, \mathcal{A}) \tag{19}$$

We can now plug this into Equation (18) to obtain

$$\left|\text{cost}(G_{\min}, \mathcal{S}) - \sum_{i \in [k]} |C_i \cap G_{\min}| \cdot \text{cost}(c_i, \mathcal{S})\right| \leq z^{z+1}\varepsilon \cdot \text{cost}(P, \mathcal{S}) + 2z\varepsilon \cdot \text{cost}(P, \mathcal{A})$$

$$\leq 2z^{z+1}\varepsilon \cdot (\text{cost}(P, \mathcal{S}) + \text{cost}(P, \mathcal{A})).$$

Rescaling $\varepsilon$ then gives the desired result. $\qquad \square$

This implies, in turn, that the remainder of the paper will only ever consider "main" or "outer" groups.

### D.5. Coreset Validity

Given a solution $\mathcal{S}$ and a coreset $\Omega$ for a group $G \in \mathcal{G}$, Algorithm 1 returns an estimated cost equal to $\sum_{p \in G \cap \Omega} w_p \text{cost}(p, \mathcal{S})$, and so we are interested in studying the error estimator

$$D_{\mathcal{S}}^\Omega(G) = \left|\sum_{p \in G \cap \Omega} w_p \text{cost}(p, S) - \text{cost}(G, S)\right|.$$

The following lemmas state how significant (in terms of a specified accuracy) the deviation of the estimated cost from the true cost can be for main and outer groups.

---

[3]To see this, consider that $(1 + 1/\varepsilon)^{z-1}$ has a binomial expansion where every term has a binomial coefficient less than $z^z$ and the terms themselves are powers of $1/\varepsilon$. Thus, dividing by $z^z$ and multiplying by $\varepsilon^z$ means that each term is less than $\varepsilon$. There are at most $z$ such terms.

**Lemma D.10** (Main Groups Estimate)**.** *Let $G \subseteq G^M$ be a main group, and let $\lambda > 4$ be an appropriately chosen constant. Then, for $\omega = 2^{\lambda z \log z} \cdot Ukz^2 \cdot (\varepsilon^{-c} + \varepsilon^{-2}) \cdot \min(\varepsilon^{-z}, k) \cdot \log^4 \varepsilon^{-1} \cdot \log(Ukz)$, Algorithm 1 yields*

$$\mathbb{E}\left[\sup_{\mathcal{S}} \frac{D_{\mathcal{S}}^{\Omega}(G)}{\text{cost}(G, \mathcal{S}) + \text{cost}(G, \mathcal{A})}\right] \leq \varepsilon.$$

**Lemma D.11** (Outer Groups Estimate)**.** *Let $G \subseteq G^O$ be an outer group, and let $\lambda > 4$ be an appropriately chosen constant. Then, for $\omega = 2^{\lambda z \log z} \cdot Ukz^2 \cdot (\varepsilon^{-c} + \varepsilon^{-2}) \cdot \min(\varepsilon^{-z}, k) \cdot \log^4 \varepsilon^{-1} \cdot \log(Ukz)$, Algorithm 1 yields*

$$\mathbb{E}\left[\sup_{\mathcal{S}} \frac{D_{\mathcal{S}}^{\Omega}(G)}{\text{cost}(P^G, \mathcal{S}) + \text{cost}(P^G, \mathcal{A})}\right] \leq \varepsilon.$$

We prove these lemmas in the following sections, but show why they imply Theorem D.8 next.

*Proof of Theorem D.8.* Let us consider $\Omega_G$ as the coreset returned by Algorithm 1 for a group $G \in \mathcal{G}$. Similarly, let $\Omega_{\mathcal{G}}$ be the union of all such coresets. We have:

$$\mathbb{E}\left[\sup_{\mathcal{S}} \frac{D_{\mathcal{S}}^{\Omega_{\mathcal{G}}}(P \cap \mathcal{G})}{\text{cost}(P, \mathcal{S}) + \text{cost}(P, \mathcal{A})}\right]$$

$$= \mathbb{E}\left[\sup_{\mathcal{S}} \frac{1}{\text{cost}(P, \mathcal{S}) + \text{cost}(P, \mathcal{A})} \cdot \left|\sum_{G \in \mathcal{G}} \sum_{p \in \Omega_G} w_p \text{cost}(p, S) - \text{cost}(G, S)\right|\right]$$

$$\leq \mathbb{E}\left[\sup_{\mathcal{S}} \frac{1}{\text{cost}(P, \mathcal{S}) + \text{cost}(P, \mathcal{A})} \cdot \sum_{G \in \mathcal{G}} D_{\mathcal{S}}^{\Omega_G}(G)\right]$$

$$\leq \mathbb{E}\left[\sup_{\mathcal{S}} \sum_{G \in G^M} \frac{\text{cost}(G, \mathcal{S}) + \text{cost}(G, \mathcal{A})}{\text{cost}(P, \mathcal{S}) + \text{cost}(P, \mathcal{A})} \cdot \mathbb{E}\left[\sup_{\mathcal{S}} \frac{D_{\mathcal{S}}^{\Omega_G}(G)}{\text{cost}(G, \mathcal{S}) + \text{cost}(G, \mathcal{A})}\right]\right.$$

$$\left. \sum_{G \in G^O} \frac{\text{cost}(P^G, \mathcal{S}) + \text{cost}(P^G, \mathcal{A})}{\text{cost}(P, \mathcal{S}) + \text{cost}(P, \mathcal{A})} \cdot \mathbb{E}\left[\sup_{\mathcal{S}} \frac{D_{\mathcal{S}}^{\Omega_G}(G)}{\text{cost}(P^G, \mathcal{S}) + \text{cost}(P^G, \mathcal{A})}\right]\right]$$

$$\leq \mathbb{E}\left[\sup_{\mathcal{S}} \sum_{G \in G^M} \frac{\text{cost}(G, \mathcal{S}) + \text{cost}(G, \mathcal{A})}{\text{cost}(P, \mathcal{S}) + \text{cost}(P, \mathcal{A})} \cdot \varepsilon + \sum_{G \in G^O} \frac{\text{cost}(P^G, \mathcal{S}) + \text{cost}(P^G, \mathcal{A})}{\text{cost}(P, \mathcal{S}) + \text{cost}(P, \mathcal{A})} \cdot \varepsilon\right] \quad \text{(Lemmas D.10 and D.11)}$$

$$\leq 2\varepsilon.$$

By Markov's inequality, we thus have that

$$\mathbb{P}\left[\sup_{\mathcal{S}} \frac{D_{\mathcal{S}}^{\Omega_{\mathcal{G}}}(P \cap \mathcal{G})}{\text{cost}(P, \mathcal{S}) + \text{cost}(P, \mathcal{A})} \leq 8\varepsilon\right] \geq 1 - \frac{1}{8\varepsilon} \cdot \mathbb{E}\left[\sup_{\mathcal{S}} \frac{D_{\mathcal{S}}^{\Omega_{\mathcal{G}}}(P \cap \mathcal{G})}{\text{cost}(P, \mathcal{S}) + \text{cost}(P, \mathcal{A})}\right] \geq \frac{3}{4}.$$

That is, $D_{\mathcal{S}}^{\Omega_{\mathcal{G}}}(P \cap \mathcal{G}) \leq 8\varepsilon \cdot (\text{cost}(P, \mathcal{S}) + \text{cost}(P, \mathcal{A}))$ for all $\mathcal{S}$, with probability at least $\frac{3}{4}$. We also need to take into account all points that do not belong to any group, and their corresponding cost estimator. It holds that

$$D_{\mathcal{S}}^{\Omega}(P) \leq D_{\mathcal{S}}^{\Omega_{\mathcal{G}}}(P \cap \mathcal{G}) + \left|\text{cost}(P \setminus \mathcal{G}, S) - \sum_{i \in [k]} |P \setminus \mathcal{G} \cap C_i| \cdot \text{cost}(c_i, S)\right|$$

$$\leq 9\varepsilon \cdot (\text{cost}(P, \mathcal{S}) + \text{cost}(P, \mathcal{A})), \quad \text{(Lemma D.9)}$$

again for all $\mathcal{S}$, with probability at least $\frac{3}{4}$. Since $\text{cost}(P, \mathcal{A}) \leq \alpha \cdot \text{OPT} \leq \alpha \cdot \text{cost}(P, \mathcal{S})$ for any $\mathcal{S}$, where $\alpha$ is the approximation factor of solution $\mathcal{A}$, it suffices to rescale $\varepsilon$ by a factor $9(1 + \alpha)$ to get that $D_{\mathcal{S}}^{\Omega}(P) \leq \varepsilon \cdot \text{cost}(P, \mathcal{S})$ as required.

We are left to bound the coreset size. To this end, we recall from Lemmas D.10 and D.11 that $\omega = 2^{\lambda z \log z} \cdot Ukz^2 \cdot (\varepsilon^{-c} + \varepsilon^{-2}) \cdot \min(\varepsilon^{-z}, k) \cdot \log^4 \varepsilon^{-1} \cdot \log(Ukz)$ for each group. Since, by Observation D.1, there at at most $O(z^2 \log^2(z\varepsilon^{-1}))$ many such groups, then the coreset size is simply

$$|\Omega| \leq O\left(2^{O(z \log z)} \cdot Uk \cdot (\varepsilon^{-c} + \varepsilon^{-2}) \cdot \min(\varepsilon^{-z}, k) \cdot \log^6 \varepsilon^{-1} \cdot \log(Uk)\right),$$

which concludes the proof. □

### D.6. Groups Estimates via Chaining

To show Lemma D.10 and Lemma D.11, we need to show concentration of the error estimator $D_\mathcal{S}^\Omega(G)$ around 0, whether $G$ is a main or an outer group. In both cases, however, the cost estimator $\sum_{p \in G \cap \Omega} w_p \text{cost}(p, \mathcal{S})$ has too large a variance. To simplify the exposition, let $v_p^\mathcal{S} = \text{cost}(p, \mathcal{S})$ be a $|P|$-dimensional cost vector for all $p \in P$ and any fixed solution $\mathcal{S}$; subsequently, let $v_p^{G,\mathcal{S}} = v_p^\mathcal{S} \cdot \mathbb{1}_{\{p \in G\}}$. Furthermore, we denote by

$$u_p^{G,\mathcal{S}} = \text{cost}(p, \mathcal{S}) \cdot \mathbb{1}_{\{p \in C \cap G \text{ and } C \in H_{G,s}\}}, \ \forall G \in G^M$$
$$u_p^{G,\mathcal{S}} = \text{cost}(p, \mathcal{S}) \cdot \mathbb{1}_{\{p \in C \cap G \text{ and } C \in F_{G,s}\}}, \ \forall G \in G^O,$$

where $H_{G,S}$ and $F_{G,S}$ are given in Definitions D.2 and D.3.

Now, as mentioned, we do not want to estimate $\|v^{G,\mathcal{S}}\|_1$ directly, but instead, we split it as $v^{G,\mathcal{S}} = v^{G,\mathcal{S}} - u^{G,\mathcal{S}} + u^{G,\mathcal{S}}$. This allows us to estimate $\|v^{G,\mathcal{S}} - u^{G,\mathcal{S}}\|_1$ and $\|u^{G,\mathcal{S}}\|_1$ separately since

$$\|v^{G,\mathcal{S}}\|_1 = \|v^{G,\mathcal{S}} - u^{G,\mathcal{S}}\|_1 + \|u^{G,\mathcal{S}}\|_1,$$

which derives from $v^{G,\mathcal{S}}, u^{G,\mathcal{S}}$ having nonnegative entries only.

**Well-Behaved Clusters.** We use a chaining argument to estimate $\|v^{G,\mathcal{S}} - u^{G,\mathcal{S}}\|_1$, which is articulated as follows: Consider an infinite sequence of $|P|$-dimensional vectors $v^{\mathcal{S},1}, v^{\mathcal{S},2}, \ldots$, where $v^{\mathcal{S},h}$ is such that

$$|\text{cost}(p, \mathcal{S}) - v_p^{\mathcal{S},h}| \leq 2^{-h} \cdot (\text{cost}(p, \mathcal{S}) + \text{cost}(p, \mathcal{A})),$$

if $p \in C \cap G$ and $C \cap G \notin H_{G,\mathcal{S}} \cup F_{G,\mathcal{S}}$, and $v_p^{\mathcal{S},h} = 0$ otherwise. In other words, $v^{\mathcal{S},h}$ is a vector belonging to a $2^{-h}$-net $\mathcal{N}_h$ approximating cost vector $v^{G,\mathcal{S}} - u^{G,\mathcal{S}}$. Let us now consider random variables

$$Y_{G,p,\mathcal{S}} = w_p v_p^{\mathcal{S},1} + \sum_{h=1}^\infty w_p (v_p^{\mathcal{S},h+1} - v_p^{\mathcal{S},h})$$

$$Y_{G,\mathcal{S}} = \sum_{p \in \Omega} Y_{G,p,\mathcal{S}}.$$

Since $v_p^{\mathcal{S},h} \xrightarrow[h \to \infty]{} v_p^{G,\mathcal{S}} - u_p^{G,\mathcal{S}}$ so that the sum is well-defined, and $Y_{G,p,\mathcal{S}} = w_p v_p^{G,\mathcal{S}} - u_p^{G,\mathcal{S}}$ as the sum telescopes, we observe that $\mathbb{E}[Y_{G,\mathcal{S}}] = \|v^{G,\mathcal{S}} - u^{G,\mathcal{S}}\|_1$. This means that we can estimate $\|v^{G,\mathcal{S}} - u^{G,\mathcal{S}}\|_1$ through $Y_{G,\mathcal{S}}$. In turn, we can write the conditions to be proven in Lemma D.10 and Lemma D.11 as

$$\mathbb{E}\left[\sup_\mathcal{S} \frac{Y_{G,\mathcal{S}} - \mathbb{E}[Y_{G,\mathcal{S}}]}{\text{cost}(G, \mathcal{S}) + \text{cost}(G, \mathcal{A})}\right] \leq \varepsilon$$

$$\mathbb{E}\left[\sup_\mathcal{S} \frac{Y_{G,\mathcal{S}} - \mathbb{E}[Y_{G,\mathcal{S}}]}{\text{cost}(P^G, \mathcal{S}) + \text{cost}(P^G, \mathcal{A})}\right] \leq \varepsilon.$$

It is not immediate how to exploit a chaining argument via weighted Boolean variables. We, thus, make use of the following symmetrization argument for which we define Gaussian random variables $X_{G,\mathcal{S}}$, whose cost estimates do not deviate from the ones of $Y_{G,\mathcal{S}}$ by too much. Formally, let us define

$$X_{G,\mathcal{S}} = \frac{\sum_{p \in \Omega} \left(g_p \cdot w_p v_p^{\mathcal{S},1} + \sum_{h=1}^\infty g_p \cdot w_p (v_p^{\mathcal{S},h+1} - v_p^{\mathcal{S},h})\right)}{\text{cost}(G, \mathcal{S}) + \text{cost}(G, \mathcal{A})}, \ \forall G \in G^M$$

$$X_{G,\mathcal{S}} = \frac{\sum_{p \in \Omega} \left( g_p \cdot w_p v_p^{\mathcal{S},1} + \sum_{h=1}^{\infty} g_p \cdot w_p (v_p^{\mathcal{S},h+1} - v_p^{\mathcal{S},h}) \right)}{\mathrm{cost}(P^G, \mathcal{S}) + \mathrm{cost}(P^G, \mathcal{A})}, \quad \forall G \in G^O,$$

where $g_p \sim \mathbf{N}(0,1)$ is one of $\omega$ independent standard normal random variables. It holds that:

**Lemma D.12** (Cost Symmetrization, Appendix B.3 in (Rudra & Wootters, 2014)). *Let* $T = \frac{1}{\mathrm{cost}(G,\mathcal{S}) + \mathrm{cost}(G,\mathcal{A})}$ *or* $T = \frac{1}{\mathrm{cost}(P^G,\mathcal{S}) + \mathrm{cost}(P^G,\mathcal{A})}$. *Then,*

$$\mathbb{E}_{\Omega} \left[ \sup_{\mathcal{S}} \left| \sum_{p \in \Omega} T \cdot (Y_{G,p,\mathcal{S}} - \mathbb{E}\left[Y_{G,p,\mathcal{S}}\right]) \right| \right] \leq \sqrt{2\pi} \cdot \mathbb{E}_{\Omega,g} \left[ \sup_{\mathcal{S}} |X_{G,\mathcal{S}}| \right].$$

With these at hand, our goal will be to show the following lemmas:

**Lemma D.13** (Gaussian Process for Main Groups Cost). *Let* $G \in G^M$. *For* $\omega = 2^{\lambda z \log z} \cdot U k z^2 \cdot (\varepsilon^{-c} + \varepsilon^{-2}) \cdot \min(\varepsilon^{-z}, k) \cdot \log^4 \varepsilon^{-1} \cdot \log(U k z)$, *we have*

$$\mathbb{E}_{\Omega,g} \left[ \sup_{\mathcal{S}} |X_{G,\mathcal{S}}| \right] \leq \varepsilon.$$

**Lemma D.14** (Gaussian Process for Outer Groups Cost). *Let* $G \in G^O$. *For* $\omega = 2^{\lambda z \log z} \cdot U k z^2 \cdot (\varepsilon^{-c} + \varepsilon^{-2}) \cdot \min(\varepsilon^{-z}, k) \cdot \log^4 \varepsilon^{-1} \cdot \log(U k z)$, *we have*

$$\mathbb{E}_{\Omega,g} \left[ \sup_{\mathcal{S}} |X_{G,\mathcal{S}}| \right] \leq \varepsilon.$$

**Huge and far clusters.** Estimating $\|u^{G,\mathcal{S}}\|_1$ does not require a chaining argument but a refined control over the estimator's variance. Specifically, we will show the following lemmas:

**Lemma D.15** (Huge Clusters Estimate). *Let* $G \in G^M$. *For* $\omega = 2^{\lambda z \log z} \cdot U k z^2 \cdot (\varepsilon^{-c} + \varepsilon^{-2}) \cdot \min(\varepsilon^{-z}, k) \cdot \log^4 \varepsilon^{-1} \cdot \log(U k z)$, *we have*

$$\mathbb{E}_{\Omega} \left[ \sup_{\mathcal{S}} \left| \frac{\sum_{p \in \Omega} w_p u_p^{G,\mathcal{S}} - \|u^{G,\mathcal{S}}\|_1}{\mathrm{cost}(G,\mathcal{S}) + \mathrm{cost}(G,\mathcal{A})} \right| \right] \leq \varepsilon.$$

**Lemma D.16** (Far Clusters Estimate). *Let* $G \in G^O$. *For* $\omega = 2^{\lambda z \log z} \cdot U k z^2 \cdot (\varepsilon^{-c} + \varepsilon^{-2}) \cdot \min(\varepsilon^{-z}, k) \cdot \log^4 \varepsilon^{-1} \cdot \log(U k z)$, *we have*

$$\mathbb{E}_{\Omega} \left[ \sup_{\mathcal{S}} \left| \frac{\sum_{p \in \Omega} w_p u_p^{G,\mathcal{S}} - \|u^{G,\mathcal{S}}\|_1}{\mathrm{cost}(P^G,\mathcal{S}) + \mathrm{cost}(P^G,\mathcal{A})} \right| \right] \leq \varepsilon.$$

Proving Lemmas D.13, D.14, D.15, D.16 is the crux of the analysis. Before proceeding with those, we show how they imply, together with Lemma D.12, Lemma D.10 and Lemma D.11.

*Proof of Lemma D.10.* We have that

$$\mathbb{E}_{\Omega} \left[ \sup_{\mathcal{S}} \frac{D_{\mathcal{S}}^{\Omega}(G)}{\mathrm{cost}(G,\mathcal{S}) + \mathrm{cost}(G,\mathcal{A})} \right]$$

$$= \mathbb{E}_{\Omega} \left[ \sup_{\mathcal{S}} \left| \frac{\|v^{G,\mathcal{S}} - u^{G,\mathcal{S}}\|_1 + \|u^{G,\mathcal{S}}\|_1 - \sum_{p \in \Omega} \left( w_p u_p^{G,\mathcal{S}} + w_p (v_p^{G,\mathcal{S}} - u_p^{G,\mathcal{S}}) \right)}{\mathrm{cost}(G,\mathcal{S}) + \mathrm{cost}(G,\mathcal{A})} \right| \right]$$

$$\leq \mathbb{E}_{\Omega} \left[ \sup_{\mathcal{S}} \left| \frac{\sum_{p \in \Omega} w_p u_p^{G,\mathcal{S}} - \|u^{G,\mathcal{S}}\|_1}{\mathrm{cost}(G,\mathcal{S}) + \mathrm{cost}(G,\mathcal{A})} \right| \right] + \mathbb{E}_{\Omega} \left[ \sup_{\mathcal{S}} \left| \frac{\sum_{p \in \Omega} w_p (v_p^{G,\mathcal{S}} - u_p^{G,\mathcal{S}}) - \|v^{G,\mathcal{S}} - u^{G,\mathcal{S}}\|_1}{\mathrm{cost}(G,\mathcal{S}) + \mathrm{cost}(G,\mathcal{A})} \right| \right]$$

$$\leq \mathbb{E}_{\Omega} \left[ \sup_{\mathcal{S}} \left| \frac{\sum_{p \in \Omega} w_p u_p^{G,\mathcal{S}} - \|u^{G,\mathcal{S}}\|_1}{\mathrm{cost}(G,\mathcal{S}) + \mathrm{cost}(G,\mathcal{A})} \right| \right] + \mathbb{E}_{\Omega} \left[ \sup_{\mathcal{S}} \left| \sum_{p \in \Omega} \frac{Y_{G,p,\mathcal{S}} - \mathbb{E}\left[Y_{G,p,\mathcal{S}}\right]}{\mathrm{cost}(G,\mathcal{S}) + \mathrm{cost}(G,\mathcal{A})} \right| \right]$$

$$\leq \mathbb{E}_{\Omega} \left[ \sup_{\mathcal{S}} \left| \frac{\sum_{p \in \Omega} w_p u_p^{G,\mathcal{S}} - \|u^{G,\mathcal{S}}\|_1}{\mathrm{cost}(G,\mathcal{S}) + \mathrm{cost}(G,\mathcal{A})} \right| \right] + \sqrt{2\pi} \cdot \mathbb{E}_{\Omega,g} \left[ \sup_{\mathcal{S}} |X_{G,\mathcal{S}}| \right] \qquad \text{(Lemma D.12)}$$

$$\leq \varepsilon + \sqrt{2\pi}\varepsilon. \qquad\qquad\qquad\qquad\qquad\qquad \text{(Lemmas D.15 and D.13)}$$

Rescaling $\varepsilon$ by a $1 + \sqrt{2\pi}$ factor concludes the proof. $\qquad\qquad\qquad\qquad\qquad\qquad\qquad\qquad\qquad \square$

We, thus, turn our attention to outer groups $G^O$, where the proof is completely identical to the above, with $P^G$ in place of $G$, and Lemmas D.16, D.14 in place of Lemmas D.15, D.13 respectively:

*Proof of Lemma D.11.* We have that

$$\mathbb{E}_\Omega \left[ \sup_{\mathcal{S}} \frac{D_{\mathcal{S}}^\Omega(G)}{\text{cost}(P^G, \mathcal{S}) + \text{cost}(P^G, \mathcal{A})} \right]$$

$$= \mathbb{E}_\Omega \left[ \sup_{\mathcal{S}} \left| \frac{\|v^{G,\mathcal{S}} - u^{G,\mathcal{S}}\|_1 + \|u^{G,\mathcal{S}}\|_1 - \sum_{p\in\Omega} \left( w_p u_p^{G,\mathcal{S}} + w_p(v_p^{G,\mathcal{S}} - u_p^{G,\mathcal{S}}) \right)}{\text{cost}(P^G, \mathcal{S}) + \text{cost}(P^G, \mathcal{A})} \right| \right]$$

$$\leq \mathbb{E}_\Omega \left[ \sup_{\mathcal{S}} \left| \frac{\sum_{p\in\Omega} w_p u_p^{G,\mathcal{S}} - \|u^{G,\mathcal{S}}\|_1}{\text{cost}(P^G, \mathcal{S}) + \text{cost}(P^G, \mathcal{A})} \right| \right] + \mathbb{E}_\Omega \left[ \sup_{\mathcal{S}} \left| \frac{\sum_{p\in\Omega} w_p(v_p^{G,\mathcal{S}} - u_p^{G,\mathcal{S}}) - \|v^{G,\mathcal{S}} - u^{G,\mathcal{S}}\|_1}{\text{cost}(P^G, \mathcal{S}) + \text{cost}(P^G, \mathcal{A})} \right| \right]$$

$$\leq \mathbb{E}_\Omega \left[ \sup_{\mathcal{S}} \left| \frac{\sum_{p\in\Omega} w_p u_p^{G,\mathcal{S}} - \|u^{G,\mathcal{S}}\|_1}{\text{cost}(P^G, \mathcal{S}) + \text{cost}(P^G, \mathcal{A})} \right| \right] + \mathbb{E}_\Omega \left[ \sup_{\mathcal{S}} \left| \sum_{p\in\Omega} \frac{Y_{G,p,\mathcal{S}} - \mathbb{E}\left[Y_{G,p,\mathcal{S}}\right]}{\text{cost}(P^G, \mathcal{S}) + \text{cost}(P^G, \mathcal{A})} \right| \right]$$

$$\leq \mathbb{E}_\Omega \left[ \sup_{\mathcal{S}} \left| \frac{\sum_{p\in\Omega} w_p u_p^{G,\mathcal{S}} - \|u^{G,\mathcal{S}}\|_1}{\text{cost}(P^G, \mathcal{S}) + \text{cost}(P^G, \mathcal{A})} \right| \right] + \sqrt{2\pi} \cdot \mathbb{E}_{\Omega,g} \left[ \sup_{\mathcal{S}} |X_{G,\mathcal{S}}| \right] \qquad\qquad \text{(Lemma D.12)}$$

$$\leq \varepsilon + \sqrt{2\pi}\varepsilon. \qquad\qquad\qquad\qquad\qquad\qquad\qquad\qquad \text{(Lemmas D.16 and D.14)}$$

Rescaling $\varepsilon$ by a $1 + \sqrt{2\pi}$ factor concludes the proof. $\qquad\qquad\qquad\qquad\qquad\qquad\qquad\qquad\qquad \square$

### D.7. Estimating $\|v^{G,\mathcal{S}} - u^{G,\mathcal{S}}\|_1$: A Gaussian Process to Estimate Groups Costs

In this section, we set out to show Lemmas D.13 and D.14. For both of them, i.e., whether $G \in G^M$ or $G \in G^O$, we define the following random variables starting from the definition of $X_{G,\mathcal{S}}$:

$$X_{G,\mathcal{S},0} = \frac{\sum_{p\in\Omega} g_p \cdot w_p v_p^{\mathcal{S},1}}{\text{cost}(G, \mathcal{S}) + \text{cost}(G, \mathcal{A})} \quad \text{and} \quad X_{G,\mathcal{S},h} = \frac{\sum_{p\in\Omega} g_p \cdot w_p(v_p^{\mathcal{S},h+1} - v_p^{\mathcal{S},h})}{\text{cost}(G, \mathcal{S}) + \text{cost}(G, \mathcal{A})}, \quad \forall G \in G^M$$

$$X_{G,\mathcal{S},0} = \frac{\sum_{p\in\Omega} g_p \cdot w_p v_p^{\mathcal{S},1}}{\text{cost}(P^G, \mathcal{S}) + \text{cost}(P^G, \mathcal{A})} \quad \text{and} \quad X_{G,\mathcal{S},h} = \frac{\sum_{p\in\Omega} g_p \cdot w_p(v_p^{\mathcal{S},h+1} - v_p^{\mathcal{S},h})}{\text{cost}(P^G, \mathcal{S}) + \text{cost}(P^G, \mathcal{A})}, \quad \forall G \in G^O.$$

We may observe that

$$\mathbb{E}_{\Omega,g} \left[ \sup_{\mathcal{S}} |X_{G,\mathcal{S}}| \right] \leq \sum_{h=0}^{\infty} \mathbb{E}_{\Omega,g} \left[ \sup_{\mathcal{S}} |X_{G,\mathcal{S},h}| \right]. \qquad\qquad\qquad\qquad (20)$$

Therefore, we need to have a handle on the number of distinct $v^{\mathcal{S},h}$'s vectors as well as on the variance of each single $X_{G,\mathcal{S},h}$ in order to be able to bound the above quantity. Specifically, we will show the lemmas through the following considerations:

(i). $X_{G,\mathcal{S},h}$ is a Gaussian random variable with variance equal to

$$\sum_{p\in\Omega} \frac{w_p(v_p^{\mathcal{S},h+1} - v_p^{\mathcal{S},h})}{\text{cost}(G, \mathcal{S}) + \text{cost}(G, \mathcal{A})} \leq \sigma \quad \text{or} \quad \sum_{p\in\Omega} \frac{w_p(v_p^{\mathcal{S},h+1} - v_p^{\mathcal{S},h})}{\text{cost}(P^G, \mathcal{S}) + \text{cost}(P^G, \mathcal{A})} \leq \sigma,$$

depending on whether we are dealing with main or outer groups, and for some $\sigma$ we will compute;

(ii). There are at most $|\mathcal{N}_{h-1}| \cdot |\mathcal{N}_h|$ many distinct $v^{\mathcal{S},h}$'s vectors that cover $\mathcal{S}$;

(iii). It is a standard fact that

$$\mathbb{E}\left[\max_{i\in[n]}|g_i|\right] \le 2\sigma\sqrt{\ln n},$$

for $n$ Gaussian random variables $g_i$, whose variance is at most $\sigma$ (see Fact D.2).

The above proof skeleton allows us to show the desired bounds for both main and outer groups. The former, however, requires a more careful analysis, while the latter is much simpler. We formalize the arguments for the former next and for the latter at the end of this section. The reason why the two analyses have to be treated separately is that, in the case of outer groups $G^O$, the variance bound $\sigma$ we are able to obtain is much better than the one achievable for main groups $G^M$ in the worst case. Even more importantly, the worst-case variance bound for $G^M$ is not enough by itself to guarantee the promised inequality of Lemma D.13. We, thus, need to split the proof for main groups into two cases: One where the estimator is good for the size of every cluster in $\mathcal{A}$ with high probability. The other where the estimator is far from being accurate, but the probability of this happening is vanishingly small.

### D.7.1. CHAINING FOR MAIN GROUPS $G^M$

As mentioned, we next show that $|C \cap G|$ is well approximated for all clusters $C$, provided enough points are sampled, with high probability. To this end, define

$$\mathcal{E}_G^M = \left\{\forall i \in [k], \sum_{p\in C_i\cap G\cap\Omega} w_p = \sum_{p\in C_i\cap G\cap\Omega} \frac{\mathrm{cost}(G,\mathcal{A})}{\omega\cdot\mathrm{cost}(p,\mathcal{A})} \in (1\pm\varepsilon)\cdot|C_i\cap G|\right\}.$$

**Lemma D.17.** *Let $G \in G^M$. Then,*

$$\mathbb{P}\left[\mathcal{E}_G^M\right] \ge 1 - k\cdot\exp\left(-\frac{\varepsilon^2\omega}{9k}\right).$$

*Proof.* First note that for every cluster $C$ in $\mathcal{A}$, $\mathbb{E}\left[\sum_{p\in C_i\cap G\cap\Omega} w_p\right] = |C\cap G|$. Hence, we need to show that these estimators concentrate around their expectation simultaneously. Consider the $j$-th sampled point $p_j$ in coreset $\Omega$, and let $w_{p_j,C} = w_p\cdot\mathbb{1}_{\{p_j\in C\cap G\}}$. Then,

$$\mathbb{V}\left[w_{p_j,C}\right] \le \mathbb{E}\left[w_{p_j,C}^2\right] = \sum_{p\in C\cap G} w_p^2\cdot\mathbb{P}\left[p\in\Omega\right] = \sum_{p\in C\cap G} \frac{\mathrm{cost}(G,\mathcal{A})}{\omega\cdot\mathrm{cost}(p,\mathcal{A})}$$

$$\le \sum_{p\in C\cap G} \frac{2k\cdot\mathrm{cost}(C\cap G,\mathcal{A})}{\omega\cdot\mathrm{cost}(p,\mathcal{A})} \le \frac{4k\cdot|C\cap G|^2}{\omega^2},$$

where the second line of inequalities follows from Claim D.1. Similarly, Claim D.1 also implies

$$w_{p_j,C} = \frac{\mathrm{cost}(G,\mathcal{A})}{\omega\cdot\mathrm{cost}(p_j,\mathcal{A})} \le \frac{4k\cdot|C\cap G|}{\omega}.$$

Thus, using Bernstein's Inequality (Fact D.1), we obtain

$$\mathbb{P}\left[\left|\sum_{p\in C\cap\Omega} w_p - |C\cap G|\right| > \varepsilon\cdot|C\cap G|\right] \le \exp\left(-\frac{\varepsilon^2\cdot|C\cap G|^2}{2\omega\cdot\frac{4k\cdot|C\cap G|^2}{\omega^2} + \frac{2}{3}\cdot\frac{4k\cdot|C\cap G|}{\omega}\cdot\varepsilon\cdot|C\cap G|}\right)$$

$$\le \exp\left(-\frac{\varepsilon^2\omega}{9k}\right).$$

A union bound over all clusters in $\mathcal{A}$ yields the lemma. $\qquad\square$

**Lemma D.18.** *Let $G \in G^M$ and $\mathcal{S}$ be an arbitrary fixed solution. Then, $X_{G,\mathcal{S},h}$ is a Gaussian random variable with mean 0 and variance*

$$\sum_{p\in\Omega}\left(\frac{w_p(v_p^{\mathcal{S},h+1} - v_p^{\mathcal{S},h})}{\mathrm{cost}(G,\mathcal{S}) + \mathrm{cost}(G,\mathcal{A})}\right)^2 \le \frac{2^{-2h+2}\cdot 2^{4z\log z}}{\omega}\cdot\varepsilon^{-2z}.$$

*Moreover, conditioned on event $\mathcal{E}_G^M$, $X_{G,\mathcal{S},h}$'s variance is*

$$\sum_{p \in \Omega} \left( \frac{w_p(v_p^{\mathcal{S},h+1} - v_p^{\mathcal{S},h})}{\text{cost}(G,\mathcal{S}) + \text{cost}(G,\mathcal{A})} \right)^2 \leq \frac{2^{-2h+2} \cdot 2^{4z \log z}}{\omega} \cdot \min(\varepsilon^{-z}, k).$$

Before proving the above lemma, let us show how it implies, together with Lemma D.17, Lemma D.13.

*Proof of Lemma D.13.* Let us recall Equation (20). Consequently, we only need to focus on terms of the form $\mathbb{E}_{\Omega,g}[\sup_{\mathcal{S}} |X_{G,\mathcal{S},h}|]$, which, by the Law of Total Expectation, can be written as

$$\mathbb{E}_{\Omega,g} \left[ \sup_{\mathcal{S}} |X_{G,\mathcal{S},h}| \mid \mathcal{E}_G^M \right] \cdot \mathbb{P}\left[ \mathcal{E}_G^M \right] + \mathbb{E}_{\Omega,g} \left[ \sup_{\mathcal{S}} |X_{G,\mathcal{S},h}| \mid \bar{\mathcal{E}}_G^M \right] \cdot \mathbb{P}\left[ \bar{\mathcal{E}}_G^M \right]. \tag{21}$$

**Conditioning on $\mathcal{E}_G^M$.** First, we note that $\mathbb{P}\left[ \mathcal{E}_G^M \right] \leq 1$. Moreover, by Lemma 2 of (Cohen-Addad et al., 2021), it is enough to consider a chain up to $t \leq \log 2\varepsilon^{-1}$, as the entire solution is captured by an $(\varepsilon, k, z)$-approximate centroid set. Using Fact D.2 and the fact that there are at most $|\mathcal{N}_{h-1}| \cdot |\mathcal{N}_h|$ many distinct $X_{G,\mathcal{S},h}$'s, we have

$$\mathbb{E}_{\Omega,g} \left[ \sup_{\mathcal{S}} |X_{G,\mathcal{S},h}| \mid \mathcal{E}_G^M \right] \leq 2\sqrt{\mathbb{V}\left[ X_{G,\mathcal{S},h} \mid \mathcal{E}_G^M \right]} \cdot \sqrt{2\log(|\mathcal{N}_{h-1}| \cdot |\mathcal{N}_h|)}$$

$$\leq 2\sqrt{\frac{2^{-2h+2} \cdot 2^{4z \log z}}{\omega} \cdot \min(\varepsilon^{-z}, k) \cdot 4\log|\mathcal{N}_h|} \tag{Lemma D.18}$$

$$\leq 2\sqrt{\frac{2^{(c-2)h} \cdot 2^{4(z \log z + 1)} \cdot \min(\varepsilon^{-z}, k) \cdot Ukz^2 \log(2^h \varepsilon^{-1})}{\omega \log^{-1} \omega}}. \tag{Assumption}$$

The last inequality, in particular, is obtained since, for any point set of size $\omega$,

$$|\mathcal{N}_h| \leq \exp\left( Ukz^2 \log \omega \cdot 2^{ch} \log\left( 2^h \varepsilon^{-1} \right) \right).$$

Note that when $c \leq 2$, the numerator above is maximized at $h = 0$, and otherwise at $h = t = \log 2\varepsilon^{-1}$. Hence, as long as

$$\frac{\omega}{\log \omega} \geq 2^{4(z \log z + 2)} \cdot Ukz^2 \cdot (\varepsilon^{-c} + \varepsilon^{-2}) \log^3 \varepsilon^{-1} \cdot \min(\varepsilon^{-z}, k),$$

we get

$$\mathbb{E}_{\Omega,g} \left[ \sup_{\mathcal{S}} |X_{G,\mathcal{S},h}| \mid \mathcal{E}_G^M \right] \leq \frac{2\varepsilon}{\log \varepsilon^{-1}}.$$

Using that, for all $a, b \geq 1$ such that $a \geq 2b \log b$, then $a \geq b \log a$, we would need $\omega \geq 2\omega' \cdot \log \omega'$, where $\omega' = 2^{4(z \log z + 2)} \cdot Ukz^2 \cdot (\varepsilon^{-c} + \varepsilon^{-2}) \cdot \min(\varepsilon^{-z}, k) \cdot \log^3 \varepsilon^{-1}$. Hence, it is sufficient to choose

$$\omega = 2^{\lambda z \log z} \cdot Ukz^2 \cdot (\varepsilon^{-c} + \varepsilon^{-2}) \cdot \min(\varepsilon^{-z}, k) \cdot \log^4 \varepsilon^{-1} \cdot \log(Ukz),$$

for some (large enough) constant $\lambda > 4$. As we are considering a chain up until $\log 2\varepsilon^{-1}$, we get

$$\sum_{h=0}^{\log 2\varepsilon^{-1}} \mathbb{E}_{\Omega,g} \left[ \sup_{\mathcal{S}} |X_{G,\mathcal{S},h}| \mid \mathcal{E}_G^M \right] \leq 4\varepsilon.$$

**Conditioning on $\bar{\mathcal{E}}_G^M$.** Lemma D.17 and a union bound over all $z^2 \log^2(z/\varepsilon)$ many groups gives us that

$$\mathbb{P}\left[ \bar{\mathcal{E}}_G^M \right] \leq k \cdot z^2 \log^2(z\varepsilon^{-1}) \cdot \exp\left( -\frac{\varepsilon^2 \omega}{9k} \right) \leq \varepsilon^{2z},$$

where the last inequality follows by an appropriate choice of $\lambda$ in the $\omega$ expression. We will use the worse of the two variance bounds given in Lemma D.18 for the term $\mathbb{E}_{\Omega,g}[\sup_{\mathcal{S}} |X_{G,\mathcal{S},h}| \mid \bar{\mathcal{E}}_G^M]$. In particular, we have that

$$\mathbb{E}_{\Omega,g} \left[ \sup_{\mathcal{S}} |X_{G,\mathcal{S},h}| \mid \bar{\mathcal{E}}_G^M \right] \leq 2\varepsilon^{-z} \frac{\varepsilon}{\log \varepsilon^{-1}}.$$

Thus,

$$\sum_{h=0}^{\log 2\varepsilon^{-1}} \mathbb{E}_{\Omega,g}\left[\sup_{\mathcal{S}} |X_{G,\mathcal{S},h}| \mid \bar{\mathcal{E}}_G^M\right] \leq 4\varepsilon^{-z+1},$$

analogously to the above derivation. Summing the obtained upper bound on the terms of Equation 21, we get

$$\sum_{h=0}^{\infty} \mathbb{E}_{\Omega,g}\left[\sup_{\mathcal{S}} |X_{G,\mathcal{S},h}|\right] \leq 4(\varepsilon + \varepsilon^{-z+1} \cdot \varepsilon^{2z}) \leq 8\varepsilon.$$

We rescale $\varepsilon$ by a factor 8 to conclude. $\qquad\square$

We are now ready to prove Lemma D.18.

*Proof of Lemma D.18.* We start by observing that $X_{G,\mathcal{S},h}$ is a sum of standard normal random variables multiplied by other (scalar) terms, succinctly $\sum_p a_p \cdot g_p$. As such, it is itself a centered Gaussian random variable with variance $\sum_p a_p^2$. Namely,

$$
\begin{aligned}
\mathbb{V}\left[X_{G,\mathcal{S},h}\right] &= \sum_{p\in\Omega} \left(\frac{w_p(v_p^{\mathcal{S},h+1} - v_p^{\mathcal{S},h})}{\mathrm{cost}(G,\mathcal{S}) + \mathrm{cost}(G,\mathcal{A})}\right)^2 \\
&= \sum_{p\in\Omega} \left(\frac{w_p(v_p^{\mathcal{S},h+1} - \mathrm{cost}(p,\mathcal{S}) + \mathrm{cost}(p,\mathcal{S}) - v_p^{\mathcal{S},h})}{\mathrm{cost}(G,\mathcal{S}) + \mathrm{cost}(G,\mathcal{A})}\right)^2 \\
&\leq \sum_{p\in\Omega} \left(\frac{w_p \cdot 3 \cdot 2^{-h-1} \cdot \mathrm{cost}(p,\mathcal{S})}{\mathrm{cost}(G,\mathcal{S}) + \mathrm{cost}(G,\mathcal{A})}\right)^2 \\
&= 9 \cdot 2^{-2h-2} \cdot \sum_{p\in\Omega} \left(\frac{\frac{\mathrm{cost}(G,\mathcal{A})}{\omega \cdot \mathrm{cost}(p,\mathcal{A})} \cdot \mathrm{cost}(p,\mathcal{S})}{\mathrm{cost}(G,\mathcal{S}) + \mathrm{cost}(G,\mathcal{A})}\right)^2,
\end{aligned}
\tag{22}
$$

where the second inequality follows because $v_p^{\mathcal{S},h} \in (1 \pm 2^{-h}) \cdot \mathrm{cost}(p,\mathcal{S})$ since the vector $v^{\mathcal{S},h} \in \mathcal{N}_h$, and similarly for $v^{\mathcal{S},h+1} \in \mathcal{N}_{h+1}$. The first bound on variance directly follows since, by assumption, $p \in C \notin H_{G,\mathcal{S}}$, that is $\frac{\mathrm{cost}(p,\mathcal{S})}{\mathrm{cost}(p,\mathcal{A})} \leq \left(\frac{4z}{\varepsilon}\right)^z$.

For the second bound, we make use of the event $\mathcal{E}_G^M$, so that we have a good estimate of cluster size for all clusters $C$. Consider $q \in \arg\min_{p\in C} \mathrm{cost}(p,\mathcal{S})$ to cheapest point in the cluster. Then, by Fact D.3, for any point $p$ and any solution $\mathcal{S}$, it holds that

$$\mathrm{cost}(p,\mathcal{S}) \leq (d(q,\mathcal{S}) + d(p,q))^z \leq \frac{2^z \cdot \mathrm{cost}(C \cap G, \mathcal{S}) + 4^z \cdot \mathrm{cost}(C \cap G, \mathcal{A})}{|C \cap G|}. \tag{23}$$

We now bound the expression in Equation 22 from above in the $\varepsilon^z < k$ case, and vice versa. In the first case, we have

$$
\begin{aligned}
&9 \cdot 2^{-2h-2} \cdot \sum_{p\in\Omega} \left(\frac{\frac{\mathrm{cost}(G,\mathcal{A})}{\omega \cdot \mathrm{cost}(p,\mathcal{A})} \cdot \mathrm{cost}(p,\mathcal{S})}{\mathrm{cost}(G,\mathcal{S}) + \mathrm{cost}(G,\mathcal{A})}\right)^2 \\
&\leq \frac{9 \cdot 2^{-2h-2} \cdot \left(\frac{4z}{\varepsilon}\right)^z \cdot \mathrm{cost}(G,\mathcal{A})}{\omega \cdot (\mathrm{cost}(G,\mathcal{S}) + \mathrm{cost}(G,\mathcal{A}))^2} \cdot \sum_{p\in\Omega} \frac{\mathrm{cost}(G,\mathcal{A})}{\omega \cdot \mathrm{cost}(p,\mathcal{A})} \cdot \mathrm{cost}(p,\mathcal{S}) && (p \in C \notin H_{G,\mathcal{S}}) \\
&\leq \frac{9 \cdot 2^{-2h-2} \cdot \left(\frac{4z}{\varepsilon}\right)^z \cdot \mathrm{cost}(G,\mathcal{A})}{\omega \cdot (\mathrm{cost}(G,\mathcal{S}) + \mathrm{cost}(G,\mathcal{A}))^2} \cdot \sum_C \left(\frac{2^z \cdot \mathrm{cost}(C \cap G, \mathcal{S}) + 4^z \cdot \mathrm{cost}(C \cap G, \mathcal{A})}{|C \cap G|}\right. \\
&\qquad\qquad\qquad\qquad\qquad\qquad\qquad\qquad\qquad \left. \cdot \sum_{p\in C\cap G\cap\Omega} \frac{\mathrm{cost}(G,\mathcal{A})}{\omega \cdot \mathrm{cost}(p,\mathcal{A})}\right) && \text{(Equation 23)}
\end{aligned}
$$

$$\leq \frac{9 \cdot 2^{-2h-2} \cdot \left(\frac{4z}{\varepsilon}\right)^z \cdot \mathrm{cost}(G, \mathcal{A})}{\omega \cdot (\mathrm{cost}(G, \mathcal{S}) + \mathrm{cost}(G, \mathcal{A}))^2} \cdot (1 + \varepsilon) \cdot \sum_C \left(2^z \cdot \mathrm{cost}(C \cap G, \mathcal{S}) + 4^z \cdot \mathrm{cost}(C \cap G, \mathcal{A})\right) \qquad (\mathcal{E}_G^M \text{ holds})$$

$$\leq \frac{2^{-2h+2} \cdot \left(\frac{4z}{\varepsilon}\right)^z \cdot \mathrm{cost}(G, \mathcal{A})}{\omega \cdot (\mathrm{cost}(G, \mathcal{S}) + \mathrm{cost}(G, \mathcal{A}))^2} \cdot 4^z \cdot (\mathrm{cost}(G, \mathcal{S}) + \mathrm{cost}(G, \mathcal{A})) \qquad (9(1 + \varepsilon) \leq 16)$$

$$\leq \frac{2^{-2h+2} \cdot \left(\frac{4z}{\varepsilon}\right)^z \cdot 4^z}{\omega}. \qquad (\mathrm{cost}(G, \mathcal{A})) \leq \mathrm{cost}(G, \mathcal{S}) + \mathrm{cost}(G, \mathcal{A}))$$

In the second case, we seek to obtain a bound depending on $k$. By Claim D.1, we know that

$$\mathrm{cost}(G, \mathcal{A}) \leq 2k \cdot \mathrm{cost}(C \cap G, \mathcal{A}) \leq 4k \cdot |C \cap G| \cdot \mathrm{cost}(p, \mathcal{A}),$$

for all points $p \in C \cap G$. Thus,

$$9 \cdot 2^{-2h-2} \cdot \sum_{p \in \Omega} \left(\frac{\frac{\mathrm{cost}(G, \mathcal{A})}{\omega \cdot \mathrm{cost}(p, \mathcal{A})} \cdot \mathrm{cost}(p, \mathcal{S})}{\mathrm{cost}(G, \mathcal{S}) + \mathrm{cost}(G, \mathcal{A})}\right)^2$$

$$\leq \frac{9 \cdot 2^{-2h-2}}{\omega \cdot (\mathrm{cost}(G, \mathcal{S}) + \mathrm{cost}(G, \mathcal{A}))^2} \cdot \sum_C \sum_{p \in C \cap G \cap \Omega} \frac{4k \cdot |C \cap G| \cdot \mathrm{cost}(p, \mathcal{A})}{\omega \cdot \mathrm{cost}(p, \mathcal{A})} \cdot \mathrm{cost}^2(p, \mathcal{S}) \qquad (\text{Claim D.1})$$

$$\leq \frac{9 \cdot 2^{-2h} \cdot k}{\omega \cdot (\mathrm{cost}(G, \mathcal{S}) + \mathrm{cost}(G, \mathcal{A}))^2} \cdot \sum_C \left(|C \cap G| \cdot \left(\frac{2^z \cdot \mathrm{cost}(C \cap G, \mathcal{S}) + 4^z \cdot \mathrm{cost}(C \cap G, \mathcal{A})}{|C \cap G|}\right)^2 \right.$$
$$\left. \cdot \sum_{p \in C \cap G \cap \Omega} \frac{\mathrm{cost}(G, \mathcal{A})}{\omega \cdot \mathrm{cost}(p, \mathcal{A})}\right) \qquad (\text{Equation 23})$$

$$\leq \frac{9 \cdot 2^{-2h} \cdot k}{\omega \cdot (\mathrm{cost}(G, \mathcal{S}) + \mathrm{cost}(G, \mathcal{A}))^2} \cdot (1 + \varepsilon) \cdot \sum_C \left(2^z \cdot \mathrm{cost}(C \cap G, \mathcal{S}) + 4^z \cdot \mathrm{cost}(C \cap G, \mathcal{A})\right)^2 \qquad (\mathcal{E}_G^M \text{ holds})$$

$$\leq \frac{2^{-2h+4} \cdot k}{\omega \cdot (\mathrm{cost}(G, \mathcal{S}) + \mathrm{cost}(G, \mathcal{A}))^2} \cdot \left(\sum_C 2^z \cdot \mathrm{cost}(C \cap G, \mathcal{S}) + 4^z \cdot \mathrm{cost}(C \cap G, \mathcal{A})\right)^2 \qquad (9(1 + \varepsilon) \leq 16)$$

$$\leq \frac{2^{-2h+2} \cdot k \cdot 16^z}{\omega}. \qquad (\mathrm{cost}(G, \mathcal{A})) \leq \mathrm{cost}(G, \mathcal{S}) + \mathrm{cost}(G, \mathcal{A}))$$

This concludes the proof. $\qquad \square$

### D.7.2. CHAINING FOR OUTER GROUPS $G^O$

We first state and prove an analog of the variance bound in Lemma D.18, but only in the worst-case, as this is sufficient. We, then, almost identically derive a proof of Lemma D.14, as we did for Lemma D.13.

**Lemma D.19.** *Let $G \in G^O$ and $\mathcal{S}$ be an arbitrary fixed solution. Then, $X_{G,\mathcal{S},h}$ is a Gaussian random variable with mean $0$ and variance*

$$\sum_{p \in \Omega} \left(\frac{w_p(v_p^{\mathcal{S},h+1} - v_p^{\mathcal{S},h})}{\mathrm{cost}(P^G, \mathcal{S}) + \mathrm{cost}(P^G, \mathcal{A})}\right)^2 \leq \frac{2^{-2h+2} \cdot 16^z}{\omega}.$$

*Proof.* Akin to Lemma D.18, we start by observing that $X_{G,\mathcal{S},h}$ is a sum of standard normal random variables multiplied by other (scalar) terms, succinctly $\sum_p a_p \cdot g_p$. As such, it is itself a centered Gaussian random variable with variance $\sum_p a_p^2$. Namely,

$$\mathbb{V}\left[X_{G,\mathcal{S},h}\right] = \sum_{p \in \Omega} \left(\frac{w_p(v_p^{\mathcal{S},h+1} - v_p^{\mathcal{S},h})}{\mathrm{cost}(P^G, \mathcal{S}) + \mathrm{cost}(P^G, \mathcal{A})}\right)^2$$

$$= \sum_{p \in \Omega} \left(\frac{w_p(v_p^{\mathcal{S},h+1} - \mathrm{cost}(p, \mathcal{S}) + \mathrm{cost}(p, \mathcal{S}) - v_p^{\mathcal{S},h})}{\mathrm{cost}(P^G, \mathcal{S}) + \mathrm{cost}(P^G, \mathcal{A})}\right)^2$$

$$\leq \sum_{p \in \Omega} \left( \frac{w_p \cdot 3 \cdot 2^{-h-1} \cdot \mathrm{cost}(p, \mathcal{S})}{\mathrm{cost}(P^G, \mathcal{S}) + \mathrm{cost}(P^G, \mathcal{A})} \right)^2$$

$$= 9 \cdot 2^{-2h-2} \cdot \sum_{p \in \Omega} \left( \frac{\frac{\mathrm{cost}(G, \mathcal{A})}{\omega \cdot \mathrm{cost}(p, \mathcal{A})} \cdot \mathrm{cost}(p, \mathcal{S})}{\mathrm{cost}(P^G, \mathcal{S}) + \mathrm{cost}(P^G, \mathcal{A})} \right)^2$$

$$\leq \frac{2^{-2h+2} \cdot 16^z}{\omega},$$

where the first inequality follows because $v_p^{\mathcal{S}, h} \in (1 \pm 2^{-h}) \cdot \mathrm{cost}(p, \mathcal{S})$ since the vector $v^{\mathcal{S}, h} \in \mathcal{N}_h$, and similarly for $v^{\mathcal{S}, h+1} \in \mathcal{N}_{h+1}$. The second inequality follows since, for all $p \in C \cap G$, we must have $\mathrm{cost}(p, \mathcal{S}) \leq 4^z \cdot \mathrm{cost}(p, \mathcal{A})$, since $C \notin F_{G, \mathcal{S}}$. $\qquad\square$

We are left to show how the above implies Lemma D.14.

*Proof of Lemma D.14.* Akin to Lemma D.13, we use Fact D.2 and the fact that there are at most $|\mathcal{N}_{h-1}| \cdot |\mathcal{N}_h|$ many distinct $X_{G, \mathcal{S}, h}$'s. Indentical calculations with respect to the ones in Lemma D.13, with a much smaller variance provided by Lemma D.19, yields

$$\sum_{h=0}^{\infty} \mathbb{E}_{\Omega, g} \left[ \sup_{\mathcal{S}} |X_{G, \mathcal{S}, h}| \right] \leq O(\varepsilon),$$

which concludes the proof by a rescaling of $\varepsilon$. $\qquad\square$

## D.8. Estimating $\|u^{G, \mathcal{S}}\|_1$: Huge and Far Clusters Estimates

The goal of this section is to show Lemmas D.15 and D.16. We begin with the former as its proof is shorter and, yet, conveys the main ideas that will be applied in the proof of the latter.

### D.8.1. ANALYSIS FOR HUGE CLUSTERS

Let us recall that we consider points $p \in C \cap G$ where $C \in H_{G, \mathcal{S}}$, i.e., their cost in the current solution $\mathcal{S}$ is much larger than the cost they have in the approximate starting solution $\mathcal{A}$, as per Definition D.2. By virtue of the point being so expensive, we have that its cost is roughly identical to the average cost of the entire cluster induced by $\mathcal{S}$ on $G$, so long as all induced cluster sizes are well-estimated.

**Lemma D.20.** *Let $G \in G^M$ and let $\varepsilon < \frac{1}{2}$. Then, for any solution $\mathcal{S}$ and any point $p \in C \cap G$ such that $C \in H_{G, \mathcal{S}}$, it holds that*

$$\mathbb{E}_{\Omega} \left[ \left| \left| \sum_{p \in \Omega} w_p u_p^{G, \mathcal{S}} - \mathrm{cost}(C \cap G, \mathcal{S}) \right| \right| \mathcal{E}_G^M \right] \leq 5\varepsilon \cdot \mathrm{cost}(C \cap G, \mathcal{S}).$$

*Proof.* Let us consider two points $p, p' \in C \cap G$ such that $C \in H_{G, \mathcal{S}}$. Since they belong to the same group, it must hold that

$$\frac{\mathrm{cost}(p', \mathcal{S})}{2} \leq \mathrm{cost}(p, \mathcal{S}) \leq 2\mathrm{cost}(p', \mathcal{S}).$$

Then, by applying Fact D.3 with $\vartheta = 1$, we get that

$$\mathrm{cost}(p, p') \leq 2^{z-1} \cdot (\mathrm{cost}(p, \mathcal{A}) + \mathrm{cost}(p', \mathcal{A})) \leq 2^{z+1} \cdot \mathrm{cost}(p', \mathcal{A}).$$

Now, applying Fact D.3 again with $\vartheta = \varepsilon$, we obtain

$$\mathrm{cost}(p, \mathcal{S}) \leq (1 + \varepsilon) \cdot \mathrm{cost}(p', \mathcal{S}) + \left( \frac{z + \varepsilon}{z} \right)^{z-1} \cdot \mathrm{cost}(p, p')$$

$$\leq (1 + \varepsilon) \cdot \mathrm{cost}(p', \mathcal{S}) + \left( \frac{z + \varepsilon}{z} \right)^{z-1} \cdot 2^{z+1} \cdot \mathrm{cost}(p', \mathcal{A})$$

$$\leq (1+\varepsilon) \cdot \mathrm{cost}(p', \mathcal{S}) + \left(\frac{z+\varepsilon}{z}\right)^{z-1} \cdot 2^{z+1} \cdot \left(\frac{\varepsilon}{4z}\right)^z \cdot \mathrm{cost}(p', \mathcal{S}) \qquad (p' \in C \cap G \text{ where } C \in H_{G,\mathcal{S}})$$

$$\leq (1+2\varepsilon) \cdot \mathrm{cost}(p', \mathcal{S}).$$

Similarly,

$$\mathrm{cost}(p', \mathcal{S}) \leq (1+\varepsilon) \cdot \mathrm{cost}(p, \mathcal{S}) + \left(\frac{z+\varepsilon}{z}\right)^{z-1} \cdot \mathrm{cost}(p, p')$$

$$\leq (1+\varepsilon) \cdot \mathrm{cost}(p, \mathcal{S}) + \left(\frac{z+\varepsilon}{z}\right)^{z-1} \cdot 2^{z+1} \cdot \mathrm{cost}(p', \mathcal{A})$$

$$\leq (1+\varepsilon) \cdot \mathrm{cost}(p, \mathcal{S}) + \left(\frac{z+\varepsilon}{z}\right)^{z-1} \cdot 2^{z+1} \cdot \left(\frac{\varepsilon}{4z}\right)^z \cdot \mathrm{cost}(p', \mathcal{S}) \qquad (p' \in C \cap G \text{ where } C \in H_{G,\mathcal{S}})$$

$$\leq (1+\varepsilon) \cdot \mathrm{cost}(p, \mathcal{S}) + \varepsilon \cdot \mathrm{cost}(p', \mathcal{S}).$$

Hence, $\mathrm{cost}(p, \mathcal{S}) \in (1 \pm 2\varepsilon) \cdot \mathrm{cost}(p', \mathcal{S})$, since $\frac{1-\varepsilon}{1+\varepsilon} \geq 1 - 2\varepsilon$. Therefore, under event $\mathcal{E}_G^M$, we can upper bound the estimated induced cluster cost as follows:

$$\sum_{p \in C \cap G \cap \Omega} w_p \cdot \mathrm{cost}(p, \mathcal{S}) \leq (1+2\varepsilon) \cdot \sum_{p \in C \cap G \cap \Omega} w_p \cdot \mathrm{cost}(p', \mathcal{S})$$

$$\leq (1+\varepsilon)(1+2\varepsilon) \cdot |C \cap G| \cdot \mathrm{cost}(p', \mathcal{S}) \qquad (\mathcal{E}_G^M \text{ holds})$$

$$\leq \frac{(1+\varepsilon)(1+2\varepsilon)}{(1-2\varepsilon)} \cdot \mathrm{cost}(C \cap G, \mathcal{S}),$$

where the second inequality holds for all clusters $C$ simultaneously since event $\mathcal{E}_G^M$ holds. Analogously, we have the following lower bound:

$$\sum_{p \in C \cap G \cap \Omega} w_p \cdot \mathrm{cost}(p, \mathcal{S}) \geq (1-2\varepsilon) \cdot \sum_{p \in C \cap G \cap \Omega} w_p \cdot \mathrm{cost}(p', \mathcal{S})$$

$$\geq (1-\varepsilon)(1-2\varepsilon) \cdot |C \cap G| \cdot \mathrm{cost}(p', \mathcal{S}) \qquad (\mathcal{E}_G^M \text{ holds})$$

$$\geq \frac{(1-\varepsilon)(1-2\varepsilon)}{(1+2\varepsilon)} \cdot \mathrm{cost}(C \cap G, \mathcal{S}).$$

The lemma follows by recognizing that, for $\varepsilon < 1/2$, $\frac{(1+\varepsilon)(1+2\varepsilon)}{(1-2\varepsilon)} \leq 1 = 5\varepsilon$ and $\frac{(1-\varepsilon)(1-2\varepsilon)}{(1+2\varepsilon)} \geq 1 - 5\varepsilon$. □

We proceed with the proof of Lemma D.15.

*Proof of Lemma D.15.* By Law of Total Expectation, we write

$$\mathbb{E}_\Omega \left[ \sup_{\mathcal{S}} \left| \frac{\sum_{p \in \Omega} w_p u_p^{G,\mathcal{S}} - \|u^{G,\mathcal{S}}\|_1}{\mathrm{cost}(G, \mathcal{S}) + \mathrm{cost}(G, \mathcal{A})} \right| \right] = \mathbb{E}_\Omega \left[ \sup_{\mathcal{S}} \left| \frac{\sum_{p \in \Omega} w_p u_p^{G,\mathcal{S}} - \|u^{G,\mathcal{S}}\|_1}{\mathrm{cost}(G, \mathcal{S}) + \mathrm{cost}(G, \mathcal{A})} \right| \, \middle| \, \mathcal{E}_G^M \right] \cdot \mathbb{P}\left[\mathcal{E}_G^M\right] \tag{24}$$

$$+ \mathbb{E}_\Omega \left[ \sup_{\mathcal{S}} \left| \frac{\sum_{p \in \Omega} w_p u_p^{G,\mathcal{S}} - \|u^{G,\mathcal{S}}\|_1}{\mathrm{cost}(G, \mathcal{S}) + \mathrm{cost}(G, \mathcal{A})} \right| \, \middle| \, \bar{\mathcal{E}}_G^M \right] \cdot \mathbb{P}\left[\bar{\mathcal{E}}_G^M\right]. \tag{25}$$

Trivially, $\mathbb{P}\left[\mathcal{E}_G^M\right] \leq 1$ and, conditioned on $\mathcal{E}_G$,

$$\sum_{C \in H_{G,\mathcal{S}}} \sum_{p \in C \cap G \cap \Omega} w_p u_p^{G,\mathcal{S}} \in (1 \pm 5\varepsilon) \cdot \sum_{C \in H_{G,\mathcal{S}}} \sum_{p \in C \cap G \cap \Omega} \mathrm{cost}(p, \mathcal{S}),$$

by Lemma D.20. Since $\|u^{G,\mathcal{S}}\|_1 \leq \mathrm{cost}(G, \mathcal{S})$ and given that all entries in $u^{G,\mathcal{S}}$ with $p \in C \cap G$ and $C \notin H_{G,\mathcal{S}}$ are zero, we obtain

$$(24) \leq 5\varepsilon.$$

On the other hand, suppose that $\sum_{p\in\Omega} w_p u_p^{G,\mathcal{S}} \leq \|u^{G,\mathcal{S}}\|_1$, then it easily follows that

$$\left| \frac{\sum_{p\in\Omega} w_p u_p^{G,\mathcal{S}} - \|u^{G,\mathcal{S}}\|_1}{\text{cost}(G,\mathcal{S}) + \text{cost}(G,\mathcal{A})} \right| \leq \frac{\|u^{G,\mathcal{S}}\|_1}{\text{cost}(G,\mathcal{S}) + \text{cost}(G,\mathcal{A})} \leq 1.$$

Otherwise, $\sum_{p\in\Omega} w_p u_p^{G,\mathcal{S}} > \|u^{G,\mathcal{S}}\|_1$, and

$$\begin{aligned}
\sum_{p\in\Omega} w_p u_p^{G,\mathcal{S}} &= \sum_{C\in H_{G,\mathcal{S}}} \sum_{p\in C\cap G\cap\Omega} \frac{\text{cost}(G,\mathcal{A})}{\omega \cdot \text{cost}(p,\mathcal{A})} \cdot u_p^{G,\mathcal{S}} \\
&\leq \sum_C \sum_{p\in C\cap G\cap\Omega} \frac{4k\cdot|C\cap G|\cdot\text{cost}(G,\mathcal{A})}{\omega\cdot\text{cost}(p,\mathcal{A})} \cdot u_p^{G,\mathcal{S}} \qquad\text{(Claim D.1)} \\
&\leq 4k\cdot\|u^{G,\mathcal{S}}\|_1,
\end{aligned}$$

which yields

$$\left| \frac{\sum_{p\in\Omega} w_p u_p^{G,\mathcal{S}} - \|u^{G,\mathcal{S}}\|_1}{\text{cost}(G,\mathcal{S}) + \text{cost}(G,\mathcal{A})} \right| \leq \frac{4k\cdot\|u^{G,\mathcal{S}}\|_1}{\text{cost}(G,\mathcal{S}) + \text{cost}(G,\mathcal{A})} \leq 4k.$$

As long as $\omega \geq 9\varepsilon^{-2}k\log\left(4k^2/\varepsilon\right)$, Lemma D.17 guarantees that $\mathbb{P}\left[\bar{\mathcal{E}}_G^M\right] \leq \varepsilon/4k$, and thus,

$$(25) \leq \varepsilon.$$

Rescaling $\varepsilon$ by a factor 6 concludes the proof. $\qquad\square$

### D.8.2. ANALYSIS FOR FAR CLUSTERS

The following derivation is similar to the previous section, with the difference that $C \in F_{G,\mathcal{S}}$. We first state and prove an analog of Lemma D.17, defining

$$\mathcal{F}_G^O = \left\{ \forall i\in[k], \sum_{p\in C_i\cap G\cap\Omega} w_p = \sum_{p\in C_i\cap G\cap\Omega} \frac{\text{cost}(G,\mathcal{A})}{\omega\cdot\text{cost}(p,\mathcal{A})} \in (1\pm\varepsilon)\cdot\text{cost}(C_i\cap G,\mathcal{A}) \right\}.$$

**Lemma D.21.** *Let $G \in G^O$. Then,*

$$\mathbb{P}\left[\mathcal{F}_G^O\right] \geq 1 - k\cdot\exp\left(-\frac{\varepsilon^2\omega}{5k}\right).$$

*Proof.* We need to show that these estimators concentrate around their expectation simultaneously. Consider the $j$-th sampled point $p_j$ in coreset $\Omega$, and let $w_{p_j,C} = w_p \cdot \mathbb{1}_{\{p_j\in C\cap G\}}$. Then,

$$\begin{aligned}
\mathbb{V}\left[w_{p_j,C}\right] \leq \mathbb{E}\left[w_{p_j,C}^2\right] &= \sum_{p\in C\cap G} w_p^2\cdot\mathbb{P}\left[p\in\Omega\right] = \frac{\text{cost}(G,\mathcal{A})}{\omega^2}\cdot\sum_{p\in C\cap G}\text{cost}(p,\mathcal{A}) \\
&\leq \frac{\text{cost}(G,\mathcal{A})}{\omega^2}\cdot\text{cost}(C\cap G,\mathcal{A}) \leq \frac{2k\cdot\text{cost}^2(C\cap G,\mathcal{A})}{\omega^2},
\end{aligned}$$

where the second line of inequalities follows from Claim D.1. Similarly, Claim D.1 also implies

$$w_{p_j,C} \leq \frac{2k\cdot\text{cost}^2(C\cap G,\mathcal{A})}{\omega^2}.$$

Thus, using Bernstein's Inequality (Fact D.1), we obtain

$$\mathbb{P}\left[|\text{cost}(C\cap G\cap\Omega,\mathcal{A}) - \text{cost}(C\cap G,\mathcal{A})| > \varepsilon\cdot\text{cost}(C\cap G,\mathcal{A})\right]$$

$$\leq \exp\left(-\frac{\varepsilon^2\cdot\text{cost}^2(C\cap G,\mathcal{A})}{2\omega\cdot\frac{2k\cdot\text{cost}^2(C\cap G,\mathcal{A})}{\omega^2} + \frac{2}{3}\cdot\frac{2k\cdot\text{cost}(C\cap G,\mathcal{A})}{\omega}\cdot\varepsilon\cdot\text{cost}(C\cap G,\mathcal{A})}\right) \leq \exp\left(-\frac{\varepsilon^2\omega}{5k}\right).$$

A union bound over all clusters in $\mathcal{A}$ yields the lemma. $\qquad\square$

**Lemma D.22.** *Let $G \in G^O$. Then, for any solution $\mathcal{S}$ and any point $p \in C \cap G$ such that $C \in F_{G,\mathcal{S}}$, it holds that*

$$\mathbb{E}_\Omega \left[ \text{cost}(C \cap G, \mathcal{S}) + \sum_{p \in C \cap G \cap \Omega} w_p \cdot \text{cost}(p, \mathcal{S}) \middle| \mathcal{F}_G^O \right] \le \varepsilon \cdot \text{cost}(C, \mathcal{S}).$$

*Proof.* Let us consider a point $p \in C \cap G$ such that $C \in \mathcal{A}$, and let $c$ be the center serving it. We know that $\text{cost}(p, \mathcal{S}) > 4^z \cdot \text{cost}(p, c)$ and thus, by triangle inequality,

$$d(c, \mathcal{S}) \ge d(p, \mathcal{S}) - d(p, c) \ge 4d(p, c) - d(p, c) \ge d(p, c).$$

This implies $\text{cost}(c, \mathcal{S}) \ge \left( \frac{4z}{\varepsilon} \right)^{2z} \cdot \frac{\text{cost}(C, c)}{|C|}$. We now define as

$$C' = \left\{ p' \in C \mid \text{cost}(p', c) \le \left( \frac{2z}{\varepsilon} \right)^z \cdot \frac{\text{cost}(C, c)}{|C|} \right\}.$$

Then, for any $p' \in C'$, applying Fact D.3 with $\vartheta = \varepsilon$ gives

$$\text{cost}(c, \mathcal{S}) \le (1 + \varepsilon) \cdot \text{cost}(p', \mathcal{S}) + \left( \frac{z + \varepsilon}{z} \right)^{z-1} \cdot \text{cost}(p', c)$$

$$\le (1 + \varepsilon) \cdot \text{cost}(p', \mathcal{S}) + \left( \frac{z + \varepsilon}{z} \right)^{z-1} \cdot \left( \frac{2z}{\varepsilon} \right)^z \cdot \frac{\text{cost}(C, c)}{|C|} \qquad (p' \in C')$$

$$\le (1 + \varepsilon) \cdot \text{cost}(p', \mathcal{S}) + \frac{\left( \frac{4z}{\varepsilon} \right)^{2z-1} \cdot \frac{\text{cost}(C, c)}{|C|}}{\text{cost}(p, c)} \cdot \text{cost}(p, c)$$

$$\le (1 + \varepsilon) \cdot \text{cost}(p', \mathcal{S}) + \varepsilon \cdot \text{cost}(p, c) \qquad (p \in G \text{ where } G \in G^O)$$

$$\le (1 + \varepsilon) \cdot \text{cost}(p', \mathcal{S}) + \varepsilon \cdot \text{cost}(c, \mathcal{S}),$$

which, in turn, means

$$\text{cost}(p', \mathcal{S}) \ge \frac{1 - \varepsilon}{1 + \varepsilon} \cdot \text{cost}(c, \mathcal{S}).$$

Moreover, we have that, for any $C \in F_{G,\mathcal{S}}$,

$$\text{cost}(C, \mathcal{S}) \ge \text{cost}(C', \mathcal{S}) = \sum_{p' \in C'} \text{cost}(p', \mathcal{S}) \ge |C'| \cdot \frac{1 - \varepsilon}{1 + \varepsilon} \cdot \text{cost}(c, \mathcal{S}) \tag{26}$$

$$\ge |C'| \cdot \frac{1 - \varepsilon}{1 + \varepsilon} \cdot \left( \frac{4z}{\varepsilon} \right)^{2z} \cdot \frac{\text{cost}(C, c)}{|C|} \ge \left( \frac{4z}{\varepsilon} \right)^{2z-1} \cdot \text{cost}(C, c), \tag{27}$$

where the third and last inequalities follow from the fact that $|C \cap G| \le \left( \frac{\varepsilon}{4z} \right)^{2z} \cdot |C|$ and $|C'| \ge (1 - \varepsilon) \cdot |C|$ by Markov's inequality. We, thus, have

$$\text{cost}(C \cap G, \mathcal{S}) \le (1 + \varepsilon) \cdot \sum_{p \in C \cap G \cap \Omega} \text{cost}(c, \mathcal{S}) + \left( \frac{2z + \varepsilon}{\varepsilon} \right)^{z-1} \cdot \text{cost}(p, c) \qquad \text{(Fact D.3)}$$

$$\le |C \cap G| \cdot (1 + \varepsilon) \cdot \text{cost}(c, \mathcal{S}) + \left( \frac{2z + \varepsilon}{\varepsilon} \right)^{z-1} \cdot \text{cost}(C \cap G, c)$$

$$\le \left( \frac{\varepsilon}{4z} \right)^{2z} \cdot |C| \cdot (1 + \varepsilon) \cdot \text{cost}(c, \mathcal{S}) + \left( \frac{2z + \varepsilon}{\varepsilon} \right)^{z-1} \cdot \text{cost}(C \cap G, c) \qquad \text{(Markov's Inequality)}$$

$$\le \left( \frac{\varepsilon}{4z} \right)^{2z} \cdot |C'| \cdot \frac{1 + \varepsilon}{1 - \varepsilon} \cdot \text{cost}(c, \mathcal{S}) + \left( \frac{2z + \varepsilon}{\varepsilon} \right)^{z-1} \cdot \text{cost}(C \cap G, c) \qquad \text{(Markov's Inequality)}$$

$$\le \left( \frac{1 + \varepsilon}{1 - \varepsilon} \right)^2 \cdot \left( \frac{\varepsilon}{4z} \right)^{2z} \cdot \text{cost}(C, \mathcal{S}) + \left( \frac{2z + \varepsilon}{\varepsilon} \right)^{z-1} \cdot \text{cost}(C \cap G, c) \qquad \text{(Equation 26)}$$

$$\leq \left(\frac{1+\varepsilon}{1-\varepsilon}\right)^2 \cdot \left(\frac{\varepsilon}{4z}\right)^{2z} \cdot \text{cost}(C, \mathcal{S}) + \left(\frac{2z+\varepsilon}{\varepsilon}\right)^{z-1} \cdot \left(\frac{\varepsilon}{4z}\right)^{2z-1} \cdot \text{cost}(C, \mathcal{S}) \qquad \text{(Equation 27)}$$

$$\leq \varepsilon \cdot \text{cost}(C, \mathcal{S}). \tag{28}$$

We would now like to bound the overall cost contribution of points in $C \cap G \cap \Omega$: To that end, observe that

$$\sum_{p \in C \cap G \cap \Omega} \frac{\text{cost}(G, \mathcal{A})}{\omega \cdot \text{cost}(p, \mathcal{A})} \leq \left(\frac{\varepsilon}{2z}\right)^{2z} \cdot \frac{|C|}{\text{cost}(C, \mathcal{A})} \cdot \text{cost}(G, \mathcal{A}) \qquad \left(\left(\frac{2z}{\varepsilon}\right)^{2z} \cdot \frac{\text{cost}(C, \mathcal{A})}{|C|} \leq \text{cost}(p, \mathcal{A})\right)$$

$$\leq (1+\varepsilon) \cdot \left(\frac{\varepsilon}{2z}\right)^{2z} \cdot \frac{|C|}{\text{cost}(C, \mathcal{A})} \cdot \text{cost}(C \cap G, \mathcal{A})$$

$$\leq (1+\varepsilon) \cdot \left(\frac{\varepsilon}{2z}\right)^{2z} \cdot |C|. \tag{29}$$

Therefore, under event $\mathcal{F}_G^O$, we can upper bound the estimated induced cluster cost as follows:

$$\text{cost}(C \cap G \cap \Omega, \mathcal{S}) = \sum_{p \in C \cap G \cap \Omega} \frac{\text{cost}(G, \mathcal{A})}{\omega \cdot \text{cost}(p, \mathcal{A})} \cdot \text{cost}(p, \mathcal{S})$$

$$\leq \sum_{p \in C \cap G \cap \Omega} \frac{\text{cost}(G, \mathcal{A})}{\omega \cdot \text{cost}(p, \mathcal{A})} \cdot \left((1+\varepsilon) \cdot \text{cost}(c, \mathcal{S}) + \left(\frac{z+\varepsilon}{\varepsilon}\right)^{z-1} \cdot \text{cost}(p, c)\right) \qquad \text{(Fact D.3)}$$

$$\leq (1+\varepsilon) \cdot \text{cost}(c, \mathcal{S}) \cdot \sum_{p \in C \cap G \cap \Omega} \frac{\text{cost}(G, \mathcal{A})}{\omega \cdot \text{cost}(p, \mathcal{A})}$$

$$+ \left(\frac{z+\varepsilon}{\varepsilon}\right)^{z-1} \cdot (1+\varepsilon) \cdot \text{cost}(C \cap G, \mathcal{A}) \qquad (\mathcal{F}_G^O \text{ holds})$$

$$\leq (1+\varepsilon)^2 \cdot \text{cost}(c, \mathcal{S}) \cdot \left(\frac{\varepsilon}{2z}\right)^{2z} \cdot |C| \qquad \text{(Equation 29)}$$

$$+ \left(\frac{z+\varepsilon}{\varepsilon}\right)^{z-1} \cdot \underbrace{(1+\varepsilon) \cdot \text{cost}(C \cap G, \mathcal{A})|}_{\leq \text{cost}(C, c)}$$

$$\leq \varepsilon \cdot \text{cost}(C, \mathcal{S}), \tag{30}$$

where the second inequality holds for all clusters $C$ simultaneously since event $\mathcal{F}_G^O$ holds, and the last by an identical derivation to the one yielding $\text{cost}(C \cap G, \mathcal{S}) \leq \varepsilon \cdot \text{cost}(C, \mathcal{S})$ above. The lemma follows by summing

$$(28) + (30) \leq 2\varepsilon \cdot \text{cost}(C, \mathcal{S}),$$

and rescaling $\varepsilon$ by a factor 2. $\qquad\square$

We proceed with the proof of Lemma D.16.

*Proof of Lemma D.16.* By Law of Total Expectation, we write

$$\mathbb{E}_\Omega \left[\sup_\mathcal{S} \left|\frac{\sum_{p \in \Omega} w_p u_p^{G, \mathcal{S}} - \|u^{G, \mathcal{S}}\|_1}{\text{cost}(P^G, \mathcal{S}) + \text{cost}(P^G, \mathcal{A})}\right|\right] = \mathbb{E}_\Omega \left[\sup_\mathcal{S} \left|\frac{\sum_{p \in \Omega} w_p u_p^{G, \mathcal{S}} - \|u^{G, \mathcal{S}}\|_1}{\text{cost}(P^G, \mathcal{S}) + \text{cost}(P^G, \mathcal{A})}\right| \Bigg| \mathcal{F}_G^O\right] \cdot \mathbb{P}\left[\mathcal{F}_G^O\right] \tag{31}$$

$$+ \mathbb{E}_\Omega \left[\sup_\mathcal{S} \left|\frac{\sum_{p \in \Omega} w_p u_p^{G, \mathcal{S}} - \|u^{G, \mathcal{S}}\|_1}{\text{cost}(P^G, \mathcal{S}) + \text{cost}(P^G, \mathcal{A})}\right| \Bigg| \bar{\mathcal{F}}_G^O\right] \cdot \mathbb{P}\left[\bar{\mathcal{F}}_G^O\right]. \tag{32}$$

Trivially, $\mathbb{P}\left[\mathcal{F}_G^O\right] \leq 1$ and, conditioned on $\mathcal{F}_G$,

$$\sum_{p \in \Omega} w_p u_p^{G, \mathcal{S}} - \|u^{G, \mathcal{S}}\|_1 = \sum_{C \in F_{G, \mathcal{S}}} \sum_{p \in C \cap G \cap \Omega} w_p u_p^{G, \mathcal{S}} + \sum_{p \in C \cap G} \text{cost}(p, \mathcal{S})$$

$$\leq \varepsilon \cdot \sum_{C \in F_{G,\mathcal{S}}} \text{cost}(C \cap G, \mathcal{S}) = \varepsilon \cdot \text{cost}(P^G, \mathcal{S}),$$

by Lemma D.22. Since $\|u^{G,\mathcal{S}}\|_1 \leq \text{cost}(G, \mathcal{S})$ and given that all entries in $u^{G,\mathcal{S}}$ with $p \in C \cap G$ and $C \notin F_{G,\mathcal{S}}$ are zero, we obtain

$$(31) \leq \varepsilon.$$

On the other hand, suppose that $\sum_{p \in \Omega} w_p u_p^{G,\mathcal{S}} \leq \|u^{G,\mathcal{S}}\|_1$, then it easily follows that

$$\left| \frac{\sum_{p \in \Omega} w_p u_p^{G,\mathcal{S}} - \|u^{G,\mathcal{S}}\|_1}{\text{cost}(P^G, \mathcal{S}) + \text{cost}(P^G, \mathcal{A})} \right| \leq \frac{\|u^{G,\mathcal{S}}\|_1}{\text{cost}(P^G, \mathcal{S}) + \text{cost}(P^G, \mathcal{A})} \leq 1.$$

Otherwise, $\sum_{p \in \Omega} w_p u_p^{G,\mathcal{S}} > \|u^{G,\mathcal{S}}\|_1$, and consider the maximum ratio $r_C = \max_{p \in C \cap G} \frac{\text{cost}(p,\mathcal{S})}{\text{cost}(p,\mathcal{A})} > 4^z$ as well as the corresponding maximizing point $p^*$. We have that

$$d(c, \mathcal{S}) \geq d(p^*, \mathcal{S}) - d(p^*, c) \geq (r_C^{1/z} - 1) \cdot d(p^*, c),$$

i.e., $r_C \leq 2^z \cdot \frac{\text{cost}(c,\mathcal{S})}{\text{cost}(p^*,c)}$. Thus,

$$\begin{aligned}
\sum_{p \in \Omega} w_p u_p^{G,\mathcal{S}} &= \sum_{C \in F_{G,\mathcal{S}}} \sum_{p \in C \cap G \cap \Omega} \frac{\text{cost}(G, \mathcal{A})}{\omega \cdot \text{cost}(p, \mathcal{A})} \cdot u_p^{G,\mathcal{S}} \\
&\leq \sum_C \sum_{p \in C \cap G \cap \Omega} \frac{2k \cdot \text{cost}(C \cap G, \mathcal{A})}{\omega \cdot \text{cost}(p, \mathcal{A})} \cdot \text{cost}(p, \mathcal{S}) && \text{(Claim D.1)} \\
&\leq \frac{2k}{\omega} \cdot \sum_C |C \cap G \cap \Omega| \cdot r_C \cdot \text{cost}(C \cap G, \mathcal{A}) \\
&\leq 2k \cdot \sum_C r_C \cdot \text{cost}(C \cap G, \mathcal{A}) \\
&\leq 2^{z+1} k \cdot \sum_C \frac{\text{cost}(C \cap G, \mathcal{A})}{\text{cost}(p^*, \mathcal{A})} \cdot \text{cost}(c, \mathcal{S}) \\
&\leq 2^{z+1} k \cdot \sum_C \left( \frac{\varepsilon}{4z} \right)^{2z} |C| \cdot \text{cost}(c, \mathcal{S}) && \text{(Markov's Inequality)} \\
&\leq 2^{2z+1} k \cdot \sum_C (\text{cost}(C, \mathcal{S}) + \text{cost}(C, \mathcal{A})) && \text{(Fact D.3)} \\
&\leq 2^{2z+1} k \cdot (\text{cost}(P^G, \mathcal{S}) + \text{cost}(P^G, \mathcal{A})).
\end{aligned}$$

This yields

$$\left| \frac{\sum_{p \in \Omega} w_p u_p^{G,\mathcal{S}} - \|u^{G,\mathcal{S}}\|_1}{\text{cost}(P^G, \mathcal{S}) + \text{cost}(P^G, \mathcal{A})} \right| \leq 2^{2z+1} k \cdot \frac{\text{cost}(P^G, \mathcal{S}) + \text{cost}(P^G, \mathcal{A})}{\text{cost}(P^G, \mathcal{S}) + \text{cost}(P^G, \mathcal{A})} = 2^{2z+1} k.$$

As long as $\omega \geq 5k \log \left( \frac{k^2}{2^{2z+1}\varepsilon} \right)$, Lemma D.21 guarantees that $\mathbb{P}\left[ \bar{\mathcal{F}}_G^O \right] \leq \frac{\varepsilon}{2^{2z+1}k}$, and thus,

$$(32) \leq \varepsilon.$$

Rescaling $\varepsilon$ by a factor 2 concludes the proof. $\qquad\square$

