# OpenReview forum: "Terminal Dimension Reduction for Time Series with Applications"
_ICML.cc/2026/Conference — ICML 2026 regular_

### Official Review · Reviewer_4qgM · 2026-03-06

**Soundness:** 3
**Presentation:** 4
**Significance:** 3
**Originality:** 4
**Overall Recommendation:** 4
**Confidence:** 4

**Summary:**

## Summary

This paper studies dimensionality reduction for high-dimensional polygonal curves under the continuous Fréchet distance. The core challenge is that the Fréchet distance depends on all points along curve segments, not just vertices, rendering standard point-wise techniques insufficient.
The authors propose the first terminal embedding for the Fréchet distance, using distinct mappings for input curves (fixed) and queries (adaptive). Their construction combines subspace-preserving sketches with an orthogonal residual term to approximately preserve all point-to-point distances, yielding (1±ε) -approximation in target dimension polynomial in the query complexity and logarithmic in input size, independent of the ambient dimension d .
As an application, they obtain dimension-free coresets for (k,ℓ,z) -clustering, improving prior bounds linear in d . Experiments on greenhouse gas concentration data demonstrate that Johnson–Lindenstrauss projections outperform PCA for preserving Fréchet distances.

**Compliance With Llm Reviewing Policy:**

Affirmed.

**Key Questions For Authors:**

## Key Questions For Authors
### Q1: Is the quartic dependence on the precision parameter inherent to the continuous Fréchet setting?

The target dimension scales linearly with the query complexity and quartically with the inverse precision, which is significantly worse than the optimal quadratic dependence for point-set terminal embeddings. Do the authors believe a lower bound exists that justifies this gap, perhaps due to preserving distances over uncountably many points along curve segments? Alternatively, could tighter analysis via improved covering arguments or alternative sketching strategies reduce the exponent? A response indicating that this dependence is likely improvable would strengthen confidence in the practical relevance of the approach.

### Q2: Can the terminal embedding guarantee be validated experimentally on input-query distance preservation?

The current experiments only evaluate pairwise distances among input curves, which tests a standard Johnson-Lindenstrauss property rather than the asymmetric input-query distances that define terminal embeddings. Could the authors provide even a small-scale experiment where a fixed sketch is built from input curves, new query curves not in the input set are embedded via the adaptive mapping, and the Fréchet distance between original and embedded input-query pairs is compared? Such validation would directly support the paper's central claim and address concerns about empirical grounding.

### Q3: What is the practical computational model for the orthogonal residual term?

Lemma 3.2's decomposition includes a residual component that is critical to the theoretical guarantee but appears expensive to compute for arbitrary queries, since it seemingly requires access to the full high-dimensional query point. Does the method assume queries arrive in the original ambient space so residuals can be computed on-the-fly, or is there a way to approximate or embed the residual itself without maintaining the high-dimensional representation? Clarifying the query model and storage assumptions would help assess feasibility in settings where high-dimensional data is costly to maintain.

### Q4: How does the coreset construction compare empirically to prior dimension-dependent methods?

While the paper proves a dimension-free coreset bound, no quantitative results are shown regarding approximation error versus coreset size or runtime gains. For typical parameters from the experimental setting, is the resulting coreset actually smaller than prior constructions in practice? And does it yield comparable or better clustering accuracy? Evidence here would strengthen the significance claim beyond asymptotic analysis.
A thoughtful response to these questions—particularly regarding experimental validation of the terminal embedding property and empirical coreset evaluation—could substantially improve my assessment of the paper's empirical credibility and practical value.

**Limitations:**

Yes

**Strengths And Weaknesses:**

## Strengths And Weaknesses
### Soundness

The technical content is rigorous, building correctly on established tools from metric embedding and computational geometry.

The work is technically sound. The core argument—decomposing distances via a low-rank projection Π spanned by O(ℓε⁻²) input segments, then controlling residuals through subspace-preserving sketches—builds correctly on established tools from metric embedding and computational geometry. Proofs follow logically from covering arguments and JL transform properties. Assumptions (polygonal curves with bounded complexity m, ℓ) are standard for this domain. The authors properly distinguish their continuous setting from prior discrete Fréchet results.

However, significant limitations temper the technical achievement. The target dimension scales as O(ℓε⁻⁴ log(nm)), with the ε⁻⁴ dependence representing a substantial gap from the O(ε⁻² log n) optimal bound for point-set terminal embeddings. Whether this quartic dependence is inherent or an artifact of the analysis remains unaddressed—no lower bounds are provided. Empirically, the evaluation uses only n=16 curves and validates only pairwise distances between input curves (a standard JL property), not the input-query distance preservation that defines terminal embeddings. The central claim—effective handling of future query curves—thus lacks experimental support. The coreset application shows no quantitative results (e.g., clustering accuracy or compression tradeoffs), and comparisons to time-series-specific baselines (e.g., DTW, PAA) are absent.

### Presentation

Presentation is generally clear and well-structured. The paper effectively positions itself against related work, distinguishing point-set terminal embeddings, discrete Fréchet results, and prior high-dimensional coreset constructions. Geometric intuition (“covering query segments with input segments”) aids understanding. However, Lemma 3.2’s approximate decomposition could be labeled more explicitly to prevent misreading as an exact equality. The experimental section should clarify what is being validated (and what is omitted), rather than implying general applicability from limited JL comparisons.

### Significance

The problem addressed—high-dimensional trajectory analysis under Fréchet distance—is relevant to climate science, robotics, and geospatial applications where dimensionality bottlenecks are real. Removing linear dependence on ambient dimension d in coreset bounds represents meaningful theoretical progress. Yet practical utility is constrained by prohibitive preprocessing costs (O(nmdℓ²ε⁻⁶ log(nm)) per Proposition 3.4) and the high ε⁻⁴ factor. The asymmetric embedding design is elegant, but the “existence-only” caveat suggests immediate deployment is not intended.

### Originality
Originality lies in extending terminal embeddings from discrete points to continuous curve structures—a non-trivial step requiring novel handling of uncountably many point pairs. The asymmetric mapping design (fixed f for inputs, adaptive g for queries) and explicit orthogonal residual treatment represent thoughtful adaptations rather than straightforward combinations. While building on standard primitives (JL sketches, covering numbers), the paper reconceptualizes their application to preserve the geometry of linear interpolation, clearly distinguishing it from prior discrete Fréchet embeddings.

Overall, this is competent theoretical work that advances understanding of geometric sketching for structured data. Its value rests primarily on conceptual novelty and proof technique rather than practical algorithms or compelling empirical demonstration. For ICML, it fits best if the community prioritizes foundational geometric methods over immediate applicability.

The main weakness of the paper is the lack of experimental validation for the terminal embedding property, which is the central theoretical claim of the work.

---

> ### Author Rebuttal · Authors · 2026-03-31
>
> 1. For our analysis, the quartic $\epsilon$-dependence is necessary. It is also natural in the context of terminal embeddings as certain techniques require this (cf. discussion in Mahabadi et al. 2018); more involved techniques are required to reach a quadratic dependency, see (Narayanan and Nelson 2019) for point embeddings, but it is currently unclear if they can be extended to work for curves.
>
> 2. This problem is naturally in co-NP. It is difficult to design a heuristic for evaluation. For a random vector, the standard guarantee of JL will ensure a good quality. Sampling random vectors until finding a "bad" one would require sampling as many as several polynomial orders more vectors than the input, which is practically not viable.
>
> 3. Beyond the fact that they are theoretically interesting and intriguing objects, the main practical interest of terminal embeddings is their existence, which allows to show correctness and tighter bounds for coreset algorithms. For this, we do not need to actually build the embedding of any curve. Even for terminal embeddings of points in $\mathbb{R}^d$ with $\ell_2$ distance, all terminal embeddings require to store the full input. Cherapanamjeri and Nelson (FOCS 2021) showed how to compute the embedding of any point in sublinear time, although they still need memory size $O(nd)$. Now we proved their existence for Frechet distance, and it is a great follow-up research question how to build them more efficiently, see Proposition 3.4 and the comments below.
>
> 4. All coreset algorithms are identical: they use the same probability distribution, and sample from it. In practice, coresets are computed by sampling a number of points dictated by the target compressed size. We show a strict improvement on previous theoretical analyses, but our result does not change the actual practical performance as the algorithm remains the same. Thus, it is difficult to introduce new experiments; this known coreset algorithm has already been shown to outperform competitors (Schwiegelshohn and Sheikh-Omar, 2022; Cohen-Addad et al. NeurIPS 2022, Draganov, Saulpic, Schwiegelshohn 2024).

---

> > ### Author Rebuttal · Reviewer_4qgM · 2026-04-02
> >
> > The rebuttal clarifies some aspects of the theoretical contribution, particularly regarding the ε⁻⁴ dependence. However, it does not sufficiently address my main concern about the lack of experimental validation of the terminal embedding property, especially for input-query distances.
> >
> > While I acknowledge that empirical evaluation of worst-case guarantees is challenging, even a small-scale validation would strengthen confidence in the practical relevance of the approach.
> >
> > I still view the paper as a solid theoretical contribution, but its empirical support remains limited.

---

> > > ### Author Response · Authors · 2026-04-03
> > >
> > > We added the requested small-scale experiment. We generated 10 input curves, and one query curve, each of length 5, in 50 dimensions, each with 10 repetitions.
> > >
> > > We reduced them to different target dimensions $t$. To this end, input curves were embedded via a Gaussian JL matrix $S$. For the query curve, the subspace matrix $\Pi$ was constructed from input points, and then embedded to obtain $S\Pi q$.
> > >
> > > The table shows average and maximum approximation ratios between original Frechet distance and terminal embedded (TE) Frechet distance, over 10 repetitions for each target dimension 5, 10, and 20. We put the corresponding numbers for plain JL, and PCA experiments next to TE, just to highlight that the performance of TE is very comparable to plain JL as expected and explained in our original response; it would be hard to find a "bad" one even if there were any.
> > >
> > > | Approximation ratios | TE5      | JL5      | PCA5     | TE10     | JL10     | PCA10    | TE20     | JL20     | PCA20    |
> > > |----------------------|----------|----------|----------|----------|----------|----------|----------|----------|----------|
> > > | Average              | 1.204975 | 1.241320 | 2.112083 | 1.153079 | 1.141533 | 1.524308 | 1.105663 | 1.134922 | 1.259703 |
> > > | Maximum              | 2.002154 | 1.860457 | 5.181483 | 1.594851 | 1.491506 | 4.605984 | 1.471862 | 1.351461 | 2.901826 |

---

### Official Review · Reviewer_xH9n · 2026-03-09

**Soundness:** 4
**Presentation:** 3
**Significance:** 3
**Originality:** 2
**Overall Recommendation:** 4
**Confidence:** 3

**Summary:**

The paper investigates the application of terminal dimensionality reduction in time series analysis. The authors extend the approach to affine line segments and use it to derive dimension-free coreset constructions for time series clustering under the Fréchet distance. Furthermore, they evaluate Johnson–Lindenstrauss embeddings based on the proposed approach on a public dataset and compare the results with PCA-based dimensionality reduction across different target dimensions.

**Compliance With Llm Reviewing Policy:**

Affirmed.

**Final Justification:**

The rebuttal provides helpful clarifications, particularly regarding cost-related aspects, and addresses several of the initial concerns. The manuscript presents a strong theoretical contribution; however, some questions remain regarding its practical applicability to large-scale time-series data. Accordingly, the evaluation is kept unchanged.

**Key Questions For Authors:**

1. How does the proposed method compare to PCA in terms of computational cost, and how does this trade-off affect its practical applicability?
2. Can the authors provide pseudocode or a clearer algorithmic description to improve the reproducibility and implementability of the proposed method?
3. Can the experimental evaluation be extended to demonstrate the impact of the proposed reduction method on actual time series clustering performance (e.g., using approximation ratios or clustering quality metrics)?

**Limitations:**

The authors did not discuss the impact statement.

**Strengths And Weaknesses:**

Soundness:
The proposed approach is theoretically investigated in detail, and the authors provide mathematical proofs for all implementation aspects, including the method's complexity. In addition, an experiment (and an anonymous repository) is provided, which strengthens the implementability of the proposed method.

Presentation:
The overall presentation of the proposed approach does not sufficiently emphasize the application aspect. The target problem concerns time series; however, neither the methodology nor the experimental section provides discussion of application-related issues. An anonymous code repository is provided, but including pseudocode (or a flowchart) could improve the readability of the paper for those who may want to implement the method in their research. A conclusion and future work section is also required to highlight the possible future directions of the research.

Significance:
Time series analysis is always open to novel and efficient approaches. The proposed approach appears to be a candidate for a novel implementation, and one indicator of this is the demonstrated experiment. The experiments show that the proposed approach is superior to the baseline approach, PCA, for different reduction orders. However, the cost of the improvement in performance is not discussed in the manuscript. Since both the JL and Fréchet approaches are computationally more expensive compared to the PCA method, the authors should discuss this trade-off to improve the applicability of their approach.

Originality:
The method combines previously suggested approaches in a new way, supported by corresponding proofs. However, the computational efficiency of the proposed method should be demonstrated to improve its applicability. Another issue that limits the novelty of the approach concerns the practical implementation. The authors mention that they consider a clustering problem, and the approximation ratio could serve as an indicator for evaluating the performance of the reduction. However, the experiments should be extended to demonstrate the benefit of using the proposed approach for the time series clustering problem.

---

> ### Author Rebuttal · Authors · 2026-03-31
>
> 1. The cost of the simple random projection, which can be done with standard matrix multiplication, is O(Ndt) for reducing N points from original dimension d to target dimension t. PCA has the same asymptotic complexity. Notably, the wall clock times were negligible compared to the time required to compute the Frechet distance, and took only a few seconds on all instances both for PCA and JL. Compared with PCA, random projections (RP) are one order of magnitude faster: it took 7.9576 seconds for PCA and 0.4064 seconds for RP, across the iterations. We will add all relevant numbers to the next revision of the paper.
> 4. We will add a pseudo-code of our experimental algorithm, in addition to the code provided online. The dimension reduction method is: generate a $d \times t$ matrix (where $t$ is the desired target dimension) with each entry being an i.i.d Gaussian. Then, multiply the data matrix with this one.
> 5. For the purpose of our experiment, we indeed only performed evaluation of distance distortion. If we understand the reviewer correctly, they are asking an experimental evaluation of the impact of terminal embedding for coresets for clustering with Frechet distance. It is difficult to imagine what such an experiment would look like, as the coreset algorithm itself (and its performance) does not change. Our work improves the theoretically known sample size bounds achieved by the same coreset algorithm, where previous theoretical bounds were loose. We will point this out more clearly and refer to existing experimental work on sample sizes and  coreset quality that were done in (Schwiegelshohn and Sheikh-Omar, 2022; Cohen-Addad et al. NeurIPS 2022, Draganov, Saulpic, Schwiegelshohn 2024). We could repeat their experiments, but in our opinion it would not constitute original work, hence we decided to use the limited space we have on novel and original content.

---

> > ### Author Rebuttal · Reviewer_xH9n · 2026-04-04
> >
> > I thank the authors for their detailed and constructive rebuttal. In particular, the clarifications regarding the cost-related aspects are helpful and, in my view, adequately address the corresponding concerns. The manuscript presents a strong and well-articulated theoretical contribution. That said, further clarification on how the proposed approach can be applied in practice for large-scale time-series data would enhance the overall impact of the work.

---

### Official Review · Reviewer_QV1L · 2026-03-11

**Soundness:** 1
**Presentation:** 1
**Significance:** 1
**Originality:** 2
**Overall Recommendation:** 2
**Confidence:** 3

**Summary:**

This paper addresses some form of dimension reduction technique for time series, called "terminal embeddings". Whilst I am able to make some sense of the mathematical statements in the paper (not through lack of mathematical experience), I was not able to understand what a terminal embedding is in practical terms and how it is put to use. This is an obstacle to me presenting a meaningful summary of the paper.

**Compliance With Llm Reviewing Policy:**

Affirmed.

**Final Justification:**

The rebutall provided some clarifications but my overall concerns about the paper still hold, so I have lowered my confidence from 4 to 3.

**Key Questions For Authors:**

As mentioned above, I think the paper is severely lacking in information about how to make use of terminal embeddings in practice. I could ask the authors to provide more information about this, but I think I am unlikely to change opinion on the paper without it being very significantly revised.

**Limitations:**

I found no explicit discussion of limitations or disadvantages.

**Strengths And Weaknesses:**

Putting myself in the position of someone who might be interested in using terminal embeddings in practice, I found it difficult to any extract practical insight from the paper. Some of the content seems very confusing in this regard. For example, on page 7,

“We first emphasize the fact that most applications of terminal embeddings, in particular their application coreset …. only require the existence of such an embedding and do not need an explicit construction.”

I am unable to make sense of this. How can any method be used in practice only on the basis of a mathematical proof of its existence, rather than any construction of the method?

This one illustrative extract is indicative of why I am not able to assess the content of the paper is sound, in the sense of having a clear, logical connection to some practical method.

I found the writing style of the paper to be  long-winded and meandering. The first few pages touch on a wide range of background issues rather than focusing simply and clearly on main messages.

The section “Terminal Embeddings for the Frechet Distance” is similarly meandering and reads more as a stream of consciousness rather than helpful, simple, clear and succinct directions for the interested researcher.

The experimental illustration section doesn’t help me either, from this section I was not able to discern what the practical method(s) under the consideration is, other than some reference to existing methods of random projections (“JL embeddings“) and PCA.

Because of this gross lack of clarity the only conclusion I think I can reasonably draw is that he paper is weak in terms of soundness and significance.

---

> ### Author Rebuttal · Authors · 2026-03-31
>
> We apologize for not making accessible to a wide audience, and will do our best to do so within given page limits in the next revision of our paper.
>
> Regarding the comment on why existence is enough for practice: the mere existence of a terminal embedding onto small dimension is key to prove that the coreset algorithm is correct (and thus that we can compute, in practice, coresets with small size). However the coreset algorithm itself *does not* need to build the terminal embedding; this is only required for its theoretical analysis, respectively to achieve dimension-independent coreset size bounds.
>
> The use of existential results such as this one has a long history in theoretical computer science in general, and coreset literature in particular. Typically, in learning theory literature, generalization results rely on the existence of small covers over all candidate hypotheses, even if those cannot be constructed explicitly or efficiently. Modern coreset constructions rely on the same techniques, and our terminal embedding is used in the formal analysis to show the existence of a small cover.
> The following coreset papers all rely *solely on the existence* of terminal embeddings for points (Huang and Vishnoi 2020, Cohen-Addad et al. 2021, Cohen-Addad et al. NeurIPS 2022, Huang et al. 2024, Bansal et al. 2024).
>
> We note that the coreset algorithm itself, implemented and analyzed e.g. in (Schwiegelshohn and Sheikh-Omar, 2022) is constructive and fast. One of our contributions is thus to show that this algorithm can be used for Frechet distance to build significantly smaller coreset.
> Coresets put into practice help to compute a small summary of a dataset, and to compute---faster and with smaller memory---a clustering solution with great accuracy. For further discussions on coresets, we refer to the surveys (Feldman, 2020) and (Munteanu, Schwiegelshohn 2018) among others. We will emphasize the practical interest of coresets.

---

> > ### Author Rebuttal · Reviewer_QV1L · 2026-04-02
> >
> > Thanks to the authors for the clarifications, which I will reflect in lowering the confidence from my previous score. My overall concerns about the paper still hold.

---

### Official Review · Reviewer_eBi4 · 2026-03-12

**Soundness:** 4
**Presentation:** 3
**Significance:** 3
**Originality:** 3
**Overall Recommendation:** 5
**Confidence:** 5

**Summary:**

This paper considers terminal dimension and coresets for curve clustering under Frechet distance. Terminal embedding has been applied to construct near-optimal coresets for k-means/k-median clustering recently. However, for curve clustering under Frechet distance, the non-linearity of terminal embedding is a main bottleneck, and the SOTA coreset size has linear dependence on the dimension.

The main contribution of this paper is a new terminal embedding for Frechet distance, which enables authors to construct dimension-free coresets. This result improves all previous results. The construction and analysis rely on the subspace distance-preserving property of the JL transformation. A simple experiment has been conducted to compare the performance of JL and PCA to preserve pairwise Frechet distance.

**Compliance With Llm Reviewing Policy:**

Affirmed.

**Final Justification:**

I have read the rebuttal and other reviews. I think this is an interesting theory paper and wish to keep my positive score.

**Key Questions For Authors:**

1. Can you give a formal algorithm of your coreset construction? Is your terminal embedding implicit in the coreset construction?

2. In the experiment, PCA is chosen as a benchmark algorithm to reduce the dimension for polygon curves. Could you please describe the algorithm (maybe a heuristic)?

**Limitations:**

yes

**Strengths And Weaknesses:**

Strengths:

1. Strong theory results. Improving SOTA results on coresets for curve clustering under Frechet distances.
2. New techniques are introduced. The construction of a new terminal embedding for the Frechet distance is interesting and may benefit future research on this problem.
3. The paper is mostly well written, though a bit technical. It is easy to follow for people familiar with recent coreset literature.

Weaknesses:

1. I feel that the experimental result of this paper is weak. Showing JL has better performance than PCA in practice seems to have little to do with the main result in this paper. As the main contribution lies on the theoretical side, I think it is ok to simply remove this part.

---

> ### Author Rebuttal · Authors · 2026-03-31
>
> **Question 1:** The coreset construction algorithm is the one from (Cohen-Addad et al., 2021). Although it requires a few definitions, we can explain it at high level as follows:
>
> First, compute a bi-criteria approximate solution (with cost $O$(OPT) and using $O(k)$ centers), using (Conradi et al., 2024). Then, partition each cluster of that solution into annuli of exponentially growing radius. Group the radius of the different clusters into "group", with the same ratio of (radius of the annulus) / (average distance to the center in the cluster). This creates $\tilde O(1)$ groups: in each of them, take a uniform sample.
>
> We will add this high-level description to the next version of our paper.
>
> The terminal embedding is implicitly used in the analysis; the coreset algorithm does not need to construct it.
>
> **Question 2:**
>
> The algorithm is a standard PCA on the matrix comprising all vertices of all input curves. We agree with the reviewer assessment on the experiments. We added them only in response to a concern we regularly received on previous publications, where reviewers asked for comparison with PCA.

---

> > ### Author Rebuttal · Reviewer_eBi4 · 2026-04-03
> >
> > Thanks for the answers.

---

### Decision · Program_Chairs · 2026-04-30

**Decision:**

Accept (regular)

**Comment:**

This paper proposes the first terminal embedding for the continuous Fréchet distance, and in turn using it to derive dimension-free coreset constructions for curve clustering. This theoretical contribution leads to improving prior bounds that depended linearly on the ambient dimension. Reviewers comment that the core technical results are rigorous and sufficiently novel, with proof techniques that thoughtfully adapt metric embedding tools to the continuous curve setting. They also largely agree that the paper’s practical aspects are underdeveloped, with some commenting on the opaque presentation and the lack of a concrete or explicit algorithmic description of how the terminal embedding or coreset is used. While the reviews and scores are high, I want to emphasize that the experimental section is limited and does not directly validate the main terminal embedding guarantee or demonstrate concrete benefits for downstream clustering tasks, so this should be improved. Nonetheless, the paper has interesting theoretical contributions and the reviewers are largely in consensus that it is a meaningful advance in the area.